# On the Convergence of Continuous Single-timescale Actor-critic

**Xuyang Chen** [1]   **Lin Zhao** [1]

## Abstract

Actor-critic algorithms have been instrumental in boosting the performance of numerous challenging applications involving continuous control, such as highly robust and agile robot motion control. However, their theoretical understanding remains largely underdeveloped. Existing analyses mostly focus on finite state-action spaces and on simplified variants of actor-critic, such as double-loop updates with i.i.d. sampling, which are often impractical for real-world applications. We consider the canonical and widely adopted single-timescale updates with Markovian sampling in continuous state-action space. Specifically, we establish finite-time convergence by introducing a novel Lyapunov analysis framework, which provides a unified convergence characterization of both the actor and the critic. Our approach is less conservative than previous methods and offers new insights into the coupled dynamics of actor-critic updates.

## 1. Introduction

Actor-critic methods have achieved substantial success in many challenging applications (Mnih et al., 2016; Silver et al., 2017; Vinyals et al., 2019; Lazaridis et al., 2020). In particular, it becomes instrumental in enabling highly robust and agile robot motion control involving continuous state-action spaces, such as quadruped locomotion control (Hoeller et al., 2024), humanoid whole-body control (Radosavovic et al., 2024), drone racing (Kaufmann et al., 2023), etc.

Despite substantial empirical success, the theoretical analysis of actor-critic is significantly behind. Most prior theoretical studies of actor-critic methods consider only finite state-action spaces and focus on their impractical variants

to simplify the analysis, including the double-loop updates and the two-timescale updates. The double-loop updates perform multiple critic updates for a fixed actor (Yang et al., 2019; Kumar et al., 2023; Agarwal et al., 2021; Xu et al., 2020b). This facilitates more accurate value function estimation, which in turn enables a more precise policy gradient estimation for the fixed actor. It allows a simple decoupled analysis of the actor and the critic. However, such an implementation is impractical due to the high sampling complexity. Another variant is the two-timescale actor-critic method (Wu et al., 2020; Xu et al., 2020c; Chen et al., 2023; Shen et al., 2023; Hong et al., 2023), which assigns a smaller step size for the actor than that of the critic, with their ratio converging to zero as the number of iterations approaches infinity (i.e., $\lim_{t \to \infty} \alpha_t/\beta_t = 0$). It allows an asymptotically decoupling of the actor and the critic in the convergence analysis, similar to performing multiple critic updates at a fixed actor. However, such artificial slowing down of the critic update is often not desired in practice.

The canonical and more practical implementation of actor-critic is the single-timescale update, where the actor and the critic are updated simultaneously with proportional step sizes at each iteration (i.e., $\alpha_t/\beta_t = c$). However, analyzing its convergence is significantly more challenging than for the aforementioned simplified variants, as the actor and critic updates are strongly coupled. The aforementioned decoupled analysis is over-conservative and cannot establish the convergence of the single-timescale actor-critic. Recent efforts to study the convergence of the single-timescale actor-critic algorithm include Chen et al. (2021), Olshevsky & Gharesifard (2023), and Chen & Zhao (2024). However, these works are limited to finite action spaces with i.i.d. sampling and do not extend to the more practical yet complex setting of **Markovian sampling** in **continuous state-action spaces** under the **single-timescale update scheme** (See the comparison in Table 1). In particular, Chen et al. (2021) and Olshevsky & Gharesifard (2023) assume i.i.d. sampling directly from the *stationary distribution* for the critic and from the *discounted state visitation distribution* for the actor. However, both of these distributions are unknown for real-time online learning and hence are impractical. Additionally, Chen & Zhao (2024) considers the simpler *undiscounted time-average reward* setting rather than the widely adopted *discounted reward* setting. The key difference is

[1]Department of Electrical and Computer Engineering, National University of Singapore, Singapore. Correspondence to: Lin Zhao <elezhli@nus.edu.sg>.

*Proceedings of the 42nd International Conference on Machine Learning*, Vancouver, Canada. PMLR 267, 2025. Copyright 2025 by the author(s).

*Table 1.* Comparison of existing works on single-timescale actor-critic methods in discounted reward setting with linear function approximation. Our work is the first to address the continuous state-action spaces and Markovian sampling.

| References | State Space | Action Space | Sampling for Critic | Sampling for Actor | Complexity |
|---|---|---|---|---|---|
| Chen et al. (2021) | Infinite | Finite | i.i.d. from stationary distribution | i.i.d. from state visitation distribution | $\mathcal{O}(\epsilon^{-2})$ |
| Olshevsky & Gharesifard (2023) | Finite | Finite | i.i.d. from stationary distribution | i.i.d. from state visitation distribution | $\mathcal{O}(\epsilon^{-2})$ |
| This paper | Infinite | Infinite | Markovian | Markovian | $\tilde{\mathcal{O}}(\epsilon^{-2})$ |

that, in the former, the policy gradient only requires stationary distribution, whereas in the latter, it depends on the visitation distribution. How to accurately approximate the visitation distribution in the single-timescale update scheme with Markovian sampling remains an open question. To tackle the aforementioned challenges, specifically,

1. We introduce a new operator-based analysis to handle the intricacies arising from the uncountable continuous space. In particular, it enables us to generalize many important bounds to the continuous space successfully (see Appendix B).

2. For the Markovian samples used to update the actor, we prove that the resulted state distribution converges to the *discounted state visitation distribution* (see Proposition 3.2). We further utilize it to accurately estimate the policy gradient in the analysis.

3. We propose a Lyapunov-based convergence analysis framework, where a novel Lyapunov function is constructed specifically for the single-timescale actor-critic algorithm. We also establish a variety of new properties (see, for example, Proposition 3.1, Proposition 4.4), which enable us to demonstrate finite-time convergence for both the actor and the critic simultaneously, with a less conservative analysis.

Moreover, we highlight that our analysis builds on the same set of common assumptions that are widely adopted in many literature (Wu et al., 2020; Chen et al., 2021; Olshevsky & Gharesifard, 2023; Chen & Zhao, 2024). In particular, Assumptions 4.1 and 4.2 are about the regularity of the problem of interest, and Assumption 4.3 can be easily satisfied by many common policy parametrizations. Overall, our work takes a significant step toward a more practical analysis of actor-critic.

### 1.1. Related Work

In this section, we review the existing works on actor-critic methods.

**Actor-Critic methods.** The actor-critic algorithm, initially proposed by (Konda & Tsitsiklis, 1999), was later extended to the natural actor-critic variant by (Kakade, 2001). The asymptotic convergence of actor-critic algorithms has been well established under various settings, as demonstrated in works by Kakade (2001), Bhatnagar et al. (2009), and Zhang et al. (2020b). More recently, many studies have focused on the finite-time convergence of actor-critic methods. They primarily focus on two variants for the ease of analysis: (1) double loop update, and (2) two-timescale update. For the double-loop variants, Kumar et al. (2023) investigated the finite-time local convergence of several actor-critic variants with linear function approximation. Wang et al. (2019) explored the global convergence of actor-critic methods with both the actor and the critic parameterized by neural networks with single hidden layers. Moreover, Gaur et al. (2024) established the last iterate convergence for actor-critic with neural networks.

For the two-timescale variants, Wu et al. (2020) established finite-time local convergence in the undiscounted time-average reward setting. Xu et al. (2020c) analyzed both local and global convergence for two-timescale (natural) actor-critic under the discounted reward setting, respectively, with multiple samples used for critic updates. Shen et al. (2023) investigated finite-time convergence for asynchronous actor-critic, while Hong et al. (2023) introduced a two-timescale stochastic approximation algorithm for bilevel optimization and two-timescale actor-critic.

There are only a few works considering the canonical and most widely adopted single-timescale variant. Fu et al. (2020) explored the least-squares temporal difference (LSTD) update for the critic, achieving the optimal policy within the energy-based policy class for both linear function approximation and neural network approximation. Zhou & Lu (2023) and Chen et al. (2024) established the global convergence of actor-critic methods for solving linear quadratic regulator. Recently, Chen et al. (2021); Olshevsky & Gharesifard (2023); Chen & Zhao (2024) investigated single-timescale actor-critic methods in general Markov

Decision Processes (MDPs) with linear function approximation, aligning with the focus of this work. Specifically, Chen et al. (2021) and Olshevsky & Gharesifard (2023) addressed the commonly used discounted reward setting, while Chen & Zhao (2024) improved upon (Wu et al., 2020) by advancing from the two-timescale to the single-timescale approach under the undiscounted time-average reward setting. A detailed review and comparison of these results can be found in Table 1 and the introduction.

**Notation.** We use san-serif letters to denote scalars and use lower and upper case bold letters to denote vectors and matrices respectively. We also use $\|\boldsymbol{\omega}\|$ to denote the $\ell_2$-norm of a vector $\boldsymbol{\omega}$ and $\|\boldsymbol{A}\|$ to denote the spectral norm of a matrix $\boldsymbol{A}$. Without further specification, we write $x_n = \mathcal{O}(y_n)$ if there exists an absolute positive constant $C$ such that $x_n \leq Cy_n$, for two sequences $\{x_n\}$ and $\{y_n\}$. We use $\tilde{\mathcal{O}}(\cdot)$ to hide logarithm factors. The total variation distance of two probability measure $\mu$ and $\nu$ is defined by $d_{TV}(\mu, \nu) := 1/2 \int_{\mathcal{X}} |\mu(dx) - \nu(dx)|$.

## 2. Preliminaries

**Markov Decision Process.** In this paper, we consider a discrete-time Markov Decision Process (MDP) defined by a tuple $\mathcal{M} = \{\mathcal{S}, \mathcal{A}, P, r, \gamma\}$, where $\mathcal{S}$ is the state space and $\mathcal{A}$ is the action space. The spaces $\mathcal{S}$ and $\mathcal{A}$ are allowed to be either finite sets or real vector spaces, i.e., $\mathcal{S} \subset \mathbb{R}^{d_s}$ and $\mathcal{A} \subset \mathbb{R}^{d_a}$. The transition kernel is denoted by $P(s_{t+1} \mid s_t, a_t) \in \mathbb{R}_{\geq 0}$, the reward function is $r : \mathcal{S} \times \mathcal{A} \to [-\bar{r}, \bar{r}]$, and $\gamma \in (0, 1)$ is the discounted factor. We also assume that the initial state is sampled from a fixed initial distribution $\eta$.

A policy $\pi_{\boldsymbol{\theta}}$ parameterized by $\boldsymbol{\theta} \in \mathcal{X}_{\Theta}$ maps a given state to a probability distribution over the action space, i.e., $a_t \sim \pi_{\boldsymbol{\theta}}(\cdot \mid s_t)$. We denote the stationary distribution induced by the policy $\pi_{\boldsymbol{\theta}}$ and the transition kernel $P$ by $\mu_{\boldsymbol{\theta}}$. The value function of a state $s$ under a policy $\pi_{\boldsymbol{\theta}}$ is the expected cumulative return when starting in $s$ and following $\pi_{\boldsymbol{\theta}}$ thereafter. It is defined as

$$V_{\boldsymbol{\theta}}(s) = \mathbb{E}_{a_t \sim \pi_{\boldsymbol{\theta}}(\cdot \mid s_t)} \left[ \sum_{t=0}^{\infty} \gamma^t r(s_t, a_t) \,\Big|\, s_0 = s \right], \quad (1)$$

where the expectation takes over the randomness of the policy $\pi_{\boldsymbol{\theta}}$ and the transition function $P$. The corresponding action-value function is the expected cumulative return when starting from state $s$, taking action $a$, and following $\pi_{\boldsymbol{\theta}}$ thereafter, which is defined as

$$Q_{\boldsymbol{\theta}}(s, a) = \mathbb{E} \left[ \sum_{t=0}^{\infty} \gamma^t r(s_t, a_t) \,\Big|\, s_0 = s, a_0 = a \right], \quad (2)$$

where we simplified the expectation notation when there is no confusion. The reinforcement learning (RL) tasks typically aim to find a policy $\pi_{\boldsymbol{\theta}}$ that maximizes the following objective function:

$$J(\boldsymbol{\theta}) = \int_{\mathcal{S}} \eta(s) V_{\boldsymbol{\theta}}(s) \, ds, \quad (3)$$

where $\eta(s)$ is a fixed initial distribution.

We denote the density at state $s'$ after transitioning for one time step from state $s$ by $P_{\boldsymbol{\theta}}(s' \mid s)$, which is defined as

$$P_{\boldsymbol{\theta}}(s' \mid s) = \int_{\mathcal{A}} P(s' \mid s, a) \pi_{\boldsymbol{\theta}}(a \mid s) \, da.$$

The corresponding state density after transitioning for $t$ time steps can be acquired by recursively applying $P_{\boldsymbol{\theta}}(s' \mid s)$, i.e.,

$$P_{\boldsymbol{\theta}}^t(s' \mid s) = \int_{\mathcal{S}} P_{\boldsymbol{\theta}}(s' \mid x) P_{\boldsymbol{\theta}}^{t-1}(x \mid s) \, dx, \, t > 1.$$

Consequently, we define the *discounted state visitation distribution* under policy $\pi_{\boldsymbol{\theta}}$ as

$$\nu_{\boldsymbol{\theta}}(s') = (1 - \gamma) \int_{\mathcal{S}} \sum_{t=0}^{\infty} \gamma^t \eta(s) P_{\boldsymbol{\theta}}^t(s' \mid s) \, ds. \quad (4)$$

It is worth noting that previous works (Chen et al., 2021; Olshevsky & Gharesifard, 2023) rely on sampling from this distribution, which is infeasible. In this work, we propose a practical sampling scheme to circumvent this impediment.

With the discounted state visitation distribution, the objective function can be reformulated as (Sutton et al., 1999)

$$J(\boldsymbol{\theta}) = \frac{1}{1 - \gamma} \int_{\mathcal{S}} \nu_{\boldsymbol{\theta}}(s) \int_{\mathcal{A}} \pi_{\boldsymbol{\theta}}(a \mid s) r(s, a) \, da ds$$
$$= \frac{1}{1 - \gamma} \mathbb{E}_{s \sim \nu_{\boldsymbol{\theta}}, a \sim \pi_{\boldsymbol{\theta}}} \Big[ r(s, a) \Big].$$

**Policy Gradient Theorem.** Policy gradient algorithms are among the most widely used approaches in continuous-action reinforcement learning. Their core concept involves adjusting the policy parameter $\boldsymbol{\theta}$ in the direction of the performance gradient $\nabla_{\boldsymbol{\theta}} J(\boldsymbol{\theta})$. These algorithms are built upon the foundational result known as the policy gradient theorem (Sutton et al., 1999):

$$\nabla_{\boldsymbol{\theta}} J(\boldsymbol{\theta}) = \frac{1}{1 - \gamma} \int_{\mathcal{S}} \nu_{\boldsymbol{\theta}}(s) \int_{\mathcal{A}} Q_{\boldsymbol{\theta}}(s, a) \nabla_{\boldsymbol{\theta}} \pi_{\boldsymbol{\theta}}(a \mid s) da ds$$
$$= \frac{1}{1 - \gamma} \mathbb{E}_{s \sim \nu_{\boldsymbol{\theta}}, a \sim \pi_{\boldsymbol{\theta}}} \Big[ Q_{\boldsymbol{\theta}}(s, a) \nabla_{\boldsymbol{\theta}} \log \pi_{\boldsymbol{\theta}}(a \mid s) \Big]. \quad (5)$$

Computing this gradient necessitates the Q-value associated with the current policy $\pi_{\boldsymbol{\theta}}$. The REINFORCE algorithm (Williams, 1992), an episodic Monte Carlo-based method, approximates the true Q-value by utilizing the cumulative rewards gathered along the sampled trajectory.

Note that for any function $b : \mathcal{S} \to \mathbb{R}$ that is independent of the action, we have

$$\int_{\mathcal{A}} b(s)\nabla \pi_{\boldsymbol{\theta}}(a|s) = b(s)\nabla \int_{\mathcal{A}} \pi_{\boldsymbol{\theta}}(a \mid s) = b(s)\nabla 1 = 0.$$

Therefore, the policy gradient theorem can be written equivalently as:

$$\nabla J(\boldsymbol{\theta}) = \frac{1}{1-\gamma}\mathbb{E}_{s,a}\big[(Q_{\boldsymbol{\theta}}(s,a) - b(s))\nabla_{\boldsymbol{\theta}} \log \pi_{\boldsymbol{\theta}}(a|s)\big],$$

where $b(s)$ is called the baseline function. A popular choice of baseline is the state-value function, which leads to the following advantage-based policy gradient

$$\nabla_{\boldsymbol{\theta}} J(\boldsymbol{\theta}) = \frac{1}{1-\gamma}\mathbb{E}_{s\sim\nu_{\boldsymbol{\theta}},a\sim\pi_{\boldsymbol{\theta}}}\big[G_{\boldsymbol{\theta}}(s,a)\nabla_{\boldsymbol{\theta}} \log \pi_{\boldsymbol{\theta}}(a|s)\big],$$

where $G_{\boldsymbol{\theta}}(s,a) = Q_{\boldsymbol{\theta}}(s,a) - V_{\boldsymbol{\theta}}(s)$ is known as the advantage function. This is the "REINFORCE with a baseline".

The baseline function can help reduce variance. However, like all Monte Carlo-based methods, it can still suffer from high variance and thus learns slowly. An alternative approach involves introducing an additional trainable model to approximate the value function, a method typically known as actor-critic methods.

## 3. Actor-Critic Methods

In this work, we analyze the classic single-sample single-timescale actor-critic method, where the critic employs bootstrapping by using a single sampled reward to update its value estimate at each iteration. We consider the following linear function approximation of the state-value function:

$$\widehat{V}_{\boldsymbol{\theta}}(s;\boldsymbol{\omega}) = \boldsymbol{\phi}(s)^{\top}\boldsymbol{\omega},$$

where $\boldsymbol{\phi}(\cdot) : \mathcal{S} \to \mathbb{R}^d$ is a known feature mapping, which satisfies $\|\boldsymbol{\phi}(\cdot)\| \leq 1$. To align $\widehat{V}_{\boldsymbol{\theta}}(s;\boldsymbol{\omega})$ with its true value $V_{\boldsymbol{\theta}}(s)$, the semi-gradient TD(0) update is employed to estimate the linear coefficient $\boldsymbol{\omega}$ (hereafter referred to as the critic):

$$\boldsymbol{\omega}_{t+1} = \boldsymbol{\omega}_t + \beta \left(r_t + \gamma\boldsymbol{\phi}(s_{t+1})^{\top}\boldsymbol{\omega}_t - \boldsymbol{\phi}(s_t)^{\top}\boldsymbol{\omega}_t\right)\boldsymbol{\phi}(s_t),$$

where $\beta$ is the step size of the critic $\boldsymbol{\omega}$ and $r_t := r(s_t, a_t)$. Denote the transition tuple as $O := (s, a, s')$ and we define the following temporal difference error

$$\delta(O,\boldsymbol{\omega}) = r(s,a) + \gamma\boldsymbol{\phi}(s')^{\top}\boldsymbol{\omega} - \boldsymbol{\phi}(s)^{\top}\boldsymbol{\omega},$$

and the update rule for the critic is then given by

$$\boldsymbol{\omega}_{t+1} = \boldsymbol{\omega}_t + \beta\delta(O_t,\boldsymbol{\omega}_t)\boldsymbol{\phi}(s_t), \tag{6}$$

where $O_t = (s_t, a_t, s_{t+1})$ denotes the $t$-th transition tuple for the critic, generated via Markovian sampling under the policy $\pi_{\boldsymbol{\theta}}$ and transition kernel $P$, such that

$$O_t = \big(s_t, a_t \sim \pi_{\boldsymbol{\theta}_t}(\cdot \mid s_t), s_{t+1} \sim P(\cdot \mid s_t, a_t)\big). \tag{7}$$

Since $\delta$ is an approximation of the advantage function, similar to REINFORCE with a baseline, the corresponding update rule for the actor can be written as:

$$\boldsymbol{\theta}_{t+1} = \boldsymbol{\theta}_t + \alpha\delta(\widehat{O}_t, \boldsymbol{\omega}_t)\nabla_{\boldsymbol{\theta}} \log \pi_{\boldsymbol{\theta}_t}(\hat{a}_t \mid \hat{s}_t), \tag{8}$$

where $\alpha$ is the step size of the actor and $\widehat{O}_t = (\hat{s}_t, \hat{a}_t, \hat{s}_{t+1})$ denotes the $t$-th transition tuple for the actor. Specifically, $\widehat{O}_t$ is also generated via the following Markovian sampling (Konda & Tsitsiklis, 2003; Shen et al., 2023)

$$\widehat{O}_t = \big(\hat{s}_t, \hat{a}_t \sim \pi_{\boldsymbol{\theta}_t}(\cdot \mid \hat{s}_t), \hat{s}_{t+1} \sim \widehat{P}(\cdot \mid \hat{s}_t, \hat{a}_t)\big),$$
$$\text{where} \quad \widehat{P} = \gamma P + (1-\gamma)\eta. \tag{9}$$

Here the transition kernel $\widehat{P}$ is defined as with probability $\gamma$, the next state follows the original transition kernel $P$; Otherwise, with probability $1 - \gamma$, the next state is sampled from the initial distribution $\eta$. Note that the above Markovian sampling generally requires a simulator whose state can be arbitrarily reset. It has a few nice properties that will be discussed shortly, which facilitate an accurate estimation of the policy gradient.

Denote the class of real-valued functions on the state space $\mathcal{S}$ by $\mathcal{F} := \{f \mid f : \mathcal{S} \to \mathbb{R}\}$. We define the operator $\mathcal{P} : \mathcal{F} \to \mathcal{F}$ acts on a state distribution $f \in \mathcal{F}$ by

$$(\mathcal{P}f)(s') = \int_{\mathcal{S}} \int_{\mathcal{A}} f(s)\pi_{\boldsymbol{\theta}}(a \mid s)P(s' \mid s, a) \, da \, ds. \tag{10}$$

We further define a reset operator $\mathcal{E} : \mathcal{F} \to \mathcal{F}$ such that it reset any state distribution $f$ to the initial distribution $\eta$:

$$(\mathcal{E}f)(s) = \eta(s).$$

Therefore, the operator $\widehat{\mathcal{P}}$ that acts on a state distribution, describing how the distribution evolves after a single step of the Markov chain induced by the policy $\pi_{\boldsymbol{\theta}}$ and the transition kernel $\widehat{P}$, can be written compactly as:

$$\widehat{\mathcal{P}} = \gamma\mathcal{P} + (1-\gamma)\mathcal{E}.$$

We show in the following proposition that the discounted state visitation distribution $\nu_{\boldsymbol{\theta}}$ defined in Eq. (4) is the stationary distribution of the Markov chain induced by policy $\pi_{\boldsymbol{\theta}}$ and transition kernel $\widehat{P}$ by showing that $\nu_{\boldsymbol{\theta}}$ is the unique fixed point of the operator $\widehat{\mathcal{P}}$.

**Proposition 3.1.** $\nu_{\boldsymbol{\theta}}(s)$ *is the unique fixed point of the operator* $\widehat{\mathcal{P}}$, *that is,*

$$(\widehat{\mathcal{P}}\nu_{\boldsymbol{\theta}})(s) = \nu_{\boldsymbol{\theta}}(s),$$

**Algorithm 1** Continuous Single-sample Single-timescale Actor-Critic with Markovian Sampling

1: **Initialize:** actor parameter $\boldsymbol{\theta}_0$, critic parameter $\boldsymbol{\omega}_0$, initial states $s_0, \hat{s}_0 \sim \eta$, stepsizes $\alpha$ for actor, $\beta$ for critic.
2: **for** $t = 0, 1, 2, \cdots, T-1$ **do**
3:     **Markovian sampling:**
4:         $O_t = \big(s_t, a_t \sim \pi_{\boldsymbol{\theta}_t}(\cdot \,|\, s_t), s_{t+1} \sim P(\cdot \,|\, s_t, a_t)\big)$,
5:         $\widehat{O}_t = \big(\hat{s}_t, \hat{a}_t \sim \pi_{\boldsymbol{\theta}_t}(\cdot \,|\, \hat{s}_t), \hat{s}_{t+1} \sim \widehat{P}(\cdot \,|\, \hat{s}_t, \hat{a}_t)\big)$.
6:     **Critic and actor update:**
7:         $\omega_{t+1} = proj_{\bar{\omega}}\big(\omega_t + \beta\delta(O_t, \omega_t)\phi(s_t)\big)$,
8:         $\boldsymbol{\theta}_{t+1} = \boldsymbol{\theta}_t + \alpha\delta(\widehat{O}_t, \boldsymbol{\omega}_t)\nabla_{\boldsymbol{\theta}} \log \pi_{\boldsymbol{\theta}_t}(\hat{a}_t \,|\, \hat{s}_t)$.
9: **end for**

*and therefore the stationary distribution of the following Markov chain induced by $\pi_{\boldsymbol{\theta}}$ and $\widehat{P}$,*

$$\hat{s}_0 \xrightarrow{(\pi_{\boldsymbol{\theta}}, \widehat{P})} \hat{s}_1 \xrightarrow{(\pi_{\boldsymbol{\theta}}, \widehat{P})} \cdots \xrightarrow{(\pi_{\boldsymbol{\theta}}, \widehat{P})} \hat{s}_t \xrightarrow{(\pi_{\boldsymbol{\theta}}, \widehat{P})} \hat{s}_{t+1}. \quad (11)$$

The above proposition justifies the actor's sampling scheme in Eq. (9), as $\widehat{O}_t$ effectively approximates the discounted state visitation distribution, which is required by the policy gradient theorem (Eq. (5)) following the actor update formula in Eq. (8).

**Proposition 3.2.** *For the Markov chain defined in Eq. (11), we have*

$$d_{TV}\big(\mathbb{P}(\hat{s}_t \in \cdot | \hat{s}_0 = s), \nu_{\boldsymbol{\theta}}(\cdot)\big) \leq \gamma^t, \; \forall t \geq 0, \forall s \in \mathcal{S}.$$

Proposition 3.2 states that the distribution of $\hat{s}_t$ converges to $\nu_{\boldsymbol{\theta}}$ geometrically with rate $\gamma$, a crucial property for managing the Markovian noise arising from the actor's Markovian sampling in Eq. (9).

We summarize the above-described actor-critic algorithm in Algorithm 1. The "continuous" refers to the general setting of continuous state-action spaces. "single-timescale" refers to the fact that the stepsizes $\alpha$ and $\beta$ are kept in constant proportion. In addition, the terminology "single-sample" follows Olshevsky & Gharesifard (2023), which refers to the fact that at each iteration, the critic and the actor are each updated using a single sample. Note that Olshevsky & Gharesifard (2023), who consider the discounted reward setting, assume access to samples from the discounted state visitation distribution and the stationary distribution for updating the actor and critic, respectively. This assumption requires a simulator capable of resetting to arbitrary states. In the simpler *time-average reward* setting (Wu et al., 2020; Chen & Zhao, 2024), the policy gradient depends solely on the stationary distribution, allowing the actor to utilize the same samples as the critic. In contrast, discounted reward setting requires the policy gradient to be computed with respect to the visitation distribution, as shown in Eq. (5),

which is more challenging. To this end, the Markovian sampling strategy introduced in Eq. (9) becomes necessary to track this distribution. Consequently, Algorithm 1 inherently supports online learning and applies to continuous control tasks.

As shown in Line 4 and Line 5 of Algorithm 1, we adopt Markovian sampling for both the critic and the actor. Specifically, the transition tuple for the critic is generated by the following Markov chain

$$s_0 \xrightarrow{(\pi_{\boldsymbol{\theta}_0}, P)} s_1 \xrightarrow{(\pi_{\boldsymbol{\theta}_1}, P)} \cdots \xrightarrow{(\pi_{\boldsymbol{\theta}_{t-1}}, P)} s_t \xrightarrow{(\pi_{\boldsymbol{\theta}_t}, P)} s_{t+1}. \quad (12)$$

while the actor's transition tuple is generated by the Markov chain

$$\hat{s}_0 \xrightarrow{(\pi_{\boldsymbol{\theta}_0}, \widehat{P})} \hat{s}_1 \xrightarrow{(\pi_{\boldsymbol{\theta}_1}, \widehat{P})} \cdots \xrightarrow{(\pi_{\boldsymbol{\theta}_{t-1}}, \widehat{P})} \hat{s}_t \xrightarrow{(\pi_{\boldsymbol{\theta}_t}, \widehat{P})} \hat{s}_{t+1}. \quad (13)$$

Note that the above Markov chains (time-inhomogeneous) differ from the one defined in Eq. (11) (time-homogeneous), as they involve a varying policy $\pi_{\boldsymbol{\theta}_t}$. This poses a major challenge for analyzing Algorithm 1, since a single sample is insufficient to accurately approximate the stationary distribution of the state under a fixed policy. Previous studies simplified their analysis by assuming i.i.d. samples drawn from the stationary distribution for the critic and from the visitation distribution for the actor. However, such sampling is infeasible in practice because both of them are unknown. In contrast, our approach is more practical since samples can be drawn directly from the Markov chain.

In Algorithm 1 Line 7, a projection ($proj(\cdot)$) is introduced to keep the critic norm-bounded by $\bar{\omega}$, which is widely adopted in the literature (Wu et al., 2020; Chen et al., 2021; Olshevsky & Gharesifard, 2023; Chen & Zhao, 2024) for analysis. Note that the projection can be handled easily, relaxed using its non-expansive property in our analysis.

## 4. Assumptions

Before presenting the main results, we will discuss several standard assumptions that are common in the literature of analyzing actor-critic with linear function approximation (Wu et al., 2020; Xu et al., 2020b; Shen et al., 2023; Chen et al., 2021; Olshevsky & Gharesifard, 2023; Chen & Zhao, 2024).

By taking the expectation of $\boldsymbol{\omega}_{t+1}$ in Eq. (6) with respect to the stationary distribution, and conditioning on $\boldsymbol{\omega}_t$, we have

$$\mathbb{E}_{\boldsymbol{\theta}}[\boldsymbol{\omega}_{t+1} \,|\, \boldsymbol{\omega}_t] = \boldsymbol{\omega}_t + \beta(\boldsymbol{b}_{\boldsymbol{\theta}} - \boldsymbol{A}_{\boldsymbol{\theta}}\boldsymbol{\omega}_t),$$

where

$$\boldsymbol{A}_{\boldsymbol{\theta}} := \mathbb{E}_{(s,a,s')}\big[\phi(s)\big(\phi(s) - \gamma\phi(s')\big)^{\top}\big], \quad (14)$$
$$\boldsymbol{b}_{\boldsymbol{\theta}} := \mathbb{E}_{(s,a)}\big[r(s,a)\phi(s)\big], \quad (15)$$

and $s \sim \mu_{\boldsymbol{\theta}}(\cdot), a \sim \pi_{\boldsymbol{\theta}}(\cdot \,|\, s), s' \sim P(\cdot \,|\, s, a)$ is the subsequent state of the $(s, a)$. It can be easily shown that (Sutton & Barto, 2018) the TD limiting point $\boldsymbol{\omega}^*(\boldsymbol{\theta})$ satisfies:

$$\boldsymbol{A}_{\boldsymbol{\theta}} \boldsymbol{\omega}^*(\boldsymbol{\theta}) = \boldsymbol{b}_{\boldsymbol{\theta}}. \tag{16}$$

**Assumption 4.1.** For any $\boldsymbol{\theta}$, the matrix $\boldsymbol{A}_{\boldsymbol{\theta}}$ defined in Eq. (14) is positive definite and its minimal eigenvalue can be lower bounded by $\lambda$.

Assumption 4.1 is commonly adopted in analyzing actor-critic (TD learning) with linear function approximation (Bhandari et al., 2018; Wu et al., 2020; Qiu et al., 2021; Chen et al., 2021; Olshevsky & Gharesifard, 2023; Chen & Zhao, 2024). It is explained as exploration since $\boldsymbol{A}_{\boldsymbol{\theta}}$ can be rank deficient without sufficient exploration (Olshevsky & Gharesifard, 2023; Chen & Zhao, 2024). Assumption 4.1 further guarantees the problem's solvability since with this assumption, we have $\boldsymbol{\omega}^*(\boldsymbol{\theta}) = \boldsymbol{A}_{\boldsymbol{\theta}}^{-1} \boldsymbol{b}_{\boldsymbol{\theta}}$. In addition, we can choose $\bar{\omega} = \bar{r}\lambda^{-1}$ so that all $\boldsymbol{\omega}^*$ lie within the projection radius $\bar{\omega}$ because $\|\boldsymbol{b}_{\boldsymbol{\theta}}\| \leq \bar{r}$ and $\|\boldsymbol{A}_{\boldsymbol{\theta}}^{-1}\| \leq \lambda^{-1}$, which justifies the projection operator introduced in Line 7 of Algorithm 1.

**Assumption 4.2** (Uniform ergodicity)**.** For any $\boldsymbol{\theta}$, denote $\mu_{\boldsymbol{\theta}}(\cdot)$ as the stationary distribution induced by the policy $\pi_{\boldsymbol{\theta}}(\cdot \,|\, s)$ and the transition kernel $P(\cdot \,|\, s, a)$. For the following Markov chain (augmented with action) generated by the policy $\pi_{\boldsymbol{\theta}}$ and transition kernel $P$, i.e.,

$$s_0 \xrightarrow{(\pi_{\boldsymbol{\theta}}, P)} s_1 \xrightarrow{(\pi_{\boldsymbol{\theta}}, P)} \cdots \xrightarrow{(\pi_{\boldsymbol{\theta}}, P)} s_t \xrightarrow{(\pi_{\boldsymbol{\theta}}, P)} s_{t+1}, \tag{17}$$

there exist $m > 0$ and $\rho \in (0, 1)$ such that

$$d_{TV}\big(\mathbb{P}(s_\tau \in \cdot \,|\, s_0 = s), \mu_{\boldsymbol{\theta}}(\cdot)\big) \leq m\rho^\tau, \forall \tau \geq 0, \forall s \in \mathcal{S}.$$

Assumption 4.2 assumes the Markov chain is geometrically mixing. It is commonly employed to characterize the noise induced by Markovian sampling in RL algorithms (Bhandari et al., 2018; Wu et al., 2020; Chen et al., 2021; Chen & Zhao, 2024). This is the counterpart of Proposition 3.2 (which is proved for analyzing the induced Markovian noise associated with the actor update). It is assumed since $P$ is a general transition kernel that lacks the $\gamma$-contraction property of the transition kernel $\widehat{P}$ established in Proposition 3.2.

To justify this assumption in the continuous space, we note that all the distributions specified by the Ornstein–Uhlenbeck (OU) process satisfy this property. The OU process converges to a Gaussian distribution with the exponential mixing time. Moreover, it can also be shown that this property holds for more general diffusion processes (Del Moral & Villemonais, 2018).

**Assumption 4.3** (Lipschitz continuity of policy)**.** Let $\pi_{\boldsymbol{\theta}}(a \,|\, s)$ be a policy parameterized by $\boldsymbol{\theta} \in \mathcal{X}_\Theta$ with bounded support. There exist positive constants $B, L_l$ and

$L$ such that for any $\boldsymbol{\theta}, \boldsymbol{\theta}_1, \boldsymbol{\theta}_2 \in \mathcal{X}_\Theta$, $s \in \mathcal{S}$, and $a \in \mathcal{A}$, it holds that:

(a) $\|\nabla \log \pi_{\boldsymbol{\theta}}(a \,|\, s)\| \leq B$,

(b) $\|\nabla \log \pi_{\boldsymbol{\theta}_1}(a \,|\, s) - \nabla \log \pi_{\boldsymbol{\theta}_2}(a \,|\, s)\| \leq L_l \|\boldsymbol{\theta}_1 - \boldsymbol{\theta}_2\|$,

(c) $|\pi_{\boldsymbol{\theta}_1}(a \,|\, s) - \pi_{\boldsymbol{\theta}_2}(a \,|\, s)| \leq L \|\boldsymbol{\theta}_1 - \boldsymbol{\theta}_2\|$.

Assumption 4.3 states the regularity of the policy which is standard in the literature of actor-critic methods (Xu et al., 2020a; Wu et al., 2020; Chen et al., 2021; Olshevsky & Gharesifard, 2023; Chen & Zhao, 2024). These conditions are sufficiently general to be satisfied by a wide range of distributions, including the uniform distribution, the truncated Gaussian distribution, and the Beta distribution with $\alpha, \beta > 1$.

With Assumption 4.3, we show in the following proposition that the policy $\pi_{\boldsymbol{\theta}}$ is Lipschitz continuous with respect to its parameter $\boldsymbol{\theta}$ in terms of the total variation distance.

**Proposition 4.4.** *There exists a positive constant $L_\pi$ such that for any $\boldsymbol{\theta}_1, \boldsymbol{\theta}_2 \in \mathcal{X}_\Theta$, it holds that*

$$d_{TV}(\pi_{\boldsymbol{\theta}_1}(\cdot \,|\, s), \pi_{\boldsymbol{\theta}_2}(\cdot \,|\, s)) \leq L_\pi \|\boldsymbol{\theta}_1 - \boldsymbol{\theta}_2\|. \tag{18}$$

We observed that Proposition 4.4 plays a key ingredient in the overall proof. With this proposition, we establish a bound on the distance between stationary distributions, as detailed in Lemma B.1 within Preliminary Lemmas in Appendix B, extending previous results from the tabular case to the continuous setting. This further facilitates the derivation of corresponding results for the discounted state visitation distribution, as presented in Lemma B.3.

## 5. Main Results

We define the following uniform upper bound for the linear function approximation error of the critic:

$$\epsilon_{\text{app}} := \sup_{\boldsymbol{\theta}} \sqrt{\mathbb{E}_{s \sim \nu_{\boldsymbol{\theta}}}(\boldsymbol{\phi}(s)^\top \boldsymbol{\omega}^*(\boldsymbol{\theta}) - V_{\boldsymbol{\theta}}(s))^2}. \tag{19}$$

The error $\epsilon_{\text{app}}$ is zero if $V_{\boldsymbol{\theta}}$ is indeed a linear function for any $\boldsymbol{\theta}$. Naturally, it can be expected that the learning errors of Algorithm 1 depend on $\epsilon_{\text{app}}$.

We define the following integer $\tau_{\text{mix}}$ that will be useful in the statement of the theorems:

$$\tau_{\text{mix}} := \min\left\{i \geq 0 \mid m\rho^{i-1} \leq \frac{1}{\sqrt{T}} \wedge \gamma^{i-1} \leq \frac{1}{\sqrt{T}}\right\},$$

where $m, \rho$ are constants defined in Assumption 4.2 and $\gamma$ is the discounted factor. Therefore, we choose

$$\tau_{\text{mix}} = \mathcal{O}(\log T)$$

such that $m\rho^{\tau_{\mathrm{mix}}-1} \leq 1/\sqrt{T}$ and $\gamma^{\tau_{\mathrm{mix}}-1} \leq 1/\sqrt{T}$. The integer $\tau_{\mathrm{mix}}$ represents the mixing time of the ergodic Markov chain defined in Eq. (11) and Eq. (17), which will be used to control the Markovian noise in the analysis.

We define $\Delta_t = \boldsymbol{\omega}_t - \boldsymbol{\omega}_t^*$ with $\boldsymbol{\omega}_t^* = \boldsymbol{\omega}^*(\boldsymbol{\theta}_t)$ to measure the critic error while $\nabla J(\boldsymbol{\theta}_t)$ serves as a measure of the actor error since for a general non-convex problem, our objective is to demonstrate that $\nabla J(\boldsymbol{\theta}_t)$ converges to zero.

**Theorem 5.1.** *Consider Algorithm 1 with $\alpha = c/\sqrt{T}, \beta = 1/\sqrt{T}$, where $c$ is a constant depending on problem parameters. Suppose Assumptions 4.1-4.3 hold, we have for $T \geq 2\tau_{\mathrm{mix}}$,*

$$\frac{1}{T - \tau_{\mathrm{mix}}} \sum_{t=\tau_{\mathrm{mix}}}^{T-1} \mathbb{E}\|\Delta_t\|^2 = \mathcal{O}\left(\frac{\log^2 T}{\sqrt{T}}\right) + \mathcal{O}(\epsilon_{\mathrm{app}}),$$

$$\frac{1}{T - \tau_{\mathrm{mix}}} \sum_{t=\tau_{\mathrm{mix}}}^{T-1} \mathbb{E}\|\nabla J(\boldsymbol{\theta}_t)\|^2 = \mathcal{O}\left(\frac{\log^2 T}{\sqrt{T}}\right) + \mathcal{O}(\epsilon_{\mathrm{app}}).$$

Theorem 5.1 establishes the finite-time convergence of Algorithm 1. If the critic approximation error $\epsilon_{\mathrm{app}}$ is zero, we see that both the critic error and the actor error diminish at a sub-linear rate of $\widetilde{\mathcal{O}}(T^{-1/2})$. The additional logarithmic term $(\log^2 T)$ is incurred by the mixing time of the Markov chain, which can be eliminated under i.i.d. sampling as will be shown in Proof Sketch. In terms of sample complexity, to obtain an $\epsilon$-approximate stationary point, it takes a number of $\widetilde{\mathcal{O}}(\epsilon^{-2})$ samples, which is typically the sample complexity of single-timescale actor-critic (Chen et al., 2021; Olshevsky & Gharesifard, 2023; Chen & Zhao, 2024).

The vanilla version of Algorithm 1 is introduced in the classic textbook (Sutton & Barto, 2018) as a canonical actor-critic algorithm with linear function approximation. As a canonical algorithm, its convergence has been a focal point of research, extensively studied across diverse settings, e.g., two-timescale (Wu et al., 2020), single-timescale (Chen et al., 2021; Olshevsky & Gharesifard, 2023; Chen & Zhao, 2024), time-average reward setting (Wu et al., 2020; Chen & Zhao, 2024), discounted setting (Chen et al., 2021; Olshevsky & Gharesifard, 2023). Notably, among all the aforementioned studies, this work is the first to address the important yet challenging setting of continuous state and action spaces. Among the widely used discounted single-timescale approaches considered, our method is the first to employ Markovian sampling for both the critic and the actor in contrast to the artificial i.i.d. sampling (Chen et al., 2021; Olshevsky & Gharesifard, 2023). Therefore, our work compares favorably by closing two significant gaps left by prior studies.

## 5.1. Proof Sketch

To better illustrate our technical contribution, we provide a proof sketch to elucidate the significance of each error term and offer insights into the methods used to address them.

The key difference between single-timescale and two-timescale (double-loop) actor-critic lies in the strong coupling of the critic and actor errors. Unlike the two-timescale approach, which sequentially analyzes the convergence of critic error and actor error, the single-timescale setting requires simultaneous treatment of both errors. To address this, we propose a novel Lyapunov analysis framework and outline the proof of Theorem 5.1 in three steps. Step 1 derives an implicit upper bound for the critic error, treating it as an intermediate result. Step 2 performs a similar implicit analysis for the actor error. Step 3 combines these results into a novel Lyapunov function, whose convergence implies the simultaneous convergence of the critic and actor.

**Step 1: An implicit bound for critic error**. Using the critic update rule, we decompose the squared critic error by (see Eq. (27))

$$\mathbb{E}\|\Delta_{t+1}\|^2 \leq \mathbb{E}\|\Delta_t\|^2 + \underbrace{2\beta^2 \mathbb{E}\|\boldsymbol{f}(O_t, \boldsymbol{\omega}_t)\|^2}_{I_1}$$
$$+ \underbrace{2\mathbb{E}\|\boldsymbol{\omega}_t^* - \boldsymbol{\omega}_{t+1}^*\|^2}_{I_2} + \underbrace{2\beta\mathbb{E}\langle\Delta_t, \bar{\boldsymbol{f}}(\boldsymbol{\omega}_t, \boldsymbol{\theta}_t)\rangle}_{I_3}$$
$$+ \underbrace{2\beta\mathbb{E}\langle\Delta_t, \boldsymbol{f}(O_t, \boldsymbol{\omega}_t) - \bar{\boldsymbol{f}}(\boldsymbol{\omega}_t, \boldsymbol{\theta}_t)\rangle}_{I_4}$$
$$+ \underbrace{2\mathbb{E}\langle\Delta_t, \boldsymbol{\omega}_t^* - \boldsymbol{\omega}_{t+1}^*\rangle}_{I_5},$$

where $\boldsymbol{f}(O, \boldsymbol{\omega}) = \delta(O, \boldsymbol{\omega})\phi(s)$ is the update term of the critic and $\bar{\boldsymbol{f}}$ is its mean value defined in Eq. (20).

$I_1$ reflects the variance of the critic update which can be bounded by $\mathcal{O}(1/\sqrt{T})$ due to its bounded update.

$I_2$ represents the difference between the moving critic target $\boldsymbol{\omega}_t^*$, which can be controlled due to its Lipschitz continuity shown in Lemma B.6.

$I_3$ is the inner product between the critic error $\Delta_t$ and its mean-path update $\bar{\boldsymbol{f}}$. It can be bounded by $-2\lambda\beta\mathbb{E}\|\Delta_t\|^2$ under Assumption 4.1 since $\boldsymbol{\omega}^*$ is the solution of Eq. (16). Note that this bound combined with first term $\mathbb{E}\|\Delta_t\|^2$ is $(1 - 2\lambda\beta)\mathbb{E}\|\Delta_t\|^2$ which implies a contraction of the critic error because the coefficient is less than 1.

$I_4$ represents the Markovian noise term, capturing the deviation between the critic's actual update $\boldsymbol{f}$ and its mean-path $\bar{\boldsymbol{f}}$. To analyze this deviation, we aim to show that the sample $O_t$ is close to its stationary distribution, as the error term $I_3$ vanishes when $O_t$ is drawn from the stationary distri-

bution. First, we demonstrate that the sample $O_t$ from the original Markov chain defined in Eq. (12) is close to the sample from the auxiliary Markov chain in Eq. (21), as their differences are limited to the last $\tau$ steps. The total variation distance between these samples is controlled by the actor's change, i.e., $\|\boldsymbol{\theta}_t - \boldsymbol{\theta}_{t-\tau}\|$, as established in Lemma B.2. By choosing $\tau = \tau_{\mathrm{mix}} = \mathcal{O}(\log T)$ and noting that the actor's update speed is $\mathcal{O}(1/\sqrt{T})$, the distance between $\boldsymbol{\theta}_t$ and $\boldsymbol{\theta}_{t-\tau}$ is bounded by $\mathcal{O}(\tau_{\mathrm{mix}}/\sqrt{T})$. Consequently, the accumulated deviation over the last $\tau_{\mathrm{mix}}$ steps is bounded by $\mathcal{O}(\tau_{\mathrm{mix}}^2/\sqrt{T}) = \mathcal{O}(\log^2 T/\sqrt{T})$, explaining the logarithmic term in the convergence rate. Next, we show that the Markov noise of the sample from the auxiliary Markov chain approaches its stationary distribution after $\tau_{\mathrm{mix}}$ steps under a fixed policy, leveraging the uniform ergodicity assumption in Assumption 4.2. This highlights the role of Assumption 4.2 in analyzing single-sample single/two-timescale algorithms with Markovian sampling. The complete analysis of this Markovian noise is shown in Lemma C.1.

$I_5$ tracks both the critic error $\Delta_t$ and the difference between the drifting critic targets $\boldsymbol{\omega}_t^*$. It can be bounded by the critic error $\Delta_t$ and the actor error $\nabla J(\boldsymbol{\theta}_t)$ after error decomposition. In contrast, the two-timescale setting can prove that $I_4$ converges to zero. To see why this is the case, note that from the Lipschitz continuity of the critic target $\boldsymbol{\omega}_t^*$ shown in Lemma B.6, error term $\boldsymbol{\omega}_t^* - \boldsymbol{\omega}_{t+1}^*$ can be bounded by the update of the actor $\boldsymbol{\theta}$, i.e., $\boldsymbol{\theta}_t - \boldsymbol{\theta}_{t+1}$. Since the update step size for the actor is $\alpha$ while the contraction of the critic error is at a rate $1 - 2\lambda\beta$, a ratio term $\alpha/\beta$ appears by moving the term $-2\lambda\beta\mathbb{E}\|\Delta_t\|^2$ to the left side of the above inequality and dividing its coefficient. Therefore, one can leave other terms in $I_4$ as constant and bound it by $\mathcal{O}(\alpha/\beta)$. Since $\lim_{t\to\infty} \alpha_t/\beta_t = 0$ in two-timescale approach, thereby directly establish the convergence of the critic. However, $\lim_{t\to\infty} \alpha_t/\beta_t = c$ in single-timescale approach which is why we can only bound $I_4$ by $\Delta_t$ and $\nabla J(\boldsymbol{\theta})$ and make an implicit upper bound for the critic error. The final bound is summarized in Theorem D.1.

**Step 2: An implicit bound for actor error**. Using the actor update rule and the smoothness property of $J(\boldsymbol{\theta})$ (Lemma B.8), we decompose the squared actor error by (see Eq. (31))

$$
\begin{aligned}
(1-\gamma)\mathbb{E}\|\nabla J(\boldsymbol{\theta}_t)\|^2 &\leq \frac{1}{\alpha}\big(\mathbb{E}\big[J(\boldsymbol{\theta}_{t+1}) - J(\boldsymbol{\theta}_t)\big]\big) \\
&+ \underbrace{\frac{\alpha L_g}{2}\mathbb{E}\|\boldsymbol{h}(\widehat{O}_t, \boldsymbol{\omega}_t, \boldsymbol{\theta}_t)\|^2}_{I_1} - \underbrace{\mathbb{E}\langle\nabla J(\boldsymbol{\theta}_t), \bar{\boldsymbol{g}}(\boldsymbol{\omega}_t^*, \boldsymbol{\theta}_t)\rangle}_{I_2} \\
&+ \underbrace{\mathbb{E}\langle\nabla J(\boldsymbol{\theta}_t), \bar{\boldsymbol{h}}(\boldsymbol{\omega}_t, \boldsymbol{\theta}_t) - \boldsymbol{h}(\widehat{O}_t, \boldsymbol{\omega}_t, \boldsymbol{\theta}_t)\rangle}_{I_3} \\
&+ \underbrace{\mathbb{E}\langle\nabla J(\boldsymbol{\theta}_t), \bar{\boldsymbol{h}}(\boldsymbol{\omega}_t^*, \boldsymbol{\theta}_t) - \bar{\boldsymbol{h}}(\boldsymbol{\omega}_t, \boldsymbol{\theta}_t)\rangle}_{I_4},
\end{aligned}
$$

where $h(O, \boldsymbol{\omega}, \boldsymbol{\theta}) = \delta(O, \boldsymbol{\omega})\nabla\log\pi_{\boldsymbol{\theta}}(a\,|\,s)$ is the update term of the actor, $\bar{\boldsymbol{h}}$ is its mean value, and $\bar{\boldsymbol{g}}(\boldsymbol{\omega}_t^*, \boldsymbol{\theta}_t)$ defined in Eq. (20) represents the approximation error of the optimal critic $\boldsymbol{\omega}_t^*$. The first term on the right-hand side of the above inequality compares the actor's performances between consecutive updates, which can be eliminated by telescoping.

$I_1$ reflects the variance of the actor update which can be controlled by $\mathcal{O}(1/\sqrt{T})$ due to its bounded update.

$I_2$ is the inner product between actor error and the approximation error of the optimal critic $\boldsymbol{\omega}_t^*$. This term is control by the approximation error $\mathcal{O}(\epsilon_{\mathrm{app}})$ defined in Eq. (19).

$I_3$ represents the Markovian noise term, capturing the deviation between the actor's actual update $\boldsymbol{h}$ and its mean-path $\bar{\boldsymbol{h}}$. Similar to the critic analysis, this noise is controlled by showing that the original Markov chain defined in Eq. (13) is close to the auxiliary Markov chain in Eq. (22). Additionally, samples from the auxiliary Markov chain approach their stationary distribution after $\tau_{\mathrm{mix}}$ steps, leveraging the uniform ergodicity property established in Proposition 3.2. This error term is bounded in Lemma C.3.

$I_4$ tracks the inner product between the actor error $\nabla J(\boldsymbol{\theta})$ and the critic error $(\Delta_t = \boldsymbol{\omega} - \boldsymbol{\omega}_t^*)$. In two-timescale actor-critic, this term goes to zero due to the convergence of the critic. However, in single-timescale approach, we can only bound this term by $\nabla J(\boldsymbol{\theta}_t)$ and $\Delta_t$ which will be treated together later. This give an implicit upper bound for the actor error. The final result of the above inequality is summarized in Theorem D.2.

**Step 3: A novel Lyapunov analysis**. From Step 1 and Step 2, we get two inequalities about the coupled critic error and actor error. Here we bring them together by defining the following Lyapunov function

$$
\mathbb{L}_t = \frac{2B}{1-\gamma}\mathbb{E}\|\Delta_t\|^2 + \frac{1-\gamma}{2B}\mathbb{E}\|\nabla J(\boldsymbol{\theta}_t)\|^2,
$$

where $2B/(1-\gamma)$ is the scaling coefficient which balances the contribution of the critic error and the actor error. Combine the results in Step 1 and Step 2 (Eq. (26) and Eq. (30)) gives an unified inequality of $\mathbb{L}_t$. We then define the total error as $\mathcal{L} := 1/(T - \tau_{\mathrm{mix}})\sum_{t=\tau_{\mathrm{mix}}}^{T-1}\mathbb{L}_t$. Telescoping from $t = T - \tau_{\mathrm{mix}}$ to $T - 1$, it can be shown that (see Eq. (33))

$$
\mathcal{L} \leq \left(\frac{2L_c Bc}{\lambda} + \frac{1}{2}\right)\mathcal{L} + \mathcal{O}\left(\frac{\log^2 T}{\sqrt{T}}\right) + \mathcal{O}(\epsilon_{\mathrm{app}}),
$$

where $c = \alpha/\beta$ is the stepsize ratio between the actor and the critic, $\gamma$ is the discounted factor, $\lambda$ is the maximum eigenvalue of $\boldsymbol{A}_{\boldsymbol{\theta}}$ defined in Assumption 4.1, and $L_c$ is the Lipschitz constant characterized in Lemma B.6. Therefore,

choosing $c < \lambda/4BL_c$ (see Eq. (34)), we have

$$\mathcal{L} = \mathcal{O}\left(\frac{\log^2 T}{\sqrt{T}}\right) + \mathcal{O}(\epsilon_{\text{app}}),$$

which implies the convergence of the critic error and the actor error simultaneously. Therefore, we finish the proof of Theorem 5.1.

## 6. Conclusion

In this paper, we provide a finite-time convergence analysis for the single-sample, single-timescale actor-critic algorithm in continuous state-action spaces. We propose a novel Lyapunov analysis framework, which allows a less conservative analysis under the same set of assumptions adopted in existing studies. Our analysis offers new insights into the coupled dynamics of actor-critic updates. Unlike prior works that assume artificial decoupling between the actor and critic, our results capture the interdependencies that arise naturally in practical implementations. Moreover, our framework and analytical techniques can serve as a foundation for studying other single-timescale reinforcement learning algorithms in continuous domains.

## Acknowledgements

This work was supported by the Singapore Ministry of Education Tier 1 Academic Research Fund (A-8001174-00-00) and Tier 2 Academic Research Funds (T2EP20123-0037, T2EP20224-0035).

## Impact Statement

This paper presents work whose goal is to advance the field of Machine Learning. There are no potential societal consequences of our work.

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

# Supplementary Material

## Table of Contents

## A. Notation

In the following, we will analyze the convergence of the above algorithm. We define the following notations:

$$
\begin{aligned}
\boldsymbol{f}(O,\boldsymbol{\omega}) &:= \big(r(s,a) + \gamma\boldsymbol{\phi}(s')^\top\boldsymbol{\omega} - \boldsymbol{\phi}(s)^\top\boldsymbol{\omega}\big)\boldsymbol{\phi}(s) \\
\bar{\boldsymbol{f}}(\boldsymbol{\omega},\boldsymbol{\theta}) &:= \mathbb{E}_{O\sim(\mu_{\boldsymbol{\theta}},\pi_{\boldsymbol{\theta}},P)}\big[f(O,\boldsymbol{\omega})\big], \\
\boldsymbol{h}(O,\boldsymbol{\omega},\boldsymbol{\theta}) &:= \big(r(s,a) + \gamma\boldsymbol{\phi}(s')^\top\boldsymbol{\omega} - \boldsymbol{\phi}(s)^\top\boldsymbol{\omega}\big)\nabla\log\pi_{\boldsymbol{\theta}}(a\,|\,s) \\
\bar{\boldsymbol{h}}(\boldsymbol{\omega},\boldsymbol{\theta}) &:= \mathbb{E}_{O\sim(\nu_{\boldsymbol{\theta}},\pi_{\boldsymbol{\theta}},P)}\big[h(O,\boldsymbol{\omega},\boldsymbol{\theta})\big], \\
\boldsymbol{g}(O,\boldsymbol{\omega},\boldsymbol{\theta}) &:= \big((\gamma\boldsymbol{\phi}(s') - \boldsymbol{\phi}(s))^\top\boldsymbol{\omega} - (\gamma V_{\boldsymbol{\theta}}(s') - V_{\boldsymbol{\theta}}(s))\big)\nabla\log\pi_{\boldsymbol{\theta}}(a\,|\,s), \\
\bar{\boldsymbol{g}}(\boldsymbol{\omega},\boldsymbol{\theta}) &:= \mathbb{E}_{O\sim(\nu_{\boldsymbol{\theta}},\pi_{\boldsymbol{\theta}},P)}\big[g(O,\boldsymbol{\omega},\boldsymbol{\theta})\big].
\end{aligned}
\tag{20}
$$

We make use of the following auxiliary Markov chain to deal with the Markovian noise.

**Auxiliary Markov Chain for the Critic:**

$$
s_{t-\tau} \xrightarrow{\boldsymbol{\theta}_{t-\tau}} a_{t-\tau} \xrightarrow{P} s_{t-\tau+1} \xrightarrow{\boldsymbol{\theta}_{t-\tau}} \tilde{a}_{t-\tau+1} \xrightarrow{P} \tilde{s}_{t-\tau+2} \xrightarrow{\boldsymbol{\theta}_{t-\tau}} \tilde{a}_{t-\tau+2} \cdots \xrightarrow{P} \tilde{s}_t \xrightarrow{\boldsymbol{\theta}_{t-\tau}} \tilde{a}_t \xrightarrow{P} \tilde{s}_{t+1}.
\tag{21}
$$

**Auxiliary Markov Chain for the Actor:**

$$
\hat{s}_{t-\tau} \xrightarrow{\boldsymbol{\theta}_{t-\tau}} \hat{a}_{t-\tau} \xrightarrow{\widehat{P}} \hat{s}_{t-\tau+1} \xrightarrow{\boldsymbol{\theta}_{t-\tau}} \bar{a}_{t-\tau+1} \xrightarrow{\widehat{P}} \bar{s}_{t-\tau+2} \xrightarrow{\boldsymbol{\theta}_{t-\tau}} \bar{a}_{t-\tau+2} \cdots \xrightarrow{\widehat{P}} \bar{s}_t \xrightarrow{\boldsymbol{\theta}_{t-\tau}} \bar{a}_t \xrightarrow{\widehat{P}} \bar{s}_{t+1}.
\tag{22}
$$

In the sequel, we denote by $\widetilde{O}_t := (\tilde{s}_t, \tilde{a}_t, \tilde{s}_{t+1})$ the tuple generated from the auxiliary Markov chain in Eq. (21) and $\bar{O}_t := (\bar{s}_t, \bar{a}_t, \bar{s}_{t+1})$ the tuple generated from the auxiliary Markov chain in Eq. (22). In comparison, $O_t := (s_t, a_t, s_{t+1})$ and $\widehat{O}_t := (\hat{s}_t, \hat{a}_t, \hat{s}_{t+1})$ denotes the tuple generated by Algorithm 1. We use $O'$ as a shorthand for an independent sample

from stationary distribution $s \sim \mu_{\boldsymbol{\theta}}, a \sim \pi_{\boldsymbol{\theta}}, s' \sim P$ and use $O''$ as a shorthand for an independent sample from discounted state visitation distribution $s \sim \nu_{\boldsymbol{\theta}}, a \sim \pi_{\boldsymbol{\theta}}, s' \sim P$.

Throughout the proof, we define $\delta_t := \delta(O_t, \boldsymbol{\omega}_t)$, and $\bar{\delta} = \bar{r} + 2\bar{\omega}$ is the uniform upper bound for $\delta$. We also define a filtration $\mathcal{F}_t = \sigma(s_0, \widehat{s}_0, a_0, \widehat{a}_0, s_1, \widehat{s}_1, a_1, \widehat{a}_1, \cdots, s_t, \widehat{s}_t)$, where $\sigma(\cdot)$ denotes the $\sigma$-algebra generated by the random variables.

## B. Preliminary Lemmas

In this section, we present several preliminary lemmas, encompassing three aspects: extending previous work to continuous settings (Lemma B.1, Lemma B.2, Lemma B.5, Lemma B.6), establishing the corresponding statistical properties for actor samples (Lemma B.3, Lemma B.4), and stating previously established results (Lemma B.7, Lemma B.8, Lemma B.9).

**Lemma B.1.** *For any $\boldsymbol{\theta}_1$ and $\boldsymbol{\theta}_2$, it holds that*

$$d_{TV}(\mu_{\boldsymbol{\theta}_1}, \mu_{\boldsymbol{\theta}_2}) \leq 2L_\pi \left( \lceil \log_\rho m^{-1} \rceil + \frac{1}{1-\rho} \right) \|\boldsymbol{\theta}_1 - \boldsymbol{\theta}_2\|,$$

$$d_{TV}(\mu_{\boldsymbol{\theta}_1} \otimes \pi_{\boldsymbol{\theta}_1}, \mu_{\boldsymbol{\theta}_2} \otimes \pi_{\boldsymbol{\theta}_2}) \leq 2L_\pi \left( 1 + \lceil \log_\rho m^{-1} \rceil + \frac{1}{1-\rho} \right) \|\boldsymbol{\theta}_1 - \boldsymbol{\theta}_2\|,$$

$$d_{TV}(\mu_{\boldsymbol{\theta}_1} \otimes \pi_{\boldsymbol{\theta}_1} \otimes P, \mu_{\boldsymbol{\theta}_2} \otimes \pi_{\boldsymbol{\theta}_2} \otimes P) \leq 2L_\pi \left( 1 + \lceil \log_\rho m^{-1} \rceil + \frac{1}{1-\rho} \right) \|\boldsymbol{\theta}_1 - \boldsymbol{\theta}_2\|.$$

**Lemma B.2.** *Given time indexes $t$ and $\tau$ such that $t \geq \tau > 0$, consider the auxiliary Markov chain in Eq. (21). Conditioning on $\mathcal{F}_{t-\tau}$, we have*

$$d_{TV}\big(\mathbb{P}(s_{t+1} \in \cdot), \mathbb{P}(\widetilde{s}_{t+1} \in \cdot)\big) \leq d_{TV}\big(\mathbb{P}(O_t \in \cdot), \mathbb{P}(\widetilde{O}_t \in \cdot)\big),$$

$$d_{TV}\big(\mathbb{P}(O_t \in \cdot), \mathbb{P}(\widetilde{O}_t \in \cdot)\big) = d_{TV}\big(\mathbb{P}((s_t, a_t) \in \cdot), \mathbb{P}((\widetilde{s}_t, \widetilde{a}_t) \in \cdot)\big),$$

$$d_{TV}\big(\mathbb{P}((s_t, a_t) \in \cdot), \mathbb{P}((\widetilde{s}_t, \widetilde{a}_t) \in \cdot)\big) \leq d_{TV}\big(\mathbb{P}(s_t \in \cdot), \mathbb{P}(\widetilde{s}_t \in \cdot)\big) + L_\pi \mathbb{E}\big[\|\boldsymbol{\theta}_t - \boldsymbol{\theta}_{t-\tau}\|\big].$$

**Lemma B.3.** *For any $\boldsymbol{\theta}_1$ and $\boldsymbol{\theta}_2$, it holds that*

$$d_{TV}(\nu_{\boldsymbol{\theta}_1}, \nu_{\boldsymbol{\theta}_2}) \leq \frac{2L_\pi}{1-\gamma} \|\boldsymbol{\theta}_1 - \boldsymbol{\theta}_2\|,$$

$$d_{TV}(\nu_{\boldsymbol{\theta}_1} \otimes \pi_{\boldsymbol{\theta}_1}, \nu_{\boldsymbol{\theta}_2} \otimes \pi_{\boldsymbol{\theta}_2}) \leq 2L_\pi \left( 1 + \frac{1}{1-\gamma} \right) \|\boldsymbol{\theta}_1 - \boldsymbol{\theta}_2\|,$$

$$d_{TV}(\nu_{\boldsymbol{\theta}_1} \otimes \pi_{\boldsymbol{\theta}_1} \otimes P, \nu_{\boldsymbol{\theta}_2} \otimes \pi_{\boldsymbol{\theta}_2} \otimes P) \leq 2L_\pi \left( 1 + \frac{1}{1-\gamma} \right) \|\boldsymbol{\theta}_1 - \boldsymbol{\theta}_2\|.$$

**Lemma B.4.** *Given time indexes $t$ and $\tau$ such that $t \geq \tau > 0$, consider the auxiliary Markov chain in Eq. (22). Conditioning on $\mathcal{F}_{t-\tau}$, we have*

$$d_{TV}\big(\mathbb{P}(\hat{s}_{t+1} \in \cdot), \mathbb{P}(\bar{s}_{t+1} \in \cdot)\big) \leq d_{TV}\big(\mathbb{P}(\widehat{O}_t \in \cdot), \mathbb{P}(\bar{O}_t \in \cdot)\big),$$

$$d_{TV}\big(\mathbb{P}(\widehat{O}_t \in \cdot), \mathbb{P}(\bar{O}_t \in \cdot)\big) = d_{TV}\big(\mathbb{P}((\hat{s}_t, \hat{a}_t) \in \cdot), \mathbb{P}((\bar{s}_t, \bar{a}_t) \in \cdot)\big),$$

$$d_{TV}\big(\mathbb{P}((\hat{s}_t, \hat{a}_t) \in \cdot), \mathbb{P}((\bar{s}_t, \bar{a}_t) \in \cdot)\big) \leq d_{TV}\big(\mathbb{P}(\hat{s}_t \in \cdot), \mathbb{P}(\bar{s}_t \in \cdot)\big) + L_\pi \mathbb{E}\big[\|\boldsymbol{\theta}_t - \boldsymbol{\theta}_{t-\tau}\|\big].$$

**Lemma B.5.** *For any $\boldsymbol{\theta}_1, \boldsymbol{\theta}_2$, we have*

$$|J(\boldsymbol{\theta}_1) - J(\boldsymbol{\theta}_2)| \leq L_J \|\boldsymbol{\theta}_1 - \boldsymbol{\theta}_2\|,$$

*where $L_J = 4\bar{r} L_\pi \big(1 + (1-\gamma)^{-1}\big)$.*

**Lemma B.6.** *There exists a constant $L_c > 0$ such that*

$$\|\boldsymbol{\omega}^*(\boldsymbol{\theta}_1) - \boldsymbol{\omega}^*(\boldsymbol{\theta}_2)\| \leq L_c \|\boldsymbol{\theta}_1 - \boldsymbol{\theta}_2\|, \forall \boldsymbol{\theta}_1, \boldsymbol{\theta}_2 \in \mathbb{R}^d,$$

*where $L_c = (8\lambda^{-2}\bar{r} + 4\lambda^{-1}\bar{r})L_\pi \big(1 + \lceil \log_\rho m^{-1} \rceil + 1/(1-\rho)\big)$.*

**Lemma B.7.** *For any $\boldsymbol{\theta}, \boldsymbol{\theta}' \in \mathbb{R}^d$, there exists constant $L_\mu$ such that $\|\nabla\mu_{\boldsymbol{\theta}} - \nabla\mu_{\boldsymbol{\theta}'}\| \leq L_\mu\|\boldsymbol{\theta} - \boldsymbol{\theta}'\|$, where $\mu_{\boldsymbol{\theta}}(s)$ is the stationary distribution under the policy $\pi_{\boldsymbol{\theta}}$.*

**Lemma B.8** ((Zhang et al., 2020a), Lemma 3.2). *For the performance function $J(\boldsymbol{\theta})$, there exists a constant $L_g > 0$ such that for all $\boldsymbol{\theta}_1, \boldsymbol{\theta}_2 \in \mathbb{R}^d$, it holds that*

$$\|\nabla J(\boldsymbol{\theta}_1) - \nabla J(\boldsymbol{\theta}_2)\| \leq L_g\|\boldsymbol{\theta}_1 - \boldsymbol{\theta}_2\|, \tag{23}$$

*which further implies*

$$J(\boldsymbol{\theta}_2) \geq J(\boldsymbol{\theta}_1) + \langle\nabla J(\boldsymbol{\theta}_1), \boldsymbol{\theta}_2 - \boldsymbol{\theta}_1\rangle - \frac{L_g}{2}\|\boldsymbol{\theta}_1 - \boldsymbol{\theta}_2\|^2. \tag{24}$$

**Lemma B.9** ((Chen et al., 2021), Proposition 8). *For any $\boldsymbol{\theta}_1, \boldsymbol{\theta}_2 \in \mathbb{R}^d$, we have*

$$\|\nabla\boldsymbol{\omega}^*(\boldsymbol{\theta}_1) - \nabla\boldsymbol{\omega}^*(\boldsymbol{\theta}_2)\| \leq L_s\|\boldsymbol{\theta}_1 - \boldsymbol{\theta}_2\|,$$

*where $L_s$ is a positive constant.*

## C. Markovian Noise

We then the following Markovian noise term

$$
\begin{aligned}
\Lambda(O, \boldsymbol{\omega}, \boldsymbol{\theta}) =& \langle\boldsymbol{\omega} - \boldsymbol{\omega}^*, \boldsymbol{f}(O, \boldsymbol{\omega}) - \bar{\boldsymbol{f}}(\boldsymbol{\omega}, \boldsymbol{\theta})\rangle, \\
\Gamma(O, \boldsymbol{\omega}, \boldsymbol{\theta}) =& \langle\boldsymbol{\omega} - \boldsymbol{\omega}^*, (\nabla\boldsymbol{\omega}^*)^\top(\bar{\boldsymbol{h}}(\boldsymbol{\omega}^*, \boldsymbol{\theta}) - \boldsymbol{h}(O, \boldsymbol{\omega}^*, \boldsymbol{\theta}))\rangle, \\
\Xi(O, \boldsymbol{\omega}, \boldsymbol{\theta}) =& \langle\nabla J(\boldsymbol{\theta}), \bar{\boldsymbol{h}}(\boldsymbol{\omega}, \boldsymbol{\theta}) - \boldsymbol{h}(O, \boldsymbol{\omega}, \boldsymbol{\theta})\rangle.
\end{aligned}
\tag{25}
$$

**Lemma C.1.** *For any $t \geq \tau_{\mathrm{mix}}$, the Markovian noise in the critic update, denoted by $\Lambda(O_t, \boldsymbol{\omega}_t, \boldsymbol{\theta}_t)$, satisfies*

$$\mathbb{E}\big[\Lambda(O_t, \boldsymbol{\omega}_t, \boldsymbol{\theta}_t)\big] \leq M_1\frac{1}{\sqrt{T}},$$

*where $M_1 = \big(8\bar{\omega}\bar{\delta}L_\pi(1 + \lceil\log_\rho m^{-1}\rceil + (1-\rho)^{-1}) + 2\bar{\delta}L_c\big)\bar{\delta}B\tau_{\mathrm{mix}}c + (8\bar{\omega} + 2\bar{\delta})\bar{\delta}\tau_{\mathrm{mix}} + 4\bar{\omega}L_\pi B\bar{\delta}^2\tau_{\mathrm{mix}}^2 c + 4\bar{\omega}\bar{\delta}$.*

**Lemma C.2.** *For any $t \geq \tau_{\mathrm{mix}} > 0$, it holds that*

$$\mathbb{E}\big[\Gamma(\widehat{O}_t, \boldsymbol{\omega}_t, \boldsymbol{\theta}_t)\big] \leq M_2\frac{1}{\sqrt{T}},$$

*where*

$$
\begin{aligned}
M_2 =& (2\bar{\delta}BL_c^2 + 4\bar{\delta}\bar{\omega}BL_s + 4\bar{\omega}L_cL_{\bar{h}})\tau_{\mathrm{mix}}\bar{\delta}Bc + 2\bar{\delta}BL_c\tau_{\mathrm{mix}}\bar{\delta} + 4\bar{\omega}\bar{\delta}BL_cL_\pi\tau_{\mathrm{mix}}^2\bar{\delta}Bc + 4\bar{\omega}\bar{\delta}BL_c, \\
L_{\bar{h}} =& \bar{\delta}L_l + 2BL_c + 4\bar{\delta}BL_\pi\big(1 + \frac{1}{1-\gamma}\big).
\end{aligned}
$$

**Lemma C.3.** *For any $t \geq \tau_{\mathrm{mix}} > 0$, it can be shown that*

$$\mathbb{E}\big[\Xi(\widehat{O}_t, \boldsymbol{\omega}_t, \boldsymbol{\theta}_t)\big] \leq M_3\frac{1}{\sqrt{T}},$$

*where*

$$
\begin{aligned}
M_3 =& (2\bar{\delta}BL_g + 2L_JL_{\bar{h}})\bar{\delta}Bc\tau_{\mathrm{mix}} + 4BL_J\bar{\delta}\tau_{\mathrm{mix}} + 2\bar{\delta}^2B^2L_JL_\pi c\tau_{\mathrm{mix}}^2 + 2\bar{\delta}BL_J, \\
L_{\bar{h}} =& \bar{\delta}L_l + 2BL_c + 4\bar{\delta}BL_\pi\big(1 + \frac{1}{1-\gamma}\big).
\end{aligned}
$$

# D. Proof of Main Theorem

## D.1. An implicit bound for critic error

**Theorem D.1.** *Choose $\alpha_t = c/\sqrt{T}, \beta_t = 1/\sqrt{T}$, for any $\tau_{\mathrm{mix}} \leq t < T$, we have*

$$\mathbb{E}\|\Delta_t\|^2 \leq \frac{1}{\lambda\beta}(\mathbb{E}\|\Delta_t\|^2 - \mathbb{E}\|\Delta_{t+1}\|^2) + \frac{2c(1-\gamma)}{\lambda}L_c\mathbb{E}\|\Delta_t\|\|\nabla J(\boldsymbol{\theta}_t)\| + \mathcal{O}(\frac{\log^2 T}{\sqrt{T}}) + \mathcal{O}(\epsilon_{\mathrm{app}}). \tag{26}$$

*Proof.* From the update rule of the critic in Line 6 of Algorithm 1, we have

$$\begin{aligned}
\|\boldsymbol{\omega}_{t+1} - \boldsymbol{\omega}_{t+1}^*\| &= \|proj_{\bar{\omega}}(\boldsymbol{\omega}_t + \beta\delta_t\boldsymbol{\phi}(s_t)) - \boldsymbol{\omega}_{t+1}^*\| \\
&= \|proj_{\bar{\omega}}(\boldsymbol{\omega}_t + \beta\delta_t\boldsymbol{\phi}(s_t)) - proj_{\bar{\omega}}(\boldsymbol{\omega}_{t+1}^*)\| \\
&\overset{(1)}{\leq} \|\boldsymbol{\omega}_t + \beta\delta_t\boldsymbol{\phi}(s_t) - \boldsymbol{\omega}_{t+1}^*\| \\
&= \|\boldsymbol{\omega}_t - \boldsymbol{\omega}_t^* + \beta\delta_t\boldsymbol{\phi}(s_t) + \boldsymbol{\omega}_t^* - \boldsymbol{\omega}_{t+1}^*\|,
\end{aligned}$$

where (1) holds because the projection function $proj_{\bar{\omega}}(\cdot)$ is 1-Lipschitz continuous. It follows that

$$\begin{aligned}
\|\Delta_{t+1}\|^2 =& \|\Delta_t + \beta\delta_t\boldsymbol{\phi}(s_t) + \boldsymbol{\omega}_t^* - \boldsymbol{\omega}_{t+1}^*\|^2 \\
=& \|\Delta_t\|^2 + \|\beta\delta_t\boldsymbol{\phi}(s_t) + \boldsymbol{\omega}_t^* - \boldsymbol{\omega}_{t+1}^*\|^2 \\
& + 2\langle\Delta_t, \beta\delta_t\boldsymbol{\phi}(s_t)\rangle + 2\langle\Delta_t, \boldsymbol{\omega}_t^* - \boldsymbol{\omega}_{t+1}^*\rangle \\
=& \|\Delta_t\|^2 + \|\beta\boldsymbol{f}(O_t, \boldsymbol{\omega}_t) + \boldsymbol{\omega}_t^* - \boldsymbol{\omega}_{t+1}^*\|^2 \\
& + 2\beta\langle\Delta_t, \boldsymbol{f}(O_t, \boldsymbol{\omega}_t) - \bar{\boldsymbol{f}}(\boldsymbol{\omega}_t, \boldsymbol{\theta}_t)\rangle \\
& + 2\beta\langle\Delta_t, \bar{\boldsymbol{f}}(\boldsymbol{\omega}_t, \boldsymbol{\theta}_t)\rangle + 2\langle\Delta_t, \boldsymbol{\omega}_t^* - \boldsymbol{\omega}_{t+1}^*\rangle \\
\leq& \|\Delta_t\|^2 + 2\beta^2\|\boldsymbol{f}(O_t, \boldsymbol{\omega}_t)\|^2 + 2\|\boldsymbol{\omega}_t^* - \boldsymbol{\omega}_{t+1}^*\|^2 \\
& + 2\beta\langle\Delta_t, \boldsymbol{f}(O_t, \boldsymbol{\omega}_t) - \bar{\boldsymbol{f}}(\boldsymbol{\omega}_t, \boldsymbol{\theta}_t)\rangle \\
& + 2\beta\langle\Delta_t, \bar{\boldsymbol{f}}(\boldsymbol{\omega}_t, \boldsymbol{\theta}_t)\rangle + 2\langle\Delta_t, \boldsymbol{\omega}_t^* - \boldsymbol{\omega}_{t+1}^*\rangle,
\end{aligned}$$

where $\boldsymbol{f}$ and $\bar{\boldsymbol{f}}$ are defined in Eq. (20).

Taking expectation up to $s_{t+1}$, we have

$$\mathbb{E}\|\Delta_{t+1}\|^2 \leq \mathbb{E}\|\Delta_t\|^2 + \underbrace{2\beta^2\mathbb{E}\|\boldsymbol{f}(O_t, \boldsymbol{\omega}_t)\|^2}_{I_1} + \underbrace{2\mathbb{E}\|\boldsymbol{\omega}_t^* - \boldsymbol{\omega}_{t+1}^*\|^2}_{I_2} + \underbrace{2\beta\mathbb{E}\langle\Delta_t, \bar{\boldsymbol{f}}(\boldsymbol{\omega}_t, \boldsymbol{\theta}_t)\rangle}_{I_3} \\
+ \underbrace{2\beta\mathbb{E}\langle\Delta_t, \boldsymbol{f}(O_t, \boldsymbol{\omega}_t) - \bar{\boldsymbol{f}}(\boldsymbol{\omega}_t, \boldsymbol{\theta}_t)\rangle}_{I_4} + \underbrace{2\mathbb{E}\langle\Delta_t, \boldsymbol{\omega}_t^* - \boldsymbol{\omega}_{t+1}^*\rangle}_{I_5}. \tag{27}$$

In the sequel, we will tackle $I_1, I_2, I_3, I_4, I_5$ respectively.

For term $I_1$, since $\|\boldsymbol{f}(O_t, \boldsymbol{\omega}_t)\| \leq \bar{\delta}$, we have

$$I_1 = 2\beta^2\mathbb{E}\|\boldsymbol{f}(O_t, \boldsymbol{\omega}_t)\|^2 \leq 2\beta^2\bar{\delta}^2.$$

For term $I_2$, from Lemma B.6, it can be shown that

$$I_2 = 2\mathbb{E}\|\boldsymbol{\omega}_t^* - \boldsymbol{\omega}_{t+1}^*\|^2 \leq 2L_c^2\mathbb{E}\|\boldsymbol{\theta}_1 - \boldsymbol{\theta}_2\|^2 \leq 2\alpha^2\bar{\delta}^2 B^2 L_c^2.$$

For term $I_3$, we have

$$\begin{aligned}
\langle\Delta_t, \bar{\boldsymbol{f}}(\boldsymbol{\omega}_t, \boldsymbol{\theta}_t)\rangle =& \langle\Delta_t, \bar{\boldsymbol{f}}(\boldsymbol{\omega}_t, \boldsymbol{\theta}_t) - \bar{\boldsymbol{f}}(\boldsymbol{\omega}_t^*, \boldsymbol{\theta}_t)\rangle \\
=& \langle\Delta_t, \mathbb{E}[(\gamma\boldsymbol{\phi}(s') - \boldsymbol{\phi}(s))^\top(\boldsymbol{\omega}_t - \boldsymbol{\omega}_t^*)\boldsymbol{\phi}(s)]\rangle \\
=& \Delta_t^\top\mathbb{E}[\boldsymbol{\phi}(s)(\gamma\boldsymbol{\phi}(s') - \boldsymbol{\phi}(s)]\Delta_t \\
=& -\Delta_t^\top\boldsymbol{A_\theta}\Delta_t \\
\leq& -\lambda\|\Delta_t\|^2.
\end{aligned}$$

It follows that

$$I_3 \leq -2\lambda\beta\mathbb{E}\|\Delta_t\|^2.$$

For term $I_4$, according to Lemma C.1, it holds that

$$I_4 = 2\beta\mathbb{E}\big[\Lambda(O_t, \boldsymbol{\omega}_t, \boldsymbol{\theta}_t)\big] \leq 2\beta M_1 \frac{1}{\sqrt{T}}.$$

For term $I_5$, we will instead give an implicit upper bound. It can be shown that

$$
\begin{aligned}
\mathbb{E}\langle\Delta_t, \boldsymbol{\omega}_t^* - \boldsymbol{\omega}_{t+1}^*\rangle &= \mathbb{E}\langle\Delta_t, \boldsymbol{\omega}_t^* - \boldsymbol{\omega}_{t+1}^* + (\nabla\boldsymbol{\omega}_t^*)^\top(\boldsymbol{\theta}_{t+1} - \boldsymbol{\theta}_t)\rangle + \mathbb{E}\langle\Delta_t, -(\nabla\boldsymbol{\omega}_t^*)^\top(\boldsymbol{\theta}_{t+1} - \boldsymbol{\theta}_t)\rangle \\
&\overset{(1)}{\leq} \frac{L_s}{2}\mathbb{E}\|\Delta_t\|\|\boldsymbol{\theta}_{t+1} - \boldsymbol{\theta}_t\|^2 + \alpha\mathbb{E}\langle\Delta_t, -(\nabla\boldsymbol{\omega}_t^*)^\top\boldsymbol{h}(\widehat{O}_t, \boldsymbol{\omega}_t, \boldsymbol{\theta}_t)\rangle \\
&\leq \alpha^2\bar{\delta}^2 B^2 L_s\bar{\omega} + \alpha\mathbb{E}\langle\Delta_t, -(\nabla\boldsymbol{\omega}_t^*)^\top\boldsymbol{h}(\widehat{O}_t, \boldsymbol{\omega}_t, \boldsymbol{\theta}_t)\rangle \\
&= \alpha^2\bar{\delta}^2 B^2 L_s\bar{\omega} + \alpha\mathbb{E}\langle\Delta_t, -(\nabla\boldsymbol{\omega}_t^*)^\top(\boldsymbol{h}(\widehat{O}_t, \boldsymbol{\omega}_t, \boldsymbol{\theta}_t) - \boldsymbol{h}(\widehat{O}_t, \boldsymbol{\omega}_t^*, \boldsymbol{\theta}_t))\rangle \\
&\quad + \alpha\mathbb{E}\langle\Delta_t, -(\nabla\boldsymbol{\omega}_t^*)^\top\boldsymbol{h}(\widehat{O}_t, \boldsymbol{\omega}_t^*, \boldsymbol{\theta}_t)\rangle \\
&\leq \alpha^2\bar{\delta}^2 B^2 L_s\bar{\omega} + 2\alpha B L_c\mathbb{E}\|\Delta_t\|^2 + \alpha\mathbb{E}\langle\Delta_t, -(\nabla\boldsymbol{\omega}_t^*)^\top\boldsymbol{h}(\widehat{O}_t, \boldsymbol{\omega}_t^*, \boldsymbol{\theta}_t)\rangle \\
&= \alpha^2\bar{\delta}^2 B^2 L_s\bar{\omega} + 2\alpha B L_c\mathbb{E}\|\Delta_t\|^2 + \alpha\underbrace{\mathbb{E}\langle\Delta_t, (\nabla\boldsymbol{\omega}_t^*)^\top(\bar{\boldsymbol{h}}(\boldsymbol{\omega}_t^*, \boldsymbol{\theta}_t) - \boldsymbol{h}(\widehat{O}_t, \boldsymbol{\omega}_t^*, \boldsymbol{\theta}_t))\rangle}_{J_1} \\
&\quad + \alpha\underbrace{\mathbb{E}\langle\Delta_t, (\nabla\boldsymbol{\omega}_t^*)^\top(\bar{\boldsymbol{g}}(\boldsymbol{\omega}_t^*, \boldsymbol{\theta}_t) - \bar{\boldsymbol{h}}(\boldsymbol{\omega}_t^*, \boldsymbol{\theta}_t))\rangle}_{J_2} + \alpha\underbrace{\mathbb{E}\langle\Delta_t, -(\nabla\boldsymbol{\omega}_t^*)^\top\bar{\boldsymbol{g}}(\boldsymbol{\omega}_t^*, \boldsymbol{\theta}_t)\rangle}_{J_3},
\end{aligned}
$$

where (1) follows from the smoothness of the optimal critic shown in Lemma B.9. We will analyze $J_1$, $J_2$, and $J_3$ individually.

For term $J_1$, from the Markovian noise analysis in Lemma C.2, we have

$$J_1 = \mathbb{E}\big[\Gamma(\widehat{O}_t, \boldsymbol{\omega}_t, \boldsymbol{\theta}_t)\big] \leq M_2\frac{1}{\sqrt{T}}.$$

For term $J_2$, from the policy gradient theorem in Eq. (5), we obtain

$$\bar{\boldsymbol{h}}(\boldsymbol{\omega}_t^*, \boldsymbol{\theta}_t) - \bar{\boldsymbol{g}}(\boldsymbol{\omega}_t^*, \boldsymbol{\theta}_t) = \mathbb{E}_{(s,a,s')\sim(\nu_{\boldsymbol{\theta}_t}, \pi_{\boldsymbol{\theta}_t}, P)}[(r(s,a) + \gamma V_{\boldsymbol{\theta}_t}(s') - V_{\boldsymbol{\theta}_t}(s))\nabla\log\pi_{\boldsymbol{\theta}_t}(a\,|\,s)] = (1-\gamma)\nabla J(\boldsymbol{\theta}_t). \quad (28)$$

It follows that

$$J_2 = \mathbb{E}\langle\Delta_t, -(\nabla\boldsymbol{\omega}_t^*)^\top(1-\gamma)\nabla J(\boldsymbol{\theta}_t)\rangle \leq (1-\gamma)L_c\mathbb{E}\|\Delta_t\|\|\nabla J(\boldsymbol{\theta}_t)\|.$$

For term $J_3$, we first show that

$$
\begin{aligned}
\bar{\boldsymbol{g}}(\boldsymbol{\omega}_t^*, \boldsymbol{\theta}_t) &\leq \sqrt{\mathbb{E}_{(s,a,s')\sim(\nu_{\boldsymbol{\theta}_t}, \pi_{\boldsymbol{\theta}_t}, P)}\|\boldsymbol{g}(O_t, \boldsymbol{\omega}_t^*, \boldsymbol{\theta}_t)\|^2} \\
&\leq \sqrt{\mathbb{E}\big[B^2((\gamma\boldsymbol{\phi}(s')^\top\boldsymbol{\omega}_t^* - \gamma V_{\boldsymbol{\theta}_t}(s')) - (\boldsymbol{\phi}(s)^\top\boldsymbol{\omega}_t^* - V_{\boldsymbol{\theta}}(s)))^2\big]} \\
&\leq \sqrt{\mathbb{E}\big[2B^2\big(\gamma^2(\boldsymbol{\phi}(s')^\top\boldsymbol{\omega}_t^* - V_{\boldsymbol{\theta}_t}(s'))^2 + (\boldsymbol{\phi}(s)^\top\boldsymbol{\omega}_t^* - V_{\boldsymbol{\theta}_t}(s))^2\big)\big]} \\
&\leq 2B\sqrt{\mathbb{E}[(\boldsymbol{\phi}(s)^\top\boldsymbol{\omega}_t^* - V_{\boldsymbol{\theta}_t}(s))^2]} \\
&= 2B\epsilon_{\text{app}}.
\end{aligned}
\quad (29)
$$

Then we have

$$J_3 \leq 4\bar{\omega}BL_c\epsilon_{\text{app}}.$$

Combining $J_1, J_2$, and $J_3$, we get

$$I_5 \leq 2\alpha^2 \bar{\delta}^2 B^2 L_s \bar{\omega} + 4\alpha B L_c \mathbb{E}\|\Delta_t\|^2 + 8\alpha\bar{\omega} B L_c \epsilon_{\mathrm{app}} + 2\alpha(1-\gamma)L_c \mathbb{E}\|\Delta_t\|\|\nabla J(\boldsymbol{\theta}_t)\| + 2\alpha M_2 \frac{1}{\sqrt{T}}.$$

Plugging $I_1, I_2, I_3, I_4$ and $I_5$ into Eq. (27), we obtain

$$\mathbb{E}\|\Delta_{t+1}\|^2 \leq \mathbb{E}\|\Delta_t\|^2 + 2\beta^2\bar{\delta}^2 + 2\alpha^2\bar{\delta}^2 B^2 L_c^2 + 2\beta M_1 \frac{1}{\sqrt{T}} - 2\lambda\beta\mathbb{E}\|\Delta_t\|^2 + 2\alpha^2\bar{\delta}^2 B^2 L_s\bar{\omega}$$

$$+ 4\alpha B L_c \mathbb{E}\|\Delta_t\|^2 + 8\alpha\bar{\omega}B L_c \epsilon_{\mathrm{app}} + 2\alpha(1-\gamma)L_c\mathbb{E}\|\Delta_t\|\|\nabla J(\boldsymbol{\theta}_t)\| + 2\alpha M_2\frac{1}{\sqrt{T}}$$

$$\overset{(1)}{\leq} (1-\lambda\beta)\mathbb{E}\|\Delta_t\|^2 + 2\alpha(1-\gamma)L_c\mathbb{E}\|\Delta_t\|\|\nabla J(\boldsymbol{\theta}_t)\|$$

$$+ (2\bar{\delta}^2 + 2c^2\bar{\delta}^2 B^2 L_c^2 + 2M_1 + 2c^2\bar{\delta}^2 B^2 L_s\bar{\omega} + 2cM_2)\frac{1}{T} + 8\alpha\bar{\omega}B L_c\epsilon_{\mathrm{app}},$$

where (1) holds as the step size ratio $c$ is chosen to satisfy $4\alpha B L_c \leq \lambda\beta$.

Rearranging the above inequality, we obtain

$$\mathbb{E}\|\Delta_t\|^2 \leq \frac{1}{\lambda\beta}(\mathbb{E}\|\Delta_t\|^2 - \mathbb{E}\|\Delta_{t+1}\|^2) + \frac{2c(1-\gamma)}{\lambda}L_c\mathbb{E}\|\Delta_t\|\|\nabla J(\boldsymbol{\theta}_t)\|$$

$$+ \lambda^{-1}(2\bar{\delta}^2 + 2c^2\bar{\delta}^2 B^2 L_c^2 + 2M_1 + 2c^2\bar{\delta}^2 B^2 L_s\bar{\omega} + 2cM_2)\frac{1}{\sqrt{T}} + 8c\bar{\omega}B L_c\epsilon_{\mathrm{app}}.$$

By leveraging the $\mathcal{O}(\cdot)$ notation, we can further summarise our implicit analysis for the critic as

$$\mathbb{E}\|\Delta_t\|^2 \leq \frac{1}{\lambda\beta}(\mathbb{E}\|\Delta_t\|^2 - \mathbb{E}\|\Delta_{t+1}\|^2) + \frac{2c(1-\gamma)}{\lambda}L_c\mathbb{E}\|\Delta_t\|\|\nabla J(\boldsymbol{\theta}_t)\| + \mathcal{O}(\frac{\log^2 T}{\sqrt{T}}) + \mathcal{O}(\epsilon_{\mathrm{app}}),$$

where the term $\log^2 T$ arises from the presence of $\tau_{\mathrm{mix}}^2$ in $M_1$ and $M_2$. Therefore, we finish the proof of Theorem D.1. $\square$

### D.2. An implicit bound for actor error

**Theorem D.2.** *Choose* $\alpha_t = c/\sqrt{T}, \beta_t = 1/\sqrt{T}$, *for any* $\tau_{\mathrm{mix}} \leq t < T$, *we have*

$$(1-\gamma)\mathbb{E}\|\nabla J(\boldsymbol{\theta}_t)\|^2 \leq \frac{1}{\alpha}(\mathbb{E}[J(\boldsymbol{\theta}_{t+1}) - J(\boldsymbol{\theta}_t)]) + 2B\mathbb{E}\|\nabla J(\boldsymbol{\theta}_t)\|\|\Delta_t\| + \mathcal{O}(\frac{\log^2 T}{\sqrt{T}}) + \mathcal{O}(\epsilon_{\mathrm{app}}). \qquad (30)$$

*Proof.* From the update rule of actor in Line 8 of Algorithm 1 and Lemma B.8, we have

$$J(\boldsymbol{\theta}_{t+1}) \geq J(\boldsymbol{\theta}_t) + \langle \nabla J(\boldsymbol{\theta}_t), \boldsymbol{\theta}_{t+1} - \boldsymbol{\theta}_t \rangle - \frac{L_g}{2}\|\boldsymbol{\theta}_t - \boldsymbol{\theta}_{t+1}\|^2$$

$$= J(\boldsymbol{\theta}_t) + \alpha\langle \nabla J(\boldsymbol{\theta}_t), \boldsymbol{h}(\widehat{O}_t, \boldsymbol{\omega}_t, \boldsymbol{\theta}_t)\rangle - \frac{L_g}{2}\alpha^2\|\boldsymbol{h}(\widehat{O}_t, \boldsymbol{\omega}_t, \boldsymbol{\theta}_t)\|^2$$

$$= J(\boldsymbol{\theta}_t) + \alpha\langle \nabla J(\boldsymbol{\theta}_t), \boldsymbol{h}(\widehat{O}_t, \boldsymbol{\omega}_t, \boldsymbol{\theta}_t) - \bar{\boldsymbol{h}}(\boldsymbol{\omega}_t, \boldsymbol{\theta}_t)\rangle + \alpha\langle \nabla J(\boldsymbol{\theta}_t), \bar{\boldsymbol{h}}(\boldsymbol{\omega}_t, \boldsymbol{\theta}_t)\rangle - \frac{L_g}{2}\alpha^2\|\boldsymbol{h}(\widehat{O}_t, \boldsymbol{\omega}_t, \boldsymbol{\theta}_t)\|^2$$

$$= J(\boldsymbol{\theta}_t) + \alpha\langle \nabla J(\boldsymbol{\theta}_t), \boldsymbol{h}(\widehat{O}_t, \boldsymbol{\omega}_t, \boldsymbol{\theta}_t) - \bar{\boldsymbol{h}}(\boldsymbol{\omega}_t, \boldsymbol{\theta}_t)\rangle + \alpha\langle \nabla J(\boldsymbol{\theta}_t), \bar{\boldsymbol{h}}(\boldsymbol{\omega}_t, \boldsymbol{\theta}_t) - \bar{\boldsymbol{h}}(\boldsymbol{\omega}_t^*, \boldsymbol{\theta}_t)\rangle$$

$$+ \alpha\langle \nabla J(\boldsymbol{\theta}_t), \bar{\boldsymbol{h}}(\boldsymbol{\omega}_t^*, \boldsymbol{\theta}_t) - \bar{\boldsymbol{g}}(\boldsymbol{\omega}_t^*, \boldsymbol{\theta}_t)\rangle + \alpha\langle \nabla J(\boldsymbol{\theta}_t), \bar{\boldsymbol{g}}(\boldsymbol{\omega}_t^*, \boldsymbol{\theta}_t)\rangle - \frac{L_g}{2}\alpha^2\|\boldsymbol{h}(\widehat{O}_t, \boldsymbol{\omega}_t, \boldsymbol{\theta}_t)\|^2$$

$$\overset{(1)}{=} J(\boldsymbol{\theta}_t) + \alpha\langle \nabla J(\boldsymbol{\theta}_t), \boldsymbol{h}(\widehat{O}_t, \boldsymbol{\omega}_t, \boldsymbol{\theta}_t) - \bar{\boldsymbol{h}}(\boldsymbol{\omega}_t, \boldsymbol{\theta}_t)\rangle + \alpha\langle \nabla J(\boldsymbol{\theta}_t), \bar{\boldsymbol{h}}(\boldsymbol{\omega}_t, \boldsymbol{\theta}_t) - \bar{\boldsymbol{h}}(\boldsymbol{\omega}_t^*, \boldsymbol{\theta}_t)\rangle$$

$$+ \alpha(1-\gamma)\|\nabla J(\boldsymbol{\theta}_t)\|^2 + \alpha\langle \nabla J(\boldsymbol{\theta}_t), \bar{\boldsymbol{g}}(\boldsymbol{\omega}_t^*, \boldsymbol{\theta}_t)\rangle - \frac{L_g}{2}\alpha^2\|\boldsymbol{h}(\widehat{O}_t, \boldsymbol{\omega}_t, \boldsymbol{\theta}_t)\|^2,$$

where (1) follows from Eq. (28).

Rearranging the above inequality and taking expectation, we have

$$(1-\gamma)\mathbb{E}\|\nabla J(\boldsymbol{\theta}_t)\|^2 \leq \frac{1}{\alpha}(\mathbb{E}[J(\boldsymbol{\theta}_{t+1}) - J(\boldsymbol{\theta}_t)]) + \underbrace{\frac{\alpha L_g}{2}\mathbb{E}\|\boldsymbol{h}(\widehat{O}_t, \boldsymbol{\omega}_t, \boldsymbol{\theta}_t)\|^2}_{I_1} - \underbrace{\mathbb{E}\langle\nabla J(\boldsymbol{\theta}_t), \bar{\boldsymbol{g}}(\boldsymbol{\omega}_t^*, \boldsymbol{\theta}_t)\rangle}_{I_2}$$
$$+ \underbrace{\mathbb{E}\langle\nabla J(\boldsymbol{\theta}_t), \bar{\boldsymbol{h}}(\boldsymbol{\omega}_t, \boldsymbol{\theta}_t) - \boldsymbol{h}(\widehat{O}_t, \boldsymbol{\omega}_t, \boldsymbol{\theta}_t)\rangle}_{I_3} + \underbrace{\mathbb{E}\langle\nabla J(\boldsymbol{\theta}_t), \bar{\boldsymbol{h}}(\boldsymbol{\omega}_t^*, \boldsymbol{\theta}_t) - \bar{\boldsymbol{h}}(\boldsymbol{\omega}_t, \boldsymbol{\theta}_t)\rangle}_{I_4}. \tag{31}$$

In the sequel, we will analyze $I_1, I_2, I_3, I_4$ one by one.

For term $I_1$, since $\boldsymbol{h}(\widehat{O}_t, \boldsymbol{\omega}_t, \boldsymbol{\theta}_t) \leq \bar{\delta}B$, we have

$$I_1 = \frac{\alpha L_g}{2}\mathbb{E}\|\boldsymbol{h}(\widehat{O}_t, \boldsymbol{\omega}_t, \boldsymbol{\theta}_t)\|^2 \leq \frac{\alpha\bar{\delta}^2 B^2 L_g}{2}.$$

For term $I_2$, from Eq. (29), we have

$$I_2 = \mathbb{E}\langle\nabla J(\boldsymbol{\theta}_t), \bar{\boldsymbol{g}}(\boldsymbol{\omega}_t^*, \boldsymbol{\theta}_t)\rangle \leq 2BL_J\epsilon_{\mathrm{app}}.$$

For term $I_3$, from Lemma C.3, we obtain

$$I_3 = \mathbb{E}\big[\Xi(\widehat{O}_t, \boldsymbol{\omega}_t, \boldsymbol{\theta}_t)\big] \leq M_3\frac{1}{\sqrt{T}}.$$

For term $I_4$, it holds that

$$I_4 = \mathbb{E}\langle\nabla J(\boldsymbol{\theta}_t), \bar{\boldsymbol{h}}(\boldsymbol{\omega}_t^*, \boldsymbol{\theta}_t) - \bar{\boldsymbol{h}}(\boldsymbol{\omega}_t, \boldsymbol{\theta}_t)\rangle \leq 2B\mathbb{E}\|\nabla J(\boldsymbol{\theta}_t)\|\|\Delta_t\|.$$

Plugging $I_1, I_2, I_3$ and $I_4$ into Eq. (31), we have

$$(1-\gamma)\mathbb{E}\|\nabla J(\boldsymbol{\theta}_t)\|^2 \leq \frac{1}{\alpha}(\mathbb{E}[J(\boldsymbol{\theta}_{t+1}) - J(\boldsymbol{\theta}_t)]) + \frac{\alpha\bar{\delta}^2 B^2 L_g}{2} + 2BL_J\epsilon_{\mathrm{app}} + M_3\frac{1}{\sqrt{T}} + 2B\mathbb{E}\|\nabla J(\boldsymbol{\theta}_t)\|\|\Delta_t\|.$$

By leveraging the $\mathcal{O}(\cdot)$ notation, we can further summarise our implicit analysis for the actor as

$$(1-\gamma)\mathbb{E}\|\nabla J(\boldsymbol{\theta}_t)\|^2 \leq \frac{1}{\alpha}(\mathbb{E}[J(\boldsymbol{\theta}_{t+1}) - J(\boldsymbol{\theta}_t)]) + 2B\mathbb{E}\|\nabla J(\boldsymbol{\theta}_t)\|\|\Delta_t\| + \mathcal{O}(\frac{\log^2 T}{\sqrt{T}}) + \mathcal{O}(\epsilon_{\mathrm{app}}),$$

where the term $\log^2 T$ arises from the presence of $\tau_{\mathrm{mix}}^2$ in $M_3$. Therefore, we complete the proof of Theorem D.2. $\qquad\square$

### D.3. A novel Lyapunov analysis

**Theorem D.3.** *Choose $\alpha_t = c/\sqrt{T}, \beta_t = 1/\sqrt{T}$, for any $T \geq 2\tau_{\mathrm{mix}}$, we have*

$$\frac{1}{T - \tau_{\mathrm{mix}}}\sum_{t=\tau_{\mathrm{mix}}}^{T-1}\mathbb{E}\|\Delta_t\|^2 = \mathcal{O}\left(\frac{\log^2 T}{\sqrt{T}}\right) + \mathcal{O}(\epsilon_{\mathrm{app}}),$$
$$\frac{1}{T - \tau_{\mathrm{mix}}}\sum_{t=\tau_{\mathrm{mix}}}^{T-1}\mathbb{E}\|\nabla J(\boldsymbol{\theta}_t)\|^2 = \mathcal{O}\left(\frac{\log^2 T}{\sqrt{T}}\right) + \mathcal{O}(\epsilon_{\mathrm{app}}). \tag{32}$$

*Proof.* Define

$$\mathbb{L}_t = \frac{2B}{1-\gamma}\mathbb{E}\|\Delta_t\|^2 + \frac{1-\gamma}{2B}\mathbb{E}\|\nabla J(\boldsymbol{\theta}_t)\|^2,$$

the sum of Eq. (26) and Eq. (30) yields

$$
\begin{aligned}
\mathbb{L}_t &\leq \frac{2B}{\lambda\beta(1-\gamma)}(\mathbb{E}\|\Delta_t\|^2 - \mathbb{E}\|\Delta_{t+1}\|^2) + \frac{4L_cBc}{\lambda}\mathbb{E}\|\Delta_t\|\|\nabla J(\boldsymbol{\theta}_t)\| \\
&\quad + \frac{1}{2B\alpha}(\mathbb{E}[J(\boldsymbol{\theta}_{t+1}) - J(\boldsymbol{\theta}_t)]) + \mathbb{E}\|\Delta_t\|\|\nabla J(\boldsymbol{\theta}_t)\| + \mathcal{O}(\frac{\log^2 T}{\sqrt{T}}) + \mathcal{O}(\epsilon_{\mathrm{app}}) \\
&\leq \left(\frac{2L_cBc}{\lambda} + \frac{1}{2}\right)\mathbb{L}_t + \frac{2B}{\lambda\beta(1-\gamma)}(\mathbb{E}\|\Delta_t\|^2 - \mathbb{E}\|\Delta_{t+1}\|^2) \\
&\quad + \frac{1}{2B\alpha}(\mathbb{E}[J(\boldsymbol{\theta}_{t+1}) - J(\boldsymbol{\theta}_t)]) + \mathcal{O}\left(\frac{\log^2 T}{\sqrt{T}}\right) + \mathcal{O}(\epsilon_{\mathrm{app}}),
\end{aligned}
$$

where the last inequality follows from $\mathbb{E}\|\Delta_t\|\|\nabla J(\boldsymbol{\theta}_t)\| \leq 1/2\mathbb{L}_t$. Since $\mathcal{L} := 1/(T-\tau_{\mathrm{mix}})\sum_{t=\tau_{\mathrm{mix}}}^{T-1}\mathbb{L}_t$, it can be shown that

$$
\begin{aligned}
\mathcal{L} &\leq \left(\frac{2L_cBc}{\lambda} + \frac{1}{2}\right)\mathcal{L} + \frac{2B}{\lambda\beta(1-\gamma)(T-\tau_{\mathrm{mix}})}\sum_{t=\tau_{\mathrm{mix}}}^{T-1}(\mathbb{E}\|\Delta_t\|^2 - \mathbb{E}\|\Delta_{t+1}\|^2) \\
&\quad + \frac{1}{2B\alpha(T-\tau_{\mathrm{mix}})}\sum_{t=\tau_{\mathrm{mix}}}^{T-1}(\mathbb{E}[J(\boldsymbol{\theta}_{t+1}) - J(\boldsymbol{\theta}_t)]) + \mathcal{O}\left(\frac{\log^2 T}{\sqrt{T}}\right) + \mathcal{O}(\epsilon_{\mathrm{app}}) \\
&\leq \left(\frac{2L_cBc}{\lambda} + \frac{1}{2}\right)\mathcal{L} + \frac{8B\bar{\omega}^2}{\lambda\beta(1-\gamma)(T-\tau_{\mathrm{mix}})} + \frac{\bar{r}}{B\alpha(T-\tau_{\mathrm{mix}})} + \mathcal{O}\left(\frac{\log^2 T}{\sqrt{T}}\right) + \mathcal{O}(\epsilon_{\mathrm{app}}).
\end{aligned}
$$

Choose $T \geq 2\tau_{\mathrm{mix}}$, we have $T - \tau_{\mathrm{mix}} \geq 1/2T$, which implies

$$
\begin{aligned}
\mathcal{L} &\leq \left(\frac{2L_cBc}{\lambda} + \frac{1}{2}\right)\mathcal{L} + \left(\frac{16B\bar{\omega}^2}{\lambda(1-\gamma)} + \frac{2\bar{r}}{Bc}\right)\frac{1}{\sqrt{T}} + \mathcal{O}\left(\frac{\log^2 T}{\sqrt{T}}\right) + \mathcal{O}(\epsilon_{\mathrm{app}}) \\
&= \left(\frac{2L_cBc}{\lambda} + \frac{1}{2}\right)\mathcal{L} + \mathcal{O}\left(\frac{\log^2 T}{\sqrt{T}}\right) + \mathcal{O}(\epsilon_{\mathrm{app}}).
\end{aligned}
$$

Overall, we have

$$
\mathcal{L} \leq \left(\frac{2L_cBc}{\lambda} + \frac{1}{2}\right)\mathcal{L} + \mathcal{O}\left(\frac{\log^2 T}{\sqrt{T}}\right) + \mathcal{O}(\epsilon_{\mathrm{app}}). \tag{33}
$$

To make $\mathcal{L}$ convergence, we need $2L_cBc/\lambda + 1/2 < 1$, which can be achieved by choosing

$$
c < \frac{\lambda}{4BL_c}. \tag{34}
$$

It follows that

$$
\mathcal{L} = \mathcal{O}\left(\frac{\log^2 T}{\sqrt{T}}\right) + \mathcal{O}(\epsilon_{\mathrm{app}}),
$$

which implies

$$
\frac{1}{T-\tau_{\mathrm{mix}}}\sum_{t=\tau_{\mathrm{mix}}}^{T-1}\mathbb{E}\|\Delta_t\|^2 = \mathcal{O}\left(\frac{\log^2 T}{\sqrt{T}}\right) + \mathcal{O}(\epsilon_{\mathrm{app}}),
$$

$$
\frac{1}{T-\tau_{\mathrm{mix}}}\sum_{t=\tau_{\mathrm{mix}}}^{T-1}\mathbb{E}\|\nabla J(\boldsymbol{\theta}_t)\|^2 = \mathcal{O}\left(\frac{\log^2 T}{\sqrt{T}}\right) + \mathcal{O}(\epsilon_{\mathrm{app}}).
$$

Therefore, we complete our proof. $\qquad\square$

# E. Proof of Propositions

**Proof of Proposition 3.1.**

*Proof.* We show that $\nu_{\boldsymbol{\theta}}$ is the stationary distribution of the Markov chain induced by $\widehat{\mathcal{P}}$ by showing that $\nu_{\boldsymbol{\theta}}$ is a fixed point of the operator $\widehat{\mathcal{P}}$, i.e.,

$$\widehat{\mathcal{P}}\nu_{\boldsymbol{\theta}} = \nu_{\boldsymbol{\theta}}.$$

Define operator $\mathcal{P}^t$ by iterative application of the operator $\mathcal{P}$:

$$(\mathcal{P}^t f)(s') = \int_{\mathcal{S}} \int_{\mathcal{A}} \pi_{\boldsymbol{\theta}}(a \,|\, s) P(s' \,|\, s, a) \mathcal{P}^{t-1} f(s) \, da ds.$$

From the definition of the operator $\mathcal{P}^t$, we can rewrite $\nu_{\boldsymbol{\theta}}$ in Eq. (4) as

$$\nu_{\boldsymbol{\theta}}(s) = (1 - \gamma) \sum_{t=0}^{\infty} \gamma^t \mathcal{P}^t \eta(s).$$

Then we have

$$\widehat{\mathcal{P}}\nu_{\boldsymbol{\theta}}(s) = (1 - \gamma)\eta(s) + \gamma \mathcal{P}\nu_{\boldsymbol{\theta}}(s). \tag{35}$$

For term $\mathcal{P}\nu_{\boldsymbol{\theta}}(s)$, it holds that

$$\begin{aligned}
\mathcal{P}\nu_{\boldsymbol{\theta}}(s) &= (1 - \gamma) \sum_{t=0}^{\infty} \gamma^t \mathcal{P}^{t+1} \eta(s) \\
&= (1 - \gamma) \sum_{k=1}^{\infty} \gamma^{k-1} \mathcal{P}^k \eta(s) \\
&= \frac{1 - \gamma}{\gamma} \sum_{k=1}^{\infty} \gamma^k \mathcal{P}^k \eta(s) \\
&= \frac{1 - \gamma}{\gamma} \Big( \frac{\nu_{\boldsymbol{\theta}}(s)}{1 - \gamma} - \eta(s) \Big) \\
&= \frac{\nu_{\boldsymbol{\theta}}(s)}{\gamma} - \frac{1 - \gamma}{\gamma} \eta(s).
\end{aligned}$$

Plugging the above result to Eq. (35), we obtain

$$\begin{aligned}
\widehat{\mathcal{P}}\nu_{\boldsymbol{\theta}}(s) &= (1 - \gamma)\eta(s) + \gamma \Big( \frac{\nu_{\boldsymbol{\theta}}(s)}{\gamma} - \frac{1 - \gamma}{\gamma} \eta(s) \Big) \\
&= (1 - \gamma)\eta(s) + \nu_{\boldsymbol{\theta}}(s) - (1 - \gamma)\eta(s) \\
&= \nu_{\boldsymbol{\theta}}(s).
\end{aligned}$$

Suppose $\widehat{\mathcal{P}}$ has two fixed points $f$ and $g$, then we have

$$\widehat{\mathcal{P}}f = f, \quad \widehat{\mathcal{P}}g = g.$$

Recall that for two probability distributions $\mu$ and $\nu$ on $\mathcal{S}$, the total variation distance is defined as

$$d_{\text{TV}}(\mu, \nu) = \frac{1}{2} \int_{\mathcal{S}} |\mu(s) - \nu(s)| \, ds.$$

Since we have

$$\widehat{\mathcal{P}}f - \widehat{\mathcal{P}}g = \gamma(\mathcal{P}f - \mathcal{P}g),$$

it follows that

$$d_{\mathrm{TV}}(\widehat{\mathcal{P}}f, \widehat{\mathcal{P}}g) = \gamma d_{TV}(\mathcal{P}f, \mathcal{P}g).$$

From Eq. (10), we know that $\mathcal{P}$ is also a Markov kernel which does not increase the total variation distance. Therefore, it can be shown that

$$d_{\mathrm{TV}}(f, g) = d_{\mathrm{TV}}(\widehat{\mathcal{P}}f, \widehat{\mathcal{P}}g) = \gamma d_{TV}(\mathcal{P}f, \mathcal{P}g) \le \gamma d_{\mathrm{TV}}(f, g).$$

Therefore, we get

$$d_{\mathrm{TV}}(f, g) = 0,$$

which means $f$ and $g$ are same distributions. Hence we finish our proof. $\qquad\square$

**Proof of Proposition 3.2.**

*Proof.* Recall that for two probability distributions $\mu$ and $\nu$ on $\mathcal{S}$, the total variation distance is defined as

$$d_{\mathrm{TV}}(\mu, \nu) = \frac{1}{2}\int_{\mathcal{S}}|\mu(s) - \nu(s)|\,ds.$$

For two distribution $f$ and $g$, we have

$$\widehat{\mathcal{P}}f - \widehat{\mathcal{P}}g = \gamma(\mathcal{P}f - \mathcal{P}g),$$

where the operator $\widehat{\mathcal{P}}$ and $\mathcal{P}$ are defined in the proof of Proposition 3.1.

It follows that

$$d_{\mathrm{TV}}(\widehat{\mathcal{P}}f, \widehat{\mathcal{P}}g) = \gamma d_{TV}(\mathcal{P}f, \mathcal{P}g).$$

From Eq. (10), we know that $\mathcal{P}$ is also a Markov kernel which does not increase the total variation distance. Therefore, it can be shown that

$$d_{\mathrm{TV}}(\widehat{\mathcal{P}}f, \widehat{\mathcal{P}}g) = \gamma d_{TV}(\mathcal{P}f, \mathcal{P}g) \le \gamma d_{\mathrm{TV}}(f, g).$$

As shown in Proposition 3.1, $\nu_{\boldsymbol{\theta}}$ is the stationary distribution of the Markov chain induced by $\widehat{\mathcal{P}}$. For any initial distribution $f$, we have

$$\begin{aligned}d_{\mathrm{TV}}(\widehat{\mathcal{P}}^t f, \nu_{\boldsymbol{\theta}}) &= d_{\mathrm{TV}}(\widehat{\mathcal{P}}^t f, \widehat{\mathcal{P}}^t \nu_{\boldsymbol{\theta}})\\ &\le \gamma^t d_{\mathrm{TV}}(f, \nu_{\boldsymbol{\theta}})\\ &\le \gamma^t.\end{aligned}$$

Thus, it completes the proof. $\qquad\square$

**Proof of Proposition 4.4.**

*Proof.* From the definition of the total variation distance, we have

$$\begin{aligned}d_{\mathrm{TV}}(\pi_{\boldsymbol{\theta}_1}(\cdot\,|\,s) - \pi_{\boldsymbol{\theta}_2}(\cdot\,|\,s)) &= \frac{1}{2}\int_{\mathcal{A}}|\pi_{\boldsymbol{\theta}_1}(a\,|\,s) - \pi_{\boldsymbol{\theta}_2}(a\,|\,s)|\,da\\ &= \frac{1}{2}\int_{\bar{\mathcal{A}}}|\pi_{\boldsymbol{\theta}_1}(a\,|\,s) - \pi_{\boldsymbol{\theta}_2}(a\,|\,s)|\,da\\ &\le \frac{1}{2}\int_{\bar{\mathcal{A}}}L\|\boldsymbol{\theta}_1 - \boldsymbol{\theta}_2\|\,da\\ &\le \frac{1}{2}\bar{A}L\|\boldsymbol{\theta}_1 - \boldsymbol{\theta}_2\|,\end{aligned}$$

where $\bar{\mathcal{A}}$ is the bounded support of $\pi_{\boldsymbol{\theta}}(a\,|\,s)$ which satisfies $\int_{\bar{\mathcal{A}}}da = \bar{A}$. Define $L_{\pi} := 1/2\bar{A}L$, which completes the proof. $\qquad\square$

# F. Proof of Preliminary Lemmas

**Proof of Lemma B.1.**

*Proof.* This is a minor adjustment to the proof of Lemma 3 in Zou et al. (2019), extending it to continuous settings.

For any $\boldsymbol{\theta}_1$ and $\boldsymbol{\theta}_2$, define the transition densities respectively as follows:

$$P_{\boldsymbol{\theta}_i}(s \,|\, ds') = \int_{\mathcal{A}} P(ds' \,|\, s, a) \pi_{\boldsymbol{\theta}_i}(a \,|\, s), \quad i = 1, 2$$

Following from Theorem 3.1 in (Mitrophanov, 2005), we obtain

$$d_{\mathrm{TV}}(\mu_{\boldsymbol{\theta}_1}, \mu_{\boldsymbol{\theta}_2}) \le (\lceil \log_\rho m^{-1} \rceil + \frac{1}{1-\rho}) \| P_{\boldsymbol{\theta}_1} - P_{\boldsymbol{\theta}_2} \|_{\mathrm{op}},$$

where $\| \cdot \|_{\mathrm{op}}$ is the operator norm defined in (Mitrophanov, 2005): $\|A\|_{\mathrm{op}} := \sup_{\|q\|_{\mathrm{TV}}=1} \|qA\|_{\mathrm{TV}}$, and $\| \cdot \|_{\mathrm{TV}}$ denotes the total-variation norm. Then we have

$$
\begin{aligned}
\| P_{\boldsymbol{\theta}_1} - P_{\boldsymbol{\theta}_2} \|_{\mathrm{op}} &= \sup_{\|q\|_{\mathrm{TV}}=1} \| \int_{\mathcal{S}} q(ds)(P_{\boldsymbol{\theta}_1} - P_{\boldsymbol{\theta}_2})(s \,|\, \cdot) \|_{\mathrm{TV}} \\
&= \sup_{\|q\|_{\mathrm{TV}}=1} \int_{\mathcal{S}} | \int_{\mathcal{S}} q(ds)(P_{\boldsymbol{\theta}_1} - P_{\boldsymbol{\theta}_2})(s \,|\, ds')| \\
&\le \sup_{\|q\|_{\mathrm{TV}}=1} \int_{\mathcal{S}} \int_{\mathcal{S}} |q(ds)| \Big| (P_{\boldsymbol{\theta}_1} - P_{\boldsymbol{\theta}_2})(s \,|\, ds')| \\
&= \sup_{\|q\|_{\mathrm{TV}}=1} \int_{\mathcal{S}} \int_{\mathcal{S}} |q(ds)| \Big| \int_{\mathcal{A}} P(ds' \,|\, s, a)(\pi_{\boldsymbol{\theta}_1}(da \,|\, s) - \pi_{\boldsymbol{\theta}_2}(da \,|\, s))| \\
&= \sup_{\|q\|_{\mathrm{TV}}=1} \int_{\mathcal{S}} \int_{\mathcal{S}} |q(ds)| \int_{\mathcal{A}} P(ds' \,|\, s, a) |(\pi_{\boldsymbol{\theta}_1}(da \,|\, s) - \pi_{\boldsymbol{\theta}_2}(da \,|\, s))| \\
&= \sup_{\|q\|_{\mathrm{TV}}=1} \int_{\mathcal{S}} |q(ds)| \int_{\mathcal{A}} |(\pi_{\boldsymbol{\theta}_1}(da \,|\, s) - \pi_{\boldsymbol{\theta}_2}(da \,|\, s))| \\
&\le 2 L_\pi \|\boldsymbol{\theta}_1 - \boldsymbol{\theta}_2\|.
\end{aligned}
$$

Therefore, we have

$$d_{TV}(\mu_{\boldsymbol{\theta}_1}, \mu_{\boldsymbol{\theta}_2}) \le 2 L_\pi (\lceil \log_\rho m^{-1} \rceil + \frac{1}{1-\rho}) \|\boldsymbol{\theta}_1 - \boldsymbol{\theta}_2\|.$$

For the second inequality, we have

$$
\begin{aligned}
d_{TV}(\mu_{\boldsymbol{\theta}_1} \otimes \pi_{\boldsymbol{\theta}_1}, \mu_{\boldsymbol{\theta}_2} \otimes \pi_{\boldsymbol{\theta}_2}) &= \frac{1}{2} \int_{\mathcal{S}} \int_{\mathcal{A}} |\mu_{\boldsymbol{\theta}_1}(ds)\pi_{\boldsymbol{\theta}_1}(da \,|\, s) - \mu_{\boldsymbol{\theta}_2}(ds)\pi_{\boldsymbol{\theta}_2}(da \,|\, s)| \\
&\le \frac{1}{2} \int_{\mathcal{S}} \int_{\mathcal{A}} |\mu_{\boldsymbol{\theta}_1}(ds)(\pi_{\boldsymbol{\theta}_1}(da \,|\, s) - \pi_{\boldsymbol{\theta}_2}(da \,|\, s))| \\
&\quad + \frac{1}{2} \int_{\mathcal{S}} \int_{\mathcal{A}} |(\mu_{\boldsymbol{\theta}_1}(ds) - \mu_{\boldsymbol{\theta}_2}(ds))\pi_{\boldsymbol{\theta}_2}(da \,|\, s))| \\
&= d_{TV}(\pi_{\boldsymbol{\theta}_1}, \pi_{\boldsymbol{\theta}_2}) + d_{TV}(\mu_{\boldsymbol{\theta}_1}, \mu_{\boldsymbol{\theta}_2}) \\
&\le L_\pi \|\boldsymbol{\theta}_1 - \boldsymbol{\theta}_2\| + 2 L_\pi (\lceil \log_\rho m^{-1} \rceil + \frac{1}{1-\rho}) \|\boldsymbol{\theta}_1 - \boldsymbol{\theta}_2\| \\
&= 2 L_\pi (1 + \lceil \log_\rho m^{-1} \rceil + \frac{1}{1-\rho}) \|\boldsymbol{\theta}_1 - \boldsymbol{\theta}_2\|.
\end{aligned}
$$

For the third inequality, we have

$$d_{\mathrm{TV}}(\mu_{\boldsymbol{\theta}_1} \otimes \pi_{\boldsymbol{\theta}_1} \otimes \mathcal{P}, \mu_{\boldsymbol{\theta}_2} \otimes \pi_{\boldsymbol{\theta}_2} \otimes \mathcal{P})$$

$$= \frac{1}{2} \int_S \int_{\mathcal{A}} \int_S |\mu_{\boldsymbol{\theta}_1}(ds)\pi_{\boldsymbol{\theta}_1}(da\,|\,s)P(ds'\,|\,s,a) - \mu_{\boldsymbol{\theta}_2}(ds)\pi_{\boldsymbol{\theta}_2}(da\,|\,s)P(ds'\,|\,s,a)|$$

$$= \frac{1}{2} \int_S \int_{\mathcal{A}} |\mu_{\boldsymbol{\theta}_1}(ds)\pi_{\boldsymbol{\theta}_1}(da\,|\,s) - \mu_{\boldsymbol{\theta}_2}(ds)\pi_{\boldsymbol{\theta}_2}(da\,|\,s)|$$

$$= d_{TV}(\mu_{\boldsymbol{\theta}_1} \otimes \pi_{\boldsymbol{\theta}_1}, \mu_{\boldsymbol{\theta}_2} \otimes \pi_{\boldsymbol{\theta}_2}),$$

which concludes the proof. $\qquad\square$

**Proof of Lemma B.2.**

*Proof.* This is a slight modification of the proof of Lemma B.2 in Wu et al. (2020), which extends it to continuous settings. From the fact that

$$\mathbb{P}(s_{t+1} \in \cdot) = \int_S \int_{\mathcal{A}} \mathbb{P}(s_t = ds, a_t = da, s_{t+1} \in \cdot),$$

we have

$$d_{\mathrm{TV}}(\mathbb{P}(s_{t+1} \in \cdot), \mathbb{P}(\tilde{s}_{t+1} \in \cdot))$$

$$= \frac{1}{2} \int_S |\int_S \int_{\mathcal{A}} \mathbb{P}(s_t = ds, a_t = da, s_{t+1} = ds') - \int_S \int_{\mathcal{A}} \mathbb{P}(\tilde{s}_t = ds, \tilde{a}_t = da, \tilde{s}_{t+1} = ds')|$$

$$\leq \frac{1}{2} \int_S \int_S \int_{\mathcal{A}} |\mathbb{P}(s_t = ds, a_t = da, s_{t+1} = ds') - \mathbb{P}(\tilde{s}_t = ds, \tilde{a}_t = da, \tilde{s}_{t+1} = ds')|$$

$$= \frac{1}{2} \int_S \int_S \int_{\mathcal{A}} |\mathbb{P}(O_t = (ds, da, ds')) - \mathbb{P}(\tilde{O}_t = (ds, da, ds'))|$$

$$= d_{TV}(\mathbb{P}(O_t \in \cdot), \mathbb{P}(\tilde{O} \in \cdot)),$$

where the last equality requires the exchange of integral which is guaranteed by Fubini's theorem since $\mathbb{P}$ is an absolute integrable function.

For the second equality, we have

$$d_{TV}(\mathbb{P}(O_t \in \cdot), \mathbb{P}(\tilde{O}_t \in \cdot))$$

$$= \int_S \int_{\mathcal{A}} \int_S |\mathbb{P}(O_t = (ds, da, ds')) - \mathbb{P}(\tilde{O}_t = (ds, da, ds'))|$$

$$= \frac{1}{2} \int_S \int_{\mathcal{A}} \int_S |P(ds'|s, a)\mathbb{P}((s_t, a_t) = (ds, da)) - P(ds'|s, a)\mathbb{P}((\tilde{s}_t, \tilde{a}_t) = (ds, da))|$$

$$= \frac{1}{2} \int_S \int_{\mathcal{A}} \int_S P(ds'|s, a)|\mathbb{P}((s_t, a_t) = (ds, da)) - \mathbb{P}((\tilde{s}_t, \tilde{a}_t) = (ds, da))|$$

$$= \frac{1}{2} \int_S \int_{\mathcal{A}} |\mathbb{P}((s_t, a_t) = (ds, da)) - \mathbb{P}((\tilde{s}_t, \tilde{a}_t) = (ds, da))|$$

$$= d_{TV}(\mathbb{P}((s_t, a_t) \in \cdot), \mathbb{P}((\tilde{s}_t, \tilde{a}_t) \in \cdot)).$$

For the third inequality, since $\boldsymbol{\theta}_t$ is dependent on $s_t$, it holds that

$$d_{\mathrm{TV}}(\mathbb{P}((s_t, a_t) \in \cdot), \mathbb{P}((\tilde{s}_t, \tilde{a}_t) \in \cdot))$$

$$= \frac{1}{2} \int_{\mathcal{S}} \int_{\mathcal{A}} |\mathbb{P}(s_t = ds, a_t = da) - \mathbb{P}(\tilde{s}_t = ds, \tilde{a}_t = da)|$$

$$= \frac{1}{2} \int_{\mathcal{S}} \int_{\mathcal{A}} | \int_{\boldsymbol{\theta}} \mathbb{P}(s_t = ds) \mathbb{P}(\boldsymbol{\theta}_t = d\boldsymbol{\theta} \,|\, s_t = s) \mathbb{P}(a_t = da \,|\, s_t = s, \boldsymbol{\theta}_t = \boldsymbol{\theta}) - \mathbb{P}(\tilde{s}_t = ds, \tilde{a}_t = da)|$$

$$= \frac{1}{2} \int_{\mathcal{S}} \int_{\mathcal{A}} |\mathbb{P}(s_t = ds) \int_{\boldsymbol{\theta}} \mathbb{P}(\boldsymbol{\theta}_t = d\boldsymbol{\theta} \,|\, s_t = s) \pi_{\boldsymbol{\theta}_t}(da \,|\, s) - \mathbb{P}(\tilde{s}_t = ds) \pi_{\boldsymbol{\theta}_{t-\tau}}(da \,|\, s)|$$

$$= \frac{1}{2} \int_{\mathcal{S}} \int_{\mathcal{A}} |\mathbb{P}(s_t = ds) \mathbb{E}[\pi_{\boldsymbol{\theta}_t}(da \,|\, s) \,|\, s_t = s] - \mathbb{P}(\tilde{s}_t = ds) \pi_{\boldsymbol{\theta}_{t-\tau}}(da \,|\, s)|$$

$$= \frac{1}{2} \int_{\mathcal{S}} \int_{\mathcal{A}} |\mathbb{P}(s_t = ds) \mathbb{E}[\pi_{\boldsymbol{\theta}_t}(da \,|\, s) \,|\, s_t = s] - \mathbb{P}(s_t = ds) \pi_{\boldsymbol{\theta}_{t-\tau}}(da \,|\, s)|$$

$$+ \frac{1}{2} \int_{\mathcal{S}} \int_{\mathcal{A}} |\mathbb{P}(s_t = ds) \pi_{\boldsymbol{\theta}_{t-\tau}}(da \,|\, s) - \mathbb{P}(\tilde{s}_t = ds) \pi_{\boldsymbol{\theta}_{t-\tau}}(da \,|\, s)|$$

$$= \frac{1}{2} \int_{\mathcal{S}} \mathbb{P}(s_t = ds) \int_{\mathcal{A}} |\mathbb{E}[\pi_{\boldsymbol{\theta}_t}(da|s)|s_t = s] - \pi_{\boldsymbol{\theta}_{t-\tau}}(da|s)|$$

$$+ d_{\mathrm{TV}}(\mathbb{P}(s_t \in \cdot), \mathbb{P}(\tilde{s}_t \in \cdot))$$

$$\leq L_\pi \mathbb{E}\|\boldsymbol{\theta}_t - \boldsymbol{\theta}_{t-\tau}\| + d_{TV}(\mathbb{P}(s_t \in \cdot), \mathbb{P}(\tilde{s}_t \in \cdot)).$$

Therefore, we finish our proof. $\square$

**Proof of Lemma B.3.**

*Proof.* Following the same proof as shown in Lemma B.1. The final results are derived by substituting the results of Lemma B.1 with $m = 1$ and $\rho = \gamma$, as outlined in Proposition 3.2. $\square$

**Proof of Lemma B.4.**

*Proof.* By the same proof as shown in Lemma B.2. $\square$

**Proof of Lemma B.5.**

*Proof.* By definition, we have

$$J(\theta_1) - J(\theta_2) = \mathbb{E}[r(s^1, a^1) - r(s^2, a^2)],$$

where $s^i \sim \nu_{\boldsymbol{\theta}_i}, a^i \sim \pi_{\boldsymbol{\theta}_i}$. Therefore, it holds that

$$J(\boldsymbol{\theta}_1) - J(\boldsymbol{\theta}_2) = \mathbb{E}[r(s^1, a^1) - r(s^1, a^1)]$$

$$\leq 2\bar{r} d_{TV}(\nu_{\boldsymbol{\theta}_1} \otimes \pi_{\boldsymbol{\theta}_1}, \nu_{\boldsymbol{\theta}_2} \otimes \pi_{\boldsymbol{\theta}_2})$$

$$\leq 4\bar{r} L_\pi (1 + \frac{1}{1-\gamma}) \|\boldsymbol{\theta}_1 - \boldsymbol{\theta}_2\|$$

$$= L_J \|\boldsymbol{\theta}_1 - \boldsymbol{\theta}_2\|.$$

$\square$

**Proof of Lemma B.6.**

*Proof.* From Eq. (16), we have

$$\boldsymbol{A_\theta} \boldsymbol{\omega}^*(\boldsymbol{\theta}) = \boldsymbol{b_\theta}.$$

where $\boldsymbol{A_\theta} := \mathbb{E}_{(s,a,s')}[\phi(s)(\phi(s) - \gamma\phi(s'))^\top)]$ and $\boldsymbol{b_\theta} := \mathbb{E}_{(s,a)}[r(s,a)\phi(s)]$. The expectation is taken over the stationary distribution $s \sim \mu_{\boldsymbol{\theta}}$, the action $a \sim \pi_{\boldsymbol{\theta}}(\cdot\,|\,s)$, and the transition probability kernel $s' \sim P(\cdot\,|\,s,a)$.

Denote $\boldsymbol{\omega}_1^*, \boldsymbol{\omega}_2^*, \hat{\boldsymbol{\omega}}_1$ as the unique solutions of the following equations respectively:

$$\boldsymbol{A}_{\boldsymbol{\theta}_1}\boldsymbol{\omega}_1^* = \boldsymbol{b}_{\boldsymbol{\theta}_1}, \quad \boldsymbol{A}_{\boldsymbol{\theta}_2}\hat{\boldsymbol{\omega}}_1 = \boldsymbol{b}_1, \quad \boldsymbol{A}_{\boldsymbol{\theta}_2}\boldsymbol{\omega}_2^* = \boldsymbol{b}_2.$$

First we bound $\|\boldsymbol{\omega}_1^* - \hat{\boldsymbol{\omega}}_1\|$. By definition, we have

$$\|\boldsymbol{\omega}_1^* - \hat{\boldsymbol{\omega}}_1\| \leq \|\boldsymbol{A}_{\boldsymbol{\theta}_1}^{-1} - \boldsymbol{A}_{\boldsymbol{\theta}_2}^{-1}\|\|\boldsymbol{b}_{\boldsymbol{\theta}_1}\|.$$

It can be shown that

$$\boldsymbol{A}_{\boldsymbol{\theta}_1}^{-1} - \boldsymbol{A}_{\boldsymbol{\theta}_2}^{-1} = \boldsymbol{A}_{\boldsymbol{\theta}_1}^{-1}(\boldsymbol{A}_{\boldsymbol{\theta}_2} - \boldsymbol{A}_{\boldsymbol{\theta}_1})\boldsymbol{A}_{\boldsymbol{\theta}_2}^{-1},$$

which implies

$$\|\boldsymbol{\omega}_1^* - \hat{\boldsymbol{\omega}}_1\| \leq \|\boldsymbol{A}_{\boldsymbol{\theta}_1}^{-1}\|\|\boldsymbol{A}_{\boldsymbol{\theta}_1} - \boldsymbol{A}_{\boldsymbol{\theta}_2}\|\|\boldsymbol{A}_{\boldsymbol{\theta}_2}^{-1}\|\|\boldsymbol{b}_{\boldsymbol{\theta}_1}\|.$$

Then we bound $\|\hat{\boldsymbol{\omega}}_1 - \boldsymbol{\omega}_2^*\|$:

$$\|\hat{\boldsymbol{\omega}}_1 - \boldsymbol{\omega}_2^*\| \leq \|\boldsymbol{A}_{\boldsymbol{\theta}_2}^{-1}\|\|\boldsymbol{b}_{\boldsymbol{\theta}_1} - \boldsymbol{b}_{\boldsymbol{\theta}_2}\|.$$

By Assumption 4.1, the eigenvalues of $\boldsymbol{A}_{\boldsymbol{\theta}_i}$ are bounded from below by $\lambda > 0$, therefore $\|\boldsymbol{A}_{\boldsymbol{\theta}_i}^{-1}\| \leq \lambda^{-1}$. Also $\|\boldsymbol{b}_{\boldsymbol{\theta}_i}\| \leq \bar{r}$, due to the assumption that $|r(s,a)| \leq \bar{r}$, and $\|\phi(s)\| \leq 1$. To bound $\|\boldsymbol{A}_{\boldsymbol{\theta}_1} - \boldsymbol{A}_{\boldsymbol{\theta}_2}\|$ and $\|\boldsymbol{b}_{\boldsymbol{\theta}_1} - \boldsymbol{b}_{\boldsymbol{\theta}_2}\|$, we first note that

$$\|\boldsymbol{A}_{\boldsymbol{\theta}_1} - \boldsymbol{A}_{\boldsymbol{\theta}_2}\| \leq 2\sup_{s,s'\in\mathcal{S}}\|\phi(s)(\phi(s) - \gamma\phi(s')^\top)\| \cdot 2d_{TV}(\mathbb{P}(O^1 \in \cdot), \mathbb{P}(O^2 \in \cdot))$$

$$\leq 4d_{\mathrm{TV}}(\mathbb{P}(O^1 \in \cdot), \mathbb{P}(O^2 \in \cdot)),$$

and

$$\|\boldsymbol{b}_{\boldsymbol{\theta}_1} - \boldsymbol{b}_{\boldsymbol{\theta}_2}\| \leq \|\mathbb{E}[r(s^1,a^1)\phi(s^1)] - \mathbb{E}[r(s^2,a^2)\phi(s^2)]\|$$

$$\leq 2\bar{r}d_{\mathrm{TV}}(\mathbb{P}(O^1 \in \cdot), \mathbb{P}(O^2 \in \cdot)),$$

where $O^i$ is the tuple obtained by $s^i \sim \mu_{\boldsymbol{\theta}_i}, a^i \sim \pi_{\boldsymbol{\theta}_i}(\cdot|s^i)$, and $s' \sim P(\cdot|s^i, a^i)$. And the total variation norm can be bounded by Lemma B.1 as:

$$d_{\mathrm{TV}}(\mathbb{P}(O^1 \in \cdot), \mathbb{P}(O^2 \in \cdot)) \leq 2L_\pi(1 + \lceil\log_\rho m^{-1}\rceil + \frac{1}{1-\rho})\|\boldsymbol{\theta}_1 - \boldsymbol{\theta}_2\|$$

Collecting the above results, we have

$$\|\boldsymbol{\omega}_2^* - \boldsymbol{\omega}_1^*\| \leq \|\boldsymbol{\omega}_1^* - \hat{\boldsymbol{\omega}}_1\| + \|\hat{\boldsymbol{\omega}}_1 - \boldsymbol{\omega}_2^*\|$$

$$\leq (8\lambda^{-2}\bar{r} + 4\lambda^{-1}\bar{r})L_\pi\left(1 + \lceil\log_\rho m^{-1}\rceil + \frac{1}{1-\rho}\right)\|\boldsymbol{\theta}_1 - \boldsymbol{\theta}_2\|,$$

and we set $L_c := (8\lambda^{-2}\bar{r} + 4\lambda^{-1}\bar{r})L_\pi(1 + \lceil\log_\rho m^{-1}\rceil + 1/(1-\rho))$ to obtain the final result. □

**Proof of Lemma B.7.**

*Proof.* Lemma B.7 is adopted as an assumption in Chen et al. (2021) and Chen & Zhao (2024), but it directly follows from Heidergott & Hordijk (2003), as pointed out by Olshevsky & Gharesifard (2023). □

**Proof of Lemma B.8.**

*Proof.* See the proof in Lemma 3.2 of Zhang et al. (2020a). □

**Proof of Lemma B.9.**

*Proof.* See the proof in Proposition 8 of Chen et al. (2021). □

# G. Proof of Markovian Noise

**Proof of Lemma C.1.**

*Proof.* We will divide the proof of this lemma into five steps.

**Step 1.** show that for any $\boldsymbol{\theta}_1, \boldsymbol{\theta}_1, \boldsymbol{\omega}$, and tuple $O(s, a, s')$, we have

$$\Lambda(O, \boldsymbol{\omega}, \boldsymbol{\theta}_1) - \Lambda(O, \boldsymbol{\omega}, \boldsymbol{\theta}_2) \le (8\bar{\omega}\bar{\delta}L_\pi(1 + \lceil \log_\rho m^{-1} \rceil + \frac{1}{1-\rho}) + 2\bar{\delta}L_c)\|\boldsymbol{\theta}_1 - \boldsymbol{\theta}_2\|. \tag{36}$$

By the definition of $\Lambda(O, \boldsymbol{\omega}, \boldsymbol{\theta})$ in Eq. (25), we have

$$
\begin{aligned}
\Lambda(O, \boldsymbol{\omega}, \boldsymbol{\theta}_1) - \Lambda(O, \boldsymbol{\omega}, \boldsymbol{\theta}_2) =& \langle \boldsymbol{\omega} - \boldsymbol{\omega}_1^*, \boldsymbol{f}(O, \boldsymbol{\omega}) - \bar{\boldsymbol{f}}(\boldsymbol{\omega}, \boldsymbol{\theta}_1) \rangle - \langle \boldsymbol{\omega} - \boldsymbol{\omega}_2^*, \boldsymbol{f}(O, \boldsymbol{\omega}) - \bar{\boldsymbol{f}}(\boldsymbol{\omega}, \boldsymbol{\theta}_2) \rangle \\
\le& \underbrace{\left| \langle \boldsymbol{\omega} - \boldsymbol{\omega}_1^*, \boldsymbol{f}(O, \boldsymbol{\omega}) - \bar{\boldsymbol{f}}(\boldsymbol{\omega}, \boldsymbol{\theta}_1) \rangle - \langle \boldsymbol{\omega} - \boldsymbol{\omega}_1^*, \boldsymbol{f}(O, \boldsymbol{\omega}) - \bar{\boldsymbol{f}}(\boldsymbol{\omega}, \boldsymbol{\theta}_2) \rangle \right|}_{I_1} \\
&+ \underbrace{\left| \langle \boldsymbol{\omega} - \boldsymbol{\omega}_1^*, \boldsymbol{f}(O, \boldsymbol{\omega}) - \bar{\boldsymbol{f}}(\boldsymbol{\omega}, \boldsymbol{\theta}_2) \rangle - \langle \boldsymbol{\omega} - \boldsymbol{\omega}_2^*, \boldsymbol{f}(O, \boldsymbol{\omega}) - \bar{\boldsymbol{f}}(\boldsymbol{\omega}, \boldsymbol{\theta}_2) \rangle \right|}_{I_2}.
\end{aligned}
$$

For term $I_1$, we have

$$
\begin{aligned}
I_1 &= \left| \langle \boldsymbol{\omega} - \boldsymbol{\omega}_1^*, \bar{\boldsymbol{f}}(\boldsymbol{\omega}, \boldsymbol{\theta}_2) - \bar{\boldsymbol{f}}(\boldsymbol{\omega}, \boldsymbol{\theta}_1) \rangle \right| \\
&\le 2\bar{\omega} \| \bar{\boldsymbol{f}}(\boldsymbol{\omega}, \boldsymbol{\theta}_2) - \bar{\boldsymbol{f}}(\boldsymbol{\omega}, \boldsymbol{\theta}_1) \| \\
&\le 4\bar{\omega}\bar{\delta} d_{TV}(\mu_{\boldsymbol{\theta}_1} \otimes \pi_{\boldsymbol{\theta}_1} \otimes \mathcal{P}, \mu_{\boldsymbol{\theta}_2} \otimes \pi_{\boldsymbol{\theta}_2} \otimes \mathcal{P}) \\
&\overset{(1)}{\le} 8\bar{\omega}\bar{\delta}L_\pi(1 + \lceil \log_\rho m^{-1} \rceil + \frac{1}{1-\rho})\|\boldsymbol{\theta}_1 - \boldsymbol{\theta}_2\|,
\end{aligned}
$$

where (1) comes from Lemma B.1.

For term $I_2$, we have

$$
\begin{aligned}
I_2 &= \left| \langle \boldsymbol{\omega}_2^* - \boldsymbol{\omega}_1^*, \boldsymbol{f}(O, \boldsymbol{\omega}) - \bar{\boldsymbol{f}}(\boldsymbol{\omega}, \boldsymbol{\theta}_2) \rangle \right| \\
&\le 2\bar{\delta} \| \boldsymbol{\omega}_1^* - \boldsymbol{\omega}_2^* \| \\
&\overset{(1)}{\le} 2\bar{\delta}L_c \| \boldsymbol{\theta}_1 - \boldsymbol{\theta}_2 \|,
\end{aligned}
$$

where (1) follows from Lemma B.6. Combining $I_1$ and $I_2$, we have

$$\Lambda(O, \boldsymbol{\omega}, \boldsymbol{\theta}_1) - \Lambda(O, \boldsymbol{\omega}, \boldsymbol{\theta}_2) \le (8\bar{\omega}\bar{\delta}L_\pi(1 + \lceil \log_\rho m^{-1} \rceil + \frac{1}{1-\rho}) + 2\bar{\delta}L_c)\|\boldsymbol{\theta}_1 - \boldsymbol{\theta}_2\|.$$

**Step 2.** show that for any $\boldsymbol{\theta}, \boldsymbol{\omega}_1, \boldsymbol{\omega}_2$, and tuple $O(s, a, s')$, we have

$$\Lambda(O, \boldsymbol{\omega}_1, \boldsymbol{\theta}) - \Lambda(O, \boldsymbol{\omega}_2, \boldsymbol{\theta}) \le (8\bar{\omega} + 2\bar{\delta})\|\boldsymbol{\omega}_1 - \boldsymbol{\omega}_2\|. \tag{37}$$

According to the definition, we have

$$
\begin{aligned}
\Lambda(O, \boldsymbol{\omega}_1, \boldsymbol{\theta}) - \Lambda(O, \boldsymbol{\omega}_2, \boldsymbol{\theta}) =& \langle \boldsymbol{\omega}_1 - \boldsymbol{\omega}^*, \boldsymbol{f}(O, \boldsymbol{\omega}_1) - \bar{\boldsymbol{f}}(\boldsymbol{\omega}_1, \boldsymbol{\theta}) \rangle - \mathbb{E}\langle \boldsymbol{\omega}_2 - \boldsymbol{\omega}^*, \boldsymbol{f}(O, \boldsymbol{\omega}_2) - \bar{\boldsymbol{f}}(\boldsymbol{\omega}_2, \boldsymbol{\theta}) \rangle \\
\le& \underbrace{\left| \langle \boldsymbol{\omega}_1 - \boldsymbol{\omega}^*, \boldsymbol{f}(O, \boldsymbol{\omega}_1) - \bar{\boldsymbol{f}}(\boldsymbol{\omega}_1, \boldsymbol{\theta}) \rangle - \langle \boldsymbol{\omega}_1 - \boldsymbol{\omega}^*, \boldsymbol{f}(O, \boldsymbol{\omega}_2) - \bar{\boldsymbol{f}}(\boldsymbol{\omega}_2, \boldsymbol{\theta}) \rangle \right|}_{I_1} \\
&+ \underbrace{\left| \langle \boldsymbol{\omega}_1 - \boldsymbol{\omega}^*, \boldsymbol{f}(O, \boldsymbol{\omega}_2) - \bar{\boldsymbol{f}}(\boldsymbol{\omega}_2, \boldsymbol{\theta}) \rangle - \langle \boldsymbol{\omega}_2 - \boldsymbol{\omega}^*, \boldsymbol{f}(O, \boldsymbol{\omega}_2) - \bar{\boldsymbol{f}}(\boldsymbol{\omega}_2, \boldsymbol{\theta}) \rangle \right|}_{I_2}.
\end{aligned}
$$

For term $I_1$, we have

$$
\begin{aligned}
I_1 &= \left| \langle \boldsymbol{\omega}_1 - \boldsymbol{\omega}^*, \boldsymbol{f}(O, \boldsymbol{\omega}_1) - \bar{\boldsymbol{f}}(\boldsymbol{\omega}_1, \boldsymbol{\theta}) - (\boldsymbol{f}(O, \boldsymbol{\omega}_2) - \bar{\boldsymbol{f}}(\boldsymbol{\omega}_2, \boldsymbol{\theta})) \rangle \right| \\
&\le 2\bar{\omega}(\| \boldsymbol{f}(O, \boldsymbol{\omega}_1) - \boldsymbol{f}(O, \boldsymbol{\omega}_2) \| + \| \bar{\boldsymbol{f}}(\boldsymbol{\omega}_2, \boldsymbol{\theta}) - \bar{\boldsymbol{f}}(\boldsymbol{\omega}_1, \boldsymbol{\theta}) \|) \\
&\le 8\bar{\omega} \| \boldsymbol{\omega}_1 - \boldsymbol{\omega}_2 \|.
\end{aligned}
$$

For term $I_2$, we have

$$I_2 = \left|\langle \boldsymbol{\omega}_2 - \boldsymbol{\omega}_1, \boldsymbol{f}(O, \boldsymbol{\omega}_2) - \bar{\boldsymbol{f}}(\boldsymbol{\omega}_2, \boldsymbol{\theta})\rangle\right| \leq 2\bar{\delta}\|\boldsymbol{\omega}_1 - \boldsymbol{\omega}_2\|.$$

Combining $I_1$ and $I_2$, we have

$$\Lambda(O, \boldsymbol{\omega}_1, \boldsymbol{\theta}) - \Lambda(O, \boldsymbol{\omega}_2, \boldsymbol{\theta}) \leq (8\bar{\omega} + 2\bar{\delta})\|\boldsymbol{\omega}_1 - \boldsymbol{\omega}_2\|.$$

**Step 3:** show that for tuples $O_t = (s_t, a_t, s_{t+1})$ and $\widetilde{O}_t = (\widetilde{s}_t, \widetilde{a}_t, \widetilde{s}_{t+1})$. Conditioning on $\mathcal{F}_{t-\tau}$, we have

$$\mathbb{E}\big[\Lambda(O_t, \boldsymbol{\omega}_{t-\tau}, \boldsymbol{\theta}_{t-\tau}) - \Lambda(\widetilde{O}_t, \boldsymbol{\omega}_{t-\tau}, \boldsymbol{\theta}_{t-\tau})\big] \leq 4\bar{\omega}\bar{\delta}L_\pi \sum_{k=t-\tau}^{t} \mathbb{E}\|\boldsymbol{\theta}_k - \boldsymbol{\theta}_{t-\tau}\|. \tag{38}$$

By the definition of total variation distance, we have

$$
\begin{aligned}
\mathbb{E}\big[\Lambda(O_t, \boldsymbol{\omega}_{t-\tau}, \boldsymbol{\theta}_{t-\tau}) - \Lambda(\widetilde{O}_t, \boldsymbol{\omega}_{t-\tau}, \boldsymbol{\theta}_{t-\tau})\big] &\leq \mathbb{E}\langle \boldsymbol{\omega}_{t-\tau} - \boldsymbol{\omega}_{t-\tau}^*, \boldsymbol{f}(O_t, \boldsymbol{\omega}_{t-\tau}) - \boldsymbol{f}(\widetilde{O}_t, \boldsymbol{\omega}_{t-\tau})\rangle \\
&\leq 4\bar{\omega}\bar{\delta}d_{TV}(\mathbb{P}(O_t \in \cdot|\mathcal{F}_{t-\tau}), \mathbb{P}(\widetilde{O}_t \in \cdot|\mathcal{F}_{t-\tau})).
\end{aligned}
\tag{39}
$$

By Lemma B.2, we get

$$
\begin{aligned}
&d_{TV}(\mathbb{P}(O_t \in \cdot|\mathcal{F}_{t-\tau}), \mathbb{P}(\widetilde{O}_t \in \cdot|\mathcal{F}_{t-\tau})) \\
&= d_{TV}(\mathbb{P}((s_t, a_t) \in \cdot|\mathcal{F}_{t-\tau}), \mathbb{P}((\widetilde{s}_t, \widetilde{a}_t) \in \cdot|\mathcal{F}_{t-\tau})) \\
&\leq d_{TV}(\mathbb{P}(s_t \in \cdot|\mathcal{F}_{t-\tau}), \mathbb{P}(\widetilde{s}_t \in \cdot|\mathcal{F}_{t-\tau})) + L_\pi\mathbb{E}\|\boldsymbol{\theta}_t - \boldsymbol{\theta}_{t-\tau}\| \\
&\leq d_{TV}(\mathbb{P}(O_{t-1} \in \cdot|\mathcal{F}_{t-\tau}), \mathbb{P}(\widetilde{O}_{t-1} \in \cdot|\mathcal{F}_{t-\tau})) + L_\pi\mathbb{E}\|\boldsymbol{\theta}_t - \boldsymbol{\theta}_{t-\tau}\|.
\end{aligned}
$$

Repeat the above argument from $t$ to $t - \tau$, we have

$$d_{TV}(\mathbb{P}(O_t \in \cdot|\mathcal{F}_{t-\tau}), \mathbb{P}(\widetilde{O}_t \in \cdot|\mathcal{F}_{t-\tau})) \leq L_\pi \sum_{k=t-\tau}^{t} \mathbb{E}\|\boldsymbol{\theta}_k - \boldsymbol{\theta}_{t-\tau}\|. \tag{40}$$

Plugging Eq. (40) into Eq. (39), we get

$$\mathbb{E}\big[\Lambda(O_t, \boldsymbol{\omega}_{t-\tau}, \boldsymbol{\theta}_{t-\tau}) - \Lambda(\widetilde{O}_t, \boldsymbol{\omega}_{t-\tau}, \boldsymbol{\theta}_{t-\tau})\big] \leq 4\bar{\omega}\bar{\delta}L_\pi \sum_{k=t-\tau}^{t} \mathbb{E}\|\boldsymbol{\theta}_k - \boldsymbol{\theta}_{t-\tau}\|.$$

**Step 4:** show that conditioning on $\mathcal{F}_{t-\tau}$, we have

$$\mathbb{E}\big[\Lambda(\widetilde{O}_t, \boldsymbol{\omega}_{t-\tau}, \boldsymbol{\theta}_{t-\tau})\big] \leq 4\bar{\omega}\bar{\delta}m\rho^{\tau-1}. \tag{41}$$

It can be shown that

$$\mathbb{E}\big[\Lambda(O'_{t-\tau}, \boldsymbol{\omega}_{t-\tau}, \boldsymbol{\theta}_{t-\tau})|\mathcal{F}_{t-\tau}\big] = 0.$$

Then we have

$$
\begin{aligned}
\mathbb{E}\big[\Lambda(\widetilde{O}_t, \boldsymbol{\omega}_{t-\tau}, \boldsymbol{\theta}_{t-\tau})\big] &= \mathbb{E}\big[\Lambda(\widetilde{O}_t, \boldsymbol{\omega}_{t-\tau}, \boldsymbol{\theta}_{t-\tau}) - \Lambda(O'_{t-\tau}, \boldsymbol{\omega}_{t-\tau}, \boldsymbol{\theta}_{t-\tau})\big] \\
&= \mathbb{E}\big[\langle \boldsymbol{\omega}_{t-\tau} - \boldsymbol{\omega}_{t-\tau}^*, \boldsymbol{f}(\widetilde{O}_t, \boldsymbol{\omega}_{t-\tau}) - \boldsymbol{f}(O'_{t-\tau}, \boldsymbol{\omega}_{t-\tau})\rangle\big] \\
&\leq 4\bar{\omega}\bar{\delta}d_{TV}(\mathbb{P}(\widetilde{O}_t = \cdot|\mathcal{F}_{t-\tau}), \mu_{\boldsymbol{\theta}_{t-\tau}} \otimes \pi_{\boldsymbol{\theta}_{t-\tau}} \otimes \mathcal{P}) \\
&\overset{(1)}{\leq} 4\bar{\omega}\bar{\delta}m\rho^{\tau-1},
\end{aligned}
$$

where (1) follows from Assumption 4.2.

**Step 5:** show that for $t \geq \tau_{\text{mix}}$, we have

$$\mathbb{E}\big[\Lambda(O_t, \boldsymbol{\omega}_t, \boldsymbol{\theta}_t)\big] \leq M_1 \frac{1}{\sqrt{T}},$$

where $M_1 = 2\bar{\delta}L_\pi(1 + \lceil \log_\rho m^{-1} \rceil + (1-\rho)^{-1})\bar{\delta}B\tau_{\text{mix}}c + 4\bar{\delta}\tau_{\text{mix}} + \bar{\delta}L_\pi\bar{\delta}B\tau_{\text{mix}}^2 c + 2\bar{\delta}$.

Combining Eq. (36), Eq. (37), Eq. (38), and Eq. (41), we have

$$
\begin{aligned}
\mathbb{E}\big[\Lambda(O_t, \boldsymbol{\omega}_t, \boldsymbol{\theta}_t)\big] =& \mathbb{E}\big[\Lambda(O_t, \boldsymbol{\omega}_t, \boldsymbol{\theta}_t) - \Lambda(O_t, \boldsymbol{\omega}_t, \boldsymbol{\theta}_{t-\tau})\big] + \mathbb{E}\big[\Lambda(O_t, \boldsymbol{\omega}_t, \boldsymbol{\theta}_{t-\tau}) - \Lambda(O_t, \boldsymbol{\omega}_{t-\tau}, \boldsymbol{\theta}_{t-\tau})\big] \\
& + \mathbb{E}\big[\Lambda(O_t, \boldsymbol{\omega}_{t-\tau}, \boldsymbol{\theta}_{t-\tau}) - \Lambda(\widetilde{O}_t, \boldsymbol{\omega}_{t-\tau}, \boldsymbol{\theta}_{t-\tau})\big] + \mathbb{E}\big[\Lambda(\widetilde{O}_t, \boldsymbol{\omega}_{t-\tau}, \boldsymbol{\theta}_{t-\tau})\big] \\
\leq& (8\bar{\omega}\bar{\delta}L_\pi(1 + \lceil \log_\rho m^{-1} \rceil + \frac{1}{1-\rho}) + 2\bar{\delta}L_c)\|\boldsymbol{\theta}_t - \boldsymbol{\theta}_{t-\tau}\| + (8\bar{\omega} + 2\bar{\delta})\|\boldsymbol{\omega}_t - \boldsymbol{\omega}_{t-\tau}\| \\
& + 4\bar{\omega}\bar{\delta}L_\pi \sum_{k=t-\tau}^{t} \mathbb{E}\|\boldsymbol{\theta}_k - \boldsymbol{\theta}_{t-\tau}\| + 4\bar{\omega}\bar{\delta}m\rho^{\tau-1} \\
\overset{(1)}{\leq}& (8\bar{\omega}\bar{\delta}L_\pi(1 + \lceil \log_\rho m^{-1} \rceil + \frac{1}{1-\rho}) + 2\bar{\delta}L_c) \sum_{k=t-\tau}^{t-1} \alpha\bar{\delta}B + (8\bar{\omega} + 2\bar{\delta}) \sum_{k=t-\tau}^{t-1} \beta\bar{\delta} \\
& + 4\bar{\omega}\bar{\delta}L_\pi \sum_{k=t-\tau}^{t} \sum_{i=t-\tau}^{k-1} \alpha\bar{\delta}B + 4\bar{\omega}\bar{\delta}m\rho^{\tau-1} \\
\leq& (8\bar{\omega}\bar{\delta}L_\pi(1 + \lceil \log_\rho m^{-1} \rceil + \frac{1}{1-\rho}) + 2\bar{\delta}L_c)\tau\bar{\delta}B\frac{c}{\sqrt{T}} + (8\bar{\omega} + 2\bar{\delta})\tau\bar{\delta}\frac{1}{\sqrt{T}} \\
& + 4\bar{\omega}\bar{\delta}L_\pi\tau^2\bar{\delta}B\frac{c}{\sqrt{T}} + 4\bar{\omega}\bar{\delta}m\rho^{\tau-1} \\
\overset{(2)}{\leq}& ((8\bar{\omega}\bar{\delta}L_\pi(1 + \lceil \log_\rho m^{-1} \rceil + \frac{1}{1-\rho}) + 2\bar{\delta}L_c)\bar{\delta}B\tau_{\text{mix}}c + (8\bar{\omega} + 2\bar{\delta})\bar{\delta}\tau_{\text{mix}} + 4\bar{\omega}L_\pi B\bar{\delta}^2\tau_{\text{mix}}^2 c + 4\bar{\omega}\bar{\delta})\frac{1}{\sqrt{T}},
\end{aligned}
$$

where (1) comes from the update rule of the critic and the actor, (2) is followed by choosing $\tau = \tau_{\text{mix}}$. Therefore, we conclude our proof. $\qquad\square$

**Proof of Lemma C.2**

*Proof.* We will divide the proof of this lemma into five steps.

**Step 1:** show that for any $O, \boldsymbol{\omega}, \boldsymbol{\theta}_1, \boldsymbol{\theta}_2$, we have

$$\|\Gamma(O, \boldsymbol{\omega}, \boldsymbol{\theta}_1) - \Gamma(O, \boldsymbol{\omega}, \boldsymbol{\theta}_2)\| \leq (2\bar{\delta}BL_c^2 + 4\bar{\delta}\bar{\omega}BL_s + 4\bar{\omega}L_cL_{\bar{h}})\|\boldsymbol{\theta}_1 - \boldsymbol{\theta}_2\|, \tag{42}$$

where $L_{\bar{h}} = \bar{\delta}L_l + 2BL_c + 4\bar{\delta}BL_\pi(1 + (1-\gamma)^{-1})$.

Since $\Gamma(O, \boldsymbol{\omega}, \boldsymbol{\theta}) = \langle \boldsymbol{\omega} - \boldsymbol{\omega}^*, (\nabla\boldsymbol{\omega}^*)^\top(\bar{\boldsymbol{h}}(\boldsymbol{\omega}^*, \boldsymbol{\theta}) - h(O, \boldsymbol{\omega}^*, \boldsymbol{\theta}))\rangle$, we represent $\bar{\boldsymbol{h}}(\boldsymbol{\omega}^*, \boldsymbol{\theta}) = \mathbb{E}_{\boldsymbol{\theta}}[h(O, \boldsymbol{\omega}^*, \boldsymbol{\theta})]$, where $\mathbb{E}_{\boldsymbol{\theta}}$ is the shorthand of $\mathbb{E}_{O \sim (\nu_{\boldsymbol{\theta}}, \pi_{\boldsymbol{\theta}}, \mathcal{P})}$. In the following, we will show that each term in $\Gamma(O, \boldsymbol{\omega}, \boldsymbol{\theta})$ is Lipschitz with respect to $\boldsymbol{\theta}$.

Term $\boldsymbol{\omega}$ is not related to $\boldsymbol{\theta}$, term $\boldsymbol{\omega}^*(\boldsymbol{\theta})$ is $L_c$-Lipschitz according to Lemma B.6, and term $\nabla\boldsymbol{\omega}^*(\boldsymbol{\theta})$ is $L_s$-Lipschitz according to Lemma B.9.

For term $\boldsymbol{h}(O, \boldsymbol{\omega}^*, \boldsymbol{\theta})$, we have

$$
\begin{aligned}
\|\boldsymbol{h}(O, \boldsymbol{\omega}_1^*, \boldsymbol{\theta}_1) - \boldsymbol{h}(O, \boldsymbol{\omega}_2^*, \boldsymbol{\theta}_2)\| &\leq \|\boldsymbol{h}(O, \boldsymbol{\omega}_1^*, \boldsymbol{\theta}_1) - \boldsymbol{h}(O, \boldsymbol{\omega}_1^*, \boldsymbol{\theta}_2)\| + \|\boldsymbol{h}(O, \boldsymbol{\omega}_1^*, \boldsymbol{\theta}_2) - \boldsymbol{h}(O, \boldsymbol{\omega}_2^*, \boldsymbol{\theta}_2)\| \\
&\leq \bar{\delta}\|\nabla\log\pi_{\boldsymbol{\theta}_1}(a \,|\, s) - \nabla\log\pi_{\boldsymbol{\theta}_2}(a \,|\, s)\| + 2B\|\boldsymbol{\omega}_1^* - \boldsymbol{\omega}_2^*\| \\
&\leq (\bar{\delta}L_l + 2BL_c)\|\boldsymbol{\theta}_1 - \boldsymbol{\theta}_2\|.
\end{aligned}
$$

Hence we have $\boldsymbol{h}(O, \boldsymbol{\omega}^*, \boldsymbol{\theta})$ is $L_h$-Lipschitz, where $L_h = \bar{\delta}L_l + 2BL_c$.

For term $\mathbb{E}_{\boldsymbol{\theta}}[\boldsymbol{h}(O, \boldsymbol{\omega}^*, \boldsymbol{\theta})]$, we have

$$
\begin{aligned}
& \|\mathbb{E}_{\boldsymbol{\theta}_1}[\boldsymbol{h}(O, \boldsymbol{\omega}_1^*, \boldsymbol{\theta}_1)] - \mathbb{E}_{\boldsymbol{\theta}_2}[\boldsymbol{h}(O, \boldsymbol{\omega}_2^*, \boldsymbol{\theta}_2)]\| \\
& \leq \|\mathbb{E}_{\boldsymbol{\theta}_1}[\boldsymbol{h}(O, \boldsymbol{\omega}_1^*, \boldsymbol{\theta}_1)] - \mathbb{E}_{\boldsymbol{\theta}_1}[\boldsymbol{h}(O, \boldsymbol{\omega}_2^*, \boldsymbol{\theta}_2)]\| + \|\mathbb{E}_{\boldsymbol{\theta}_1}[\boldsymbol{h}(O, \boldsymbol{\omega}_2^*, \boldsymbol{\theta}_2)] - \mathbb{E}_{\boldsymbol{\theta}_2}[\boldsymbol{h}(O, \boldsymbol{\omega}_2^*, \boldsymbol{\theta}_2)]\| \\
& \leq \mathbb{E}_{\boldsymbol{\theta}_1}\|\boldsymbol{h}(O, \boldsymbol{\omega}_1^*, \boldsymbol{\theta}_1) - \boldsymbol{h}(O, \boldsymbol{\omega}_2^*, \boldsymbol{\theta}_2)\| + \|\mathbb{E}_{\boldsymbol{\theta}_1}[\boldsymbol{h}(O, \boldsymbol{\omega}_2^*, \boldsymbol{\theta}_2)] - \mathbb{E}_{\boldsymbol{\theta}_2}[\boldsymbol{h}(O, \boldsymbol{\omega}_2^*, \boldsymbol{\theta}_2)]\| \\
& \leq L_h\|\boldsymbol{\theta}_1 - \boldsymbol{\theta}_2\| + \|\mathbb{E}_{\boldsymbol{\theta}_1}[\boldsymbol{h}(O, \boldsymbol{\omega}_2^*, \boldsymbol{\theta}_2)] - \mathbb{E}_{\boldsymbol{\theta}_2}[h(O, \boldsymbol{\omega}_2^*, \boldsymbol{\theta}_2)]\| \\
& \leq L_h\|\boldsymbol{\theta}_1 - \boldsymbol{\theta}_2\| + 2\bar{\delta}B d_{TV}(\nu_{\boldsymbol{\theta}_1} \otimes \pi_{\boldsymbol{\theta}_1} \otimes P, \nu_{\boldsymbol{\theta}_2} \otimes \pi_{\boldsymbol{\theta}_2} \otimes P) \\
& \overset{(1)}{\leq} (L_h + 4\bar{\delta}BL_\pi(1 + \frac{1}{1-\gamma}))\|\boldsymbol{\theta}_1 - \boldsymbol{\theta}_2\| \\
& = L_{\bar{h}}\|\boldsymbol{\theta}_1 - \boldsymbol{\theta}_2\|,
\end{aligned}
$$

where (1) follows from Lemma B.3 and $L_{\bar{h}} = \bar{\delta}L_l + 2BL_c + 4\bar{\delta}BL_\pi(1 + (1-\gamma)^{-1})$.

Then we have $\boldsymbol{\omega} - \boldsymbol{\omega}_{\boldsymbol{\theta}}^*$ is $2\bar{\omega}$-bounded and $L_c$-Lipschitz; $\nabla\boldsymbol{\omega}_{\boldsymbol{\theta}}^*$ is $L_c$-bounded and $L_s$-Lipschitz; $\mathbb{E}_{\boldsymbol{\theta}}[\boldsymbol{h}(O, \boldsymbol{\omega}^*, \boldsymbol{\theta})] - \boldsymbol{h}(O, \boldsymbol{\omega}^*, \boldsymbol{\theta})$ is $2\bar{\delta}B$-bounded and $2L_{\bar{h}}$-Lipschitz. By the triangle inequality, we have

$$
\|\Gamma(O, \boldsymbol{\omega}, \boldsymbol{\theta}_1) - \Gamma(O, \boldsymbol{\omega}, \boldsymbol{\theta}_2)\| \leq (2\bar{\delta}BL_c^2 + 4\bar{\delta}\bar{\omega}BL_s + 4\bar{\omega}L_cL_{\bar{h}})\|\boldsymbol{\theta}_1 - \boldsymbol{\theta}_2\|.
$$

**Step 2:** show that for any $O, \boldsymbol{\omega}_1, \boldsymbol{\omega}_2, \boldsymbol{\theta}$, we have

$$
\|\Gamma(O, \boldsymbol{\omega}_1, \boldsymbol{\theta}) - \Gamma(O, \boldsymbol{\omega}_2, \boldsymbol{\theta})\| \leq 2\bar{\delta}BL_c\|\boldsymbol{\omega}_1 - \boldsymbol{\omega}_2\|. \tag{43}
$$

It can be shown that

$$
\|\Gamma(O, \boldsymbol{\omega}_1, \boldsymbol{\theta}) - \Gamma(O, \boldsymbol{\omega}_2, \boldsymbol{\theta})\| = \langle \boldsymbol{\omega}_1 - \boldsymbol{\omega}_2, (\nabla\boldsymbol{\omega}^*)^\top(\bar{\boldsymbol{h}}(\boldsymbol{\omega}^*, \boldsymbol{\theta}) - \boldsymbol{h}(O, \boldsymbol{\omega}^*, \boldsymbol{\theta}))\rangle \leq 2\bar{\delta}BL_c\|\boldsymbol{\omega}_1 - \boldsymbol{\omega}_2\|.
$$

**Step 3:** show that for tuples $\widehat{O}_t = (\hat{s}_t, \hat{a}_t, \hat{s}_{t+1})$ and $\bar{O}_t = (\bar{s}_t, \bar{a}_t, \bar{s}_{t+1})$. Conditioning on $\mathcal{F}_{t-\tau}$, we have

$$
\|\mathbb{E}[\Gamma(\widehat{O}_t, \boldsymbol{\omega}_{t-\tau}, \boldsymbol{\theta}_{t-\tau}) - \Gamma(\bar{O}_t, \boldsymbol{\omega}_{t-\tau}, \boldsymbol{\theta}_{t-\tau})]\| \leq 4\bar{\omega}\bar{\delta}BL_cL_\pi \sum_{k=t-\tau}^t \mathbb{E}\|\boldsymbol{\theta}_k - \boldsymbol{\theta}_{t-\tau}\|. \tag{44}
$$

By definition of $\Gamma(O, \boldsymbol{\omega}, \boldsymbol{\theta})$ in Eq. (25), we have

$$
\begin{aligned}
& \|\mathbb{E}[\Gamma(\widehat{O}_t, \boldsymbol{\omega}_{t-\tau}, \boldsymbol{\theta}_{t-\tau}) - \Gamma(\bar{O}_t, \boldsymbol{\omega}_{t-\tau}, \boldsymbol{\theta}_{t-\tau})]\| \\
& = \|\mathbb{E}[\langle \boldsymbol{\omega}_{t-\tau} - \boldsymbol{\omega}_{t-\tau}^*, (\nabla\boldsymbol{\omega}_{t-\tau}^*)^\top(\boldsymbol{h}(\widehat{O}_t, \boldsymbol{\omega}_{t-\tau}^*, \boldsymbol{\theta}_{t-\tau}) - \boldsymbol{h}(\bar{O}_t, \boldsymbol{\omega}_{t-\tau}^*, \boldsymbol{\theta}_{t-\tau}))\rangle]\| \\
& \leq 4\bar{\omega}\bar{\delta}BL_c d_{TV}(\mathbb{P}(\widehat{O}_t \in \cdot|\mathcal{F}_{t-\tau}), \mathbb{P}(\bar{O}_t \in \cdot|\mathcal{F}_{t-\tau})),
\end{aligned} \tag{45}
$$

where the inequality comes from the definition of total variation distance.

By Lemma B.4, we get

$$
\begin{aligned}
& d_{TV}(\mathbb{P}(\widehat{O}_t \in \cdot|\mathcal{F}_{t-\tau}), \mathbb{P}(\bar{O}_t \in \cdot|\mathcal{F}_{t-\tau})) \\
& = d_{TV}(\mathbb{P}((\hat{s}_t, \hat{a}_t) \in \cdot|\mathcal{F}_{t-\tau}), \mathbb{P}((\bar{s}_t, \bar{a}_t) \in \cdot|\mathcal{F}_{t-\tau})) \\
& \leq d_{TV}(\mathbb{P}(\hat{s}_t \in \cdot|\mathcal{F}_{t-\tau}), \mathbb{P}(\bar{s}_t \in \cdot|\mathcal{F}_{t-\tau})) + L_\pi \mathbb{E}\|\boldsymbol{\theta}_t - \boldsymbol{\theta}_{t-\tau}\| \\
& \leq d_{TV}(\mathbb{P}(\widehat{O}_{t-1} \in \cdot|\mathcal{F}_{t-\tau}), \mathbb{P}(\bar{O}_{t-1} \in \cdot|\mathcal{F}_{t-\tau})) + L_\pi \mathbb{E}\|\boldsymbol{\theta}_t - \boldsymbol{\theta}_{t-\tau}\|.
\end{aligned}
$$

Repeat the above argument from $t$ to $t - \tau$, we have

$$
d_{TV}(\mathbb{P}(\widehat{O}_t \in \cdot|\mathcal{F}_{t-\tau}), \mathbb{P}(\bar{O}_t \in \cdot|\mathcal{F}_{t-\tau})) \leq L_\pi \sum_{k=t-\tau}^t \mathbb{E}\|\boldsymbol{\theta}_k - \boldsymbol{\theta}_{t-\tau}\|. \tag{46}
$$

Plugging Eq. (46) into Eq. (45), we have

$$
\|\mathbb{E}[\Gamma(\widehat{O}_t, \boldsymbol{\omega}_{t-\tau}, \boldsymbol{\theta}_{t-\tau}) - \Gamma(\bar{O}_t, \boldsymbol{\omega}_{t-\tau}, \boldsymbol{\theta}_{t-\tau})]\| \leq 4\bar{\omega}\bar{\delta}BL_cL_\pi \sum_{k=t-\tau}^t \mathbb{E}\|\boldsymbol{\theta}_k - \boldsymbol{\theta}_{t-\tau}\|.
$$

**Step 4:** show that conditioning on $\mathcal{F}_{t-\tau}$, we have

$$\|\mathbb{E}[\Gamma(\bar{O}_t, \boldsymbol{\omega}_{t-\tau}, \boldsymbol{\theta}_{t-\tau})]\| \leq 4\bar{\omega}\bar{\delta}BL_c\gamma^{\tau-1}. \tag{47}$$

It can be shown that

$$\|\mathbb{E}[\Gamma(\bar{O}_t, \boldsymbol{\omega}_{t-\tau}, \boldsymbol{\theta}_{t-\tau})]\| \overset{(1)}{=} \|\mathbb{E}[\Gamma(\bar{O}_t, \boldsymbol{\omega}_{t-\tau}, \boldsymbol{\theta}_{t-\tau}) - \Gamma(O''_{t-\tau}, \boldsymbol{\omega}_{t-\tau}, \boldsymbol{\theta}_{t-\tau})]\|$$
$$\overset{(2)}{\leq} 4\bar{\omega}\bar{\delta}BL_c d_{TV}(\mathbb{P}(\bar{O}_t \in \cdot|\mathcal{F}_{t-\tau}), \nu_{\boldsymbol{\theta}_{t-\tau}} \otimes \pi_{\boldsymbol{\theta}_{t-\tau}} \otimes P),$$

where (1) is due to the fact that $O''_{t-\tau}$ is from the discounted state visitation distribution which satisfies $\mathbb{E}[\Gamma(O''_{t-\tau}, \boldsymbol{\omega}_{t-\tau}, \boldsymbol{\theta}_{t-\tau})|\mathcal{F}_{t-\tau}] = 0$ and (2) follows from the definition of total variation distance. From Proposition 3.2, we know that

$$d_{TV}(\mathbb{P}(\bar{s}_t \in \cdot), \nu_{\boldsymbol{\theta}_{t-\tau}}) \leq \gamma^{\tau-1}.$$

Therefore, we have

$$\|\mathbb{E}[\Gamma(\bar{O}_t, \boldsymbol{\omega}_{t-\tau}, \boldsymbol{\theta}_{t-\tau})]\| \leq 4\bar{\omega}\bar{\delta}BL_c d_{TV}(\mathbb{P}(\bar{O}_t \in \cdot|\mathcal{F}_{t-\tau}), \nu_{\boldsymbol{\theta}_{t-\tau}} \otimes \pi_{\boldsymbol{\theta}_{t-\tau}} \otimes P)$$
$$= 4\bar{\omega}\bar{\delta}BL_c d_{TV}(\mathbb{P}((\bar{s}_t, \bar{a}_t) \in \cdot|\mathcal{F}_{t-\tau}, \mu_{\boldsymbol{\theta}_{t-\tau}} \otimes \pi_{\boldsymbol{\theta}_{t-\tau}})$$
$$= 4\bar{\omega}\bar{\delta}BL_c d_{TV}(\mathbb{P}(\bar{s}_t \in \cdot|\mathcal{F}_{t-\tau}), \mu_{\boldsymbol{\theta}_{t-\tau}})$$
$$\leq 4\bar{\omega}\bar{\delta}BL_c\gamma^{\tau-1}.$$

**Step 5:** show that for $t \geq \tau_{\text{mix}}$, we have

$$\mathbb{E}[\Gamma(\widehat{O}_t, \boldsymbol{\omega}_t, \boldsymbol{\theta}_t)] \leq M_2 \frac{1}{\sqrt{T}},$$

where $M_2 = (2\bar{\delta}BL_c^2 + 4\bar{\delta}\bar{\omega}BL_s + 4\bar{\omega}L_cL_{\bar{h}})\bar{\delta}Bc\tau_{\text{mix}} + 2\bar{\delta}BL_c\bar{\delta}\tau_{\text{mix}} + 4\bar{\omega}\bar{\delta}BL_cL_\pi\bar{\delta}Bc\tau_{\text{mix}}^2 + 4\bar{\omega}\bar{\delta}BL_c$.

Combining Eq. (42), Eq. (43), Eq. (44), and Eq. (47), we have

$$\mathbb{E}[\Gamma(\widehat{O}_t, \boldsymbol{\omega}_t, \boldsymbol{\theta}_t)] = \mathbb{E}[\Gamma(\widehat{O}_t, \boldsymbol{\omega}_t, \boldsymbol{\theta}_t) - \Gamma(\widehat{O}_t, \boldsymbol{\omega}_t, \boldsymbol{\theta}_{t-\tau})] + \mathbb{E}[\Gamma(\widehat{O}_t, \boldsymbol{\omega}_t, \boldsymbol{\theta}_{t-\tau}) - \Gamma(\widehat{O}_t, \boldsymbol{\omega}_{t-\tau}, \boldsymbol{\theta}_{t-\tau})]$$
$$+ \mathbb{E}[\Gamma(\widehat{O}_t, \boldsymbol{\omega}_{t-\tau}, \boldsymbol{\theta}_{t-\tau}) - \Gamma(\bar{O}_t, \boldsymbol{\omega}_{t-\tau}, \boldsymbol{\theta}_{t-\tau})] + \mathbb{E}[\Gamma(\bar{O}_t, \boldsymbol{\omega}_{t-\tau}, \boldsymbol{\theta}_{t-\tau})]$$
$$\leq (2\bar{\delta}BL_c^2 + 4\bar{\delta}\bar{\omega}BL_s + 4\bar{\omega}L_cL_{\bar{h}})\|\boldsymbol{\theta}_t - \boldsymbol{\theta}_{t-\tau}\| + 2\bar{\delta}BL_c\|\boldsymbol{\omega}_t - \boldsymbol{\omega}_{t-\tau}\|$$
$$+ 4\bar{\omega}\bar{\delta}BL_cL_\pi \sum_{k=t-\tau}^{t} \mathbb{E}\|\boldsymbol{\theta}_k - \boldsymbol{\theta}_{t-\tau}\| + 4\bar{\omega}\bar{\delta}BL_c\gamma^{\tau-1}$$
$$\overset{(1)}{\leq} (2\bar{\delta}BL_c^2 + 4\bar{\delta}\bar{\omega}BL_s + 4\bar{\omega}L_cL_{\bar{h}}) \sum_{k=t-\tau}^{t-1} \alpha\bar{\delta}B + 2\bar{\delta}BL_c \sum_{k=t-\tau}^{t-1} \beta\bar{\delta}$$
$$+ 4\bar{\omega}\bar{\delta}BL_cL_\pi \sum_{k=t-\tau}^{t} \sum_{i=t-\tau}^{k-1} \alpha\bar{\delta}B + 4\bar{\omega}\bar{\delta}BL_c\gamma^{\tau-1}$$
$$\leq (2\bar{\delta}BL_c^2 + 4\bar{\delta}\bar{\omega}BL_s + 4\bar{\omega}L_cL_{\bar{h}})\tau\bar{\delta}B\frac{c}{\sqrt{T}} + 2\bar{\delta}BL_c\tau\bar{\delta}\frac{1}{\sqrt{T}}$$
$$+ 4\bar{\omega}\bar{\delta}BL_cL_\pi\tau^2\bar{\delta}B\frac{c}{\sqrt{T}} + 4\bar{\omega}\bar{\delta}BL_c\gamma^{\tau-1}$$
$$\overset{(2)}{\leq} ((2\bar{\delta}BL_c^2 + 4\bar{\delta}\bar{\omega}BL_s + 4\bar{\omega}L_cL_{\bar{h}})\bar{\delta}Bc\tau_{\text{mix}} + 2\bar{\delta}BL_c\bar{\delta}\tau_{\text{mix}} + 4\bar{\omega}\bar{\delta}BL_cL_\pi\bar{\delta}Bc\tau_{\text{mix}}^2 + 4\bar{\omega}\bar{\delta}BL_c)\frac{1}{\sqrt{T}},$$

where (1) comes from the update rule of the critic and the actor, (2) is followed by choosing $\tau = \tau_{\text{mix}}$. Thus we conclude our proof. □

**Proof of Lemma C.3**

*Proof.* We will divide the proof of this lemma into four steps.

**Step 1:** show that for any $O, \boldsymbol{\theta}_1, \boldsymbol{\theta}_2$, we have

$$\|\Xi(O, \boldsymbol{\omega}, \boldsymbol{\theta}_1) - \Xi(O, \boldsymbol{\omega}, \boldsymbol{\theta}_2)\| \leq (2\bar{\delta}BL_g + 2L_J L_{\bar{h}})\|\boldsymbol{\theta}_1 - \boldsymbol{\theta}_2\|. \tag{48}$$

Since $\Xi(O, \boldsymbol{\omega}, \boldsymbol{\theta}) = \langle \nabla J(\boldsymbol{\theta}), \mathbb{E}_{\boldsymbol{\theta}}[\boldsymbol{h}(O, \boldsymbol{\omega}, \boldsymbol{\theta})] - \boldsymbol{h}(O, \boldsymbol{\omega}, \boldsymbol{\theta}) \rangle$, we will show that each term in $\Xi(O, \boldsymbol{\omega}, \boldsymbol{\theta})$ is Lipschitz.

For the term $\nabla J(\boldsymbol{\theta})$, we know it's $L_J$-bounded and $L_g$-Lipschitz. For term $\mathbb{E}_{\boldsymbol{\theta}}[\boldsymbol{h}(O, \boldsymbol{\omega}, \boldsymbol{\theta})] - \boldsymbol{h}(O, \boldsymbol{\omega}, \boldsymbol{\theta})$, by the same argument shown in the proof of Lemma C.2, it's $2\bar{\delta}B$-bounded and $2L_{\bar{h}}$-Lipschitz. By the triangle inequality, we have

$$\|\Xi(O, \boldsymbol{\omega}, \boldsymbol{\theta}_1) - \Xi(O, \boldsymbol{\omega}, \boldsymbol{\theta}_2)\| \leq (2\bar{\delta}BL_g + 2L_J L_{\bar{h}})\|\boldsymbol{\theta}_1 - \boldsymbol{\theta}_2\|.$$

**Step 2:** show that for any $O, \boldsymbol{\omega}_1, \boldsymbol{\omega}_2, \boldsymbol{\theta}$, we have

$$\|\Xi(O, \boldsymbol{\omega}_1, \boldsymbol{\theta}) - \Xi(O, \boldsymbol{\omega}_2, \boldsymbol{\theta})\| \leq 4BL_J\|\boldsymbol{\omega}_1 - \boldsymbol{\omega}_2\|. \tag{49}$$

It follows that

$$\begin{aligned}
\|\Xi(O, \boldsymbol{\omega}_1, \boldsymbol{\theta}) - \Xi(O, \boldsymbol{\omega}_2, \boldsymbol{\theta})\| &= |\langle \nabla J(\boldsymbol{\theta}), \boldsymbol{h}(O, \boldsymbol{\omega}_1, \boldsymbol{\theta}) - \boldsymbol{h}(O, \boldsymbol{\omega}_2, \boldsymbol{\theta}) \rangle| \\
&\quad + |\langle \nabla J(\boldsymbol{\theta}), \mathbb{E}_{\boldsymbol{\theta}}[\boldsymbol{h}(O, \boldsymbol{\omega}_1, \boldsymbol{\theta})] - \mathbb{E}_{\boldsymbol{\theta}}[\boldsymbol{h}(O, \boldsymbol{\omega}_2, \boldsymbol{\theta})] \rangle| \\
&\leq 2BL_J\|\boldsymbol{\omega}_1 - \boldsymbol{\omega}_2\| + 2BL_J\|\boldsymbol{\omega}_1 - \boldsymbol{\omega}_2\| \\
&= 4BL_J\|\boldsymbol{\omega}_1 - \boldsymbol{\omega}_2\|.
\end{aligned}$$

**Step 3:** show that for tuples $\widehat{O}_t = (\hat{s}_t, \hat{a}_t, \hat{s}_{t+1})$ and $\bar{O}_t = (\bar{s}_t, \bar{a}_t, \bar{s}_{t+1})$. Conditioning on $\mathcal{F}_{t-\tau}$, we have

$$\|\Xi(\widehat{O}_t, \boldsymbol{\omega}_{t-\tau}, \boldsymbol{\theta}_{t-\tau}) - \Xi(\bar{O}_t, \boldsymbol{\omega}_{t-\tau}, \boldsymbol{\theta}_{t-\tau})\| \leq \bar{\delta}BL_J L_\pi \sum_{k=t-\tau}^{t} \mathbb{E}\|\boldsymbol{\theta}_k - \boldsymbol{\theta}_{t-\tau}\|. \tag{50}$$

By definition of $\Xi(O, \boldsymbol{\omega}, \boldsymbol{\theta})$, we have

$$\begin{aligned}
\|\mathbb{E}[\Xi(\widehat{O}_t, \boldsymbol{\omega}_{t-\tau}, \boldsymbol{\theta}_{t-\tau}) - \Xi(\bar{O}_t, \boldsymbol{\omega}_{t-\tau}, \boldsymbol{\theta}_{t-\tau})]\| &= \|\mathbb{E}[\langle \nabla J(\boldsymbol{\theta}_{t-\tau}), \boldsymbol{h}(\widehat{O}_t, \boldsymbol{\omega}_{t-\tau}, \boldsymbol{\theta}_{t-\tau}) - \boldsymbol{h}(\bar{O}_t, \boldsymbol{\omega}_{t-\tau}, \boldsymbol{\theta}_{t-\tau})]\| \\
&\leq 2\bar{\delta}BL_J d_{TV}(\mathbb{P}(\widehat{O}_t \in \cdot | \mathcal{F}_{t-\tau}), \mathbb{P}(\bar{O}_t \in \cdot | \mathcal{F}_{t-\tau})),
\end{aligned}$$

where the inequality comes from the definition of total variation distance. The total variation distance between $\widehat{O}_t$ and $\bar{O}_t$ has been computed in Eq. (46). Plugging Eq. (46) into the above inequality, we get

$$\|\mathbb{E}[\Xi(\widehat{O}_t, \boldsymbol{\omega}_{t-\tau}, \boldsymbol{\theta}_{t-\tau}) - \Xi(\bar{O}_t, \boldsymbol{\omega}_{t-\tau}, \boldsymbol{\theta}_{t-\tau})]\| \leq 2\bar{\delta}BL_J L_\pi \sum_{k=t-\tau}^{t} \mathbb{E}\|\boldsymbol{\theta}_k - \boldsymbol{\theta}_{t-\tau}\|.$$

**Step 4:** show that conditioning on $\mathcal{F}_{t-\tau}$, we have

$$\|\mathbb{E}[\Xi(\bar{O}_t, \boldsymbol{\omega}_{t-\tau}, \boldsymbol{\theta}_{t-\tau})]\| \leq 2\bar{\delta}BL_J \gamma^{\tau-1}. \tag{51}$$

It holds that

$$\begin{aligned}
\|\mathbb{E}[\Xi(\bar{O}_t, \boldsymbol{\omega}_{t-\tau}, \boldsymbol{\theta}_{t-\tau})]\| &\overset{(1)}{=} \|\mathbb{E}[\Xi(\bar{O}_t, \boldsymbol{\omega}_{t-\tau}, \boldsymbol{\theta}_{t-\tau}) - \Xi(O''_{t-\tau}, \boldsymbol{\omega}_{t-\tau}, \boldsymbol{\theta}_{t-\tau})]\| \\
&\overset{(2)}{\leq} 2\bar{\delta}BL_J d_{TV}(\mathbb{P}(\bar{O}_t \in \cdot | \mathcal{F}_{t-\tau}), \nu_{\boldsymbol{\theta}_{t-\tau}} \otimes \pi_{\boldsymbol{\theta}_{t-\tau}} \otimes P) \\
&= 2\bar{\delta}BL_J d_{TV}(\mathbb{P}((\bar{s}_t, \bar{a}_t) \in \cdot | \mathcal{F}_{t-\tau}), \nu_{\boldsymbol{\theta}_{t-\tau}} \otimes \pi_{\boldsymbol{\theta}_{t-\tau}}) \\
&= 2\bar{\delta}BL_J d_{TV}(\mathbb{P}(\bar{s}_t \in \cdot | \mathcal{F}_{t-\tau}), \nu_{\boldsymbol{\theta}_{t-\tau}}) \\
&\overset{(3)}{\leq} 2\bar{\delta}BL_J \gamma^{\tau-1},
\end{aligned}$$

where (1) is due to the fact that $O''_{t-\tau}$ is sampled from the discounted state visitation distribution which satisfies $\mathbb{E}[\Xi(O''_{t-\tau}, \boldsymbol{\omega}_{t-\tau}, \boldsymbol{\theta}_{t-\tau}) | \mathcal{F}_{t-\tau}] = 0$, (2) follows from the definition of total variation distance, and (3) comes from Proposition 3.2.

**Step 5:** show that for $t \geq \tau_{\mathrm{mix}}$, we have

$$\mathbb{E}\big[\Xi(\widehat{O}_t, \boldsymbol{\omega}_t, \boldsymbol{\theta}_t)\big] \leq M_3 \frac{1}{\sqrt{T}},$$

where $M_3 = (2\bar{\delta}BL_g + 2L_J L_{\bar{h}})\bar{\delta}Bc\tau_{\mathrm{mix}} + 4BL_J\bar{\delta}\tau_{\mathrm{mix}} + 2\bar{\delta}^2 B^2 L_J L_\pi c\tau_{\mathrm{mix}}^2 + 2\bar{\delta}BL_J$.

Combining Eq. (48), Eq. (49), Eq. (50), and Eq. (51), we can decompose the Markovian bias as

$$
\begin{aligned}
\mathbb{E}\big[\Xi(\widehat{O}_t, \boldsymbol{\omega}_t, \boldsymbol{\theta}_t)\big] &= \mathbb{E}\big[\Xi(\widehat{O}_t, \boldsymbol{\omega}_t, \boldsymbol{\theta}_t) - \Xi(\widehat{O}_t, \boldsymbol{\omega}_t, \boldsymbol{\theta}_{t-\tau})\big] + \mathbb{E}\big[\Xi(\widehat{O}_t, \boldsymbol{\omega}_t, \boldsymbol{\theta}_{t-\tau}) - \Xi(\widehat{O}_t, \boldsymbol{\omega}_{t-\tau}, \boldsymbol{\theta}_{t-\tau})\big] \\
&\quad + \mathbb{E}\big[\Xi(\widehat{O}_t, \boldsymbol{\omega}_{t-\tau}, \boldsymbol{\theta}_{t-\tau}) - \Xi(\bar{O}_t, \boldsymbol{\omega}_{t-\tau}, \boldsymbol{\theta}_{t-\tau})\big] + \mathbb{E}\big[\Xi(\bar{O}_t, \boldsymbol{\omega}_{t-\tau}, \boldsymbol{\theta}_{t-\tau})\big] \\
&\leq (2\bar{\delta}BL_g + 2L_J L_{\bar{h}})\|\boldsymbol{\theta}_t - \boldsymbol{\theta}_{t-\tau}\| + 4BL_J\|\boldsymbol{\omega}_1 - \boldsymbol{\omega}_2\| \\
&\quad + 2\bar{\delta}BL_J L_\pi \sum_{k=t-\tau}^{t} \mathbb{E}\|\boldsymbol{\theta}_k - \boldsymbol{\theta}_{t-\tau}\| + 2\bar{\delta}BL_J\gamma^{\tau-1} \\
&\overset{(1)}{\leq} (2\bar{\delta}BL_g + 2L_J L_{\bar{h}}) \sum_{k=t-\tau}^{t-1} \alpha\bar{\delta}B + 4BL_J \sum_{k=t-\tau}^{t-1} \beta\bar{\delta} + 2\bar{\delta}BL_J L_\pi \sum_{k=t-\tau}^{t}\sum_{i=t-\tau}^{k-1} \alpha\bar{\delta}B + 2\bar{\delta}BL_J\gamma^{\tau-1} \\
&\overset{(2)}{\leq} \big((2\bar{\delta}BL_g + 2L_J L_{\bar{h}})\bar{\delta}Bc\tau_{\mathrm{mix}} + 4BL_J\bar{\delta}\tau_{\mathrm{mix}} + 2\bar{\delta}^2 B^2 L_J L_\pi c\tau_{\mathrm{mix}}^2 + 2\bar{\delta}BL_J\big)\frac{1}{\sqrt{T}},
\end{aligned}
$$

where (1) owes to the update rule of the actor and (2) is followed by choosing $\tau = \tau_{\mathrm{mix}}$. Hence we conclude our proof. $\square$

