# OpenReview forum: "On the Convergence of Continuous Single-timescale Actor-critic"
_ICML.cc/2025/Conference — ICML 2025 poster_

### Official Review · Reviewer_R9ou · 2025-02-26

**Overall Recommendation:** 3

**Summary:**

The paper presents a finite time convergence analysis of the single timescale, single loop actor-critic algorithm with Markovian sampling in the discounted reward continuous state action space setting. The main contributions include - (a) extending to continuous state action space using an operator based analysis; (b) Markovian sampling for both actor and critic; and (c) novel Lyapunov function to analyze actor and critic simultaneously.

**Claims And Evidence:**

While the problem setting (continuous state action space; single timescale; Markovian sampling) considered in this paper is interesting and important to gain a theoretical understanding of, I believe the authors overclaim the novelty of the contributions of the paper in the following ways:
- new operator based analysis to handle intricacies arising from uncountable state space - the lemmas presented in Appendix B are a very straightforward extension from the discrete to continuous state action space
- Markovian sampling in the discounted setting has been already analyzed in Section 4 of [1]
- Lyapunov function to analyze the error of critic and actor together in Theorem D.3 is akin to the interconnected system analysis in Theorem 3.5 of previous work [2]

\
[1] Towards Understanding Asynchronous Advantage Actor-critic: Convergence and Linear Speedup. H. Shen, K. Zhang, M. Hong, T. Chen.  IEEE Transactions on Signal Processing, 2023.

[2] Finite-Time Analysis of Single-Timescale Actor-Critic. X. Chen, L. Zhao. NeurIPS, 2023.

**Essential References Not Discussed:**

While Shen et al. 2023 has been discussed in the Introduction, I believe it should also be added to Table 1.

**Experimental Designs Or Analyses:**

N/A

**Methods And Evaluation Criteria:**

N/A

**Other Comments Or Suggestions:**

N/A

**Other Strengths And Weaknesses:**

- Continuous state action spaces are an important problem to consider
- The paper is well written and easy to follow with appropriate background material presented and proofs well organized

**Questions For Authors:**

Please correct me if I am mistaken in my understanding of the contributions of the paper.

**Relation To Broader Scientific Literature:**

While the problem being considered is interesting, I believe the analysis techniques used and the resultant theoretical claims are a combination of methods used in prior literature. See section on Claims and Evidence for further details.

**Theoretical Claims:**

All theoretical claims are sound and are presented with well written proofs.

---

> ### Author Rebuttal · Authors · 2025-04-01
>
> **(Claim 1: on new operator-based analysis.)**
>
> Operator-based analysis is employed not only to handle the lemmas in Appendix B but also to accommodate the continuous distribution throughout the proof. For instance, it is used in establishing Propositions 3.1 and 3.2.
>
> Prior to our work, it remained unclear how to extend single-timescale actor-critic methods to continuous action spaces. This is precisely why (Chen et al, 2021) and (Chen \& Zhao, 2024) consider continuous state spaces but restrict the action space to be finite—an uncommon and limiting assumption.
>
> In analyzing the most challenging case of single-timescale actor-critic methods, one must rely on several previously established lemmas (see Appendix B) that state the regularity of the problem. However, these lemmas are derived for finite action space assume Lipschitz constants that scale with the number of actions ($|\mathcal{A}|$), which, however, becomes meaningless in the case of the uncountable continuous space. We observe that the dependence on $|\mathcal{A}|$ stems from the need to bound the evolution of the stochastic processes defined in Eq. (12) and Eq. (13) over the action space. This evolution is mainly governed by the policy $\pi_{\theta}$. We demonstrate (formally established in Proposition 4.4) that if the total variation distance between two policies is Lipschitz continuous, then the evolution of the stochastic process in the action space can be effectively controlled without assuming the finiteness of the action set. This novel insight and development allow us to extend several foundational lemmas (Appendix B) to the continuous action setting. Such extension is further enabled by the proposed operator-based analysis, which conveniently handles the continuous distribution throughout the proof, including in the verification of Propositions 3.1 and 3.2.
>
> **(Claim 2: on Markovian sampling in the discounted setting.)**
>
> We acknowledge that Markovian sampling in the discounted setting has been analyzed in [1]. Nonetheless, this does not weaken our contribution.
>
> First, several key propositions essential for our analysis are not presented in [1]. For instance, Proposition 3.1 plays a central role in our framework, yet it is not established in [1]. Even Proposition 3.2, while stated in [1], is assumed without proof. In contrast, we rigorously establish these results within a continuous setting.
>
> Furthermore, our work addresses Markovian sampling under a **single-timescale** formulation, whereas [1] considers the **two-timescale case**. As a result, the treatment of Markovian noise in our analysis is fundamentally different from those in [1]. In fact, our considered setting is significantly more challenging than the two-timescale case and requires the development of new analysis techniques.
>
> We acknowledge that [1] is a valuable contribution. As emphasized in the Introduction, the analyses of single-timescale and two-timescale actor-critic methods are fundamentally different. Since Table 1 is intended to compare only single-timescale methods, we did not include [1] in the comparison.
>
> **(Claim 3: on Lyapunov function.)**
>
> We respectfully disagree with the claim that our use of a Lyapunov function is akin to the interconnected system approach. In our analysis, we sum the critic and actor errors into a unified loss function $\mathcal{L}$ and convert all error terms into a single inequality involving $\mathcal{L}$, as shown in Eq. (33). In contrast, (Chen \& Zhao, 2024) track three distinct error terms and construct an interconnected system of three coupled inequalities. It remains unclear whether the latter case admits a Lyapunov-like formulation that combines the three errors into a single function and establishes a unified inequality.
>
> That said, we agree that from a more abstract mathematical standpoint, all such approaches can be interpreted as sequences of intricate inequality manipulations.

---

### Official Review · Reviewer_z7XF · 2025-03-12

**Overall Recommendation:** 3

**Summary:**

The paper analyzes the single-timescale actor-critic with TD(0) updates for the critic and REINFORCE with a baseline for the actor, with linear function approximation for continuous state-action spaces. The samples are taken from a single (Markovian) trajectory, while a generative model that enables an independent sample from the given initial state distribution is assumed. Under standard assumptions on the Lipschitz continuity of the policy parameterization and uniform ergodicity, convergence to a stationary point up to a function approximation error was proved.

**Claims And Evidence:**

It is a theoretical paper, and the claims are supported with corresponding statements along with their proofs.

**Essential References Not Discussed:**

The list of references seems sufficient.

**Experimental Designs Or Analyses:**

No experimental designs or analyses in the paper.

**Methods And Evaluation Criteria:**

There is no numerical study in the paper.

**Other Comments Or Suggestions:**

- The use of linear function approximation to address large state-action spaces can be specified earlier. It is mentioned for the first time in Section 3, which is a bit late.

- In (7), (9) and Algorithm 1 (Lines 4 and 5), the time index of the policy parameter \theta was not given. I guess it should be \theta_t in these equations.

**Other Strengths And Weaknesses:**

Please see my previous comments regarding the sampling.

Regarding Assumption 4.3, does a specific class of policies (e.g., log-linear) satisfy these Lipschitz continuity claims?

**Questions For Authors:**

The analysis in this paper mainly establishes convergence to a stationary point. The natural policy gradient approach (Kakade, 2001), (Cen et al., 2020), (Agarwal et al., 2021) can provide finite-time optimality guarantees up to the usual function approximation errors. Is it possible to extend this approach to the natural actor-critic (NAC) setting? In those results the iteration complexity grows at a rate O(log|A|), and it would be quite interesting to see whether it can be possible to achieve convergence to an optimal policy in a continuous action space where |A|=\infty.

**Relation To Broader Scientific Literature:**

The paper is complementing the theoretical study of single-timescale actor-critic methods in two directions: (1) continuous/uncountable state-action spaces, and (2) Markovian sampling.

The main challenges that are inherent to continuous state-action spaces can be more elaborated. In its current form, given the uniform Lipchitz continuity of the parametric policies, linear function approximation and uniform ergodicity, the analysis seems similar to the actor-critic methods for countable state-action spaces. What makes it particularly challenging and different in this particular setting, and how were these challenges addressed? This could be very important.

Secondly, the Markovian sampling for \widehat{O}_t in (9) requires independent samples from the initial state distribution \eta with probability (1-\gamma). If one has access to this mechanism, i.i.d. sampling can also be performed. As such, the need to use such a simulator weakens the sampling argument in this paper. Can this requirement to have i.i.d. samples from \eta be removed?

**Theoretical Claims:**

I checked the proofs of the main theorems. They seem to be correct.

---

> ### Author Rebuttal · Authors · 2025-04-01
>
> **(Q1: challenges in dealing with continuous action space.)**
>
> Previous results derived for finite action space assume Lipschitz constants that scale with the number of actions ($|\mathcal{A}|$), which, however, becomes meaningless in the case of the uncountable continuous space. We observe that the dependence on $|\mathcal{A}|$ stems from the need to bound the evolution of the stochastic processes defined in Eq. (12) and Eq. (13) over the action space. This evolution is mainly governed by the policy $\pi_{\theta}$. We demonstrate (formally established in Proposition 4.4) that if the total variation distance between two policies is Lipschitz continuous, then the evolution of the stochastic process in the action space can be effectively controlled without assuming the finiteness of the action set. This novel insight and development allow us to extend several foundational lemmas (Appendix B) to the continuous action setting. Such extension is further enabled by the proposed operator-based analysis, which conveniently handles the continuous distribution throughout the proof, including in the verification of Propositions 3.1 and 3.2.
>
> **(Q2: requirement to have i.i.d. samples from $\eta$.)**
>
> Thank you for your question. Note that existing analysis requires i.i.d. samples from the discounted state visitation distribution for updating the actor, not from the initial distribution. The former is unknown. It is difficult to directly sample it, even approximately, in a simulator given its form. The initial distribution is a predefined distribution over states, and its i.i.d sampling is easy. The key challenge here is how to sample from the unknown discounted state visitation distribution approximately.
> The assumption of a known initial distribution is standard in the discounted setting. As shown in Eq. (3), the objective function is defined as the integral of the value function $V_\theta(s)$, weighted by the initial state distribution $\eta$, over the state space. Accordingly, we assume access to $\eta$, which allows i.i.d. sampling from this distribution.
> This requirement might be removed by identifying a class of behavior policies for state sampling that are guaranteed to provide a good approximation to the true value functions and policy gradient estimate. The existing Markov chain $(\pi_\theta, \widehat{P})$ may serve as a special example to examine the desired characteristics that such distributions should satisfy. It will be our future research.
>
> **(Q3: on Assumption 4.3.)**
>
> Log-linear policies satisfy this assumption under the condition that the feature map $\phi(s,a)$ is bounded and the action space is finite. Moreover, as noted in our manuscript, certain continuous policy classes—such as the uniform distribution, truncated Gaussian distribution, and Beta distribution—can also satisfy this assumption.
>
> **(On other suggestions.)**
>
> Thanks for your advice. We will mention the use of linear function approximation to address large state-action spaces earlier. We will fix the time index of the policy parameter $\theta$, which is $\theta_t$.
>
> Thank you for highlighting this interesting problem. Establishing optimality guarantees for AC or NAC methods typically requires the underlying optimization problem to be convex or to satisfy properties such as gradient domination. We view the convergence of AC and NAC to an optimal policy in continuous action spaces as an important direction for future work.

---

### Official Review · Reviewer_4oXm · 2025-03-13

**Overall Recommendation:** 3

**Summary:**

This paper addresses the theoretical understanding of single-timescale actor-critic (AC) algorithms in continuous state-action spaces, a widely used reinforcement learning (RL) approach for continuous control tasks such as robotics. While actor-critic methods have demonstrated empirical success, existing theoretical analyses largely focus on finite state-action spaces and impractical simplifications like double-loop updates or two-timescale learning, which introduce artificial decoupling between the actor and critic. The paper aims to close this gap by establishing finite-time convergence guarantees for the canonical single-timescale AC algorithm with Markovian sampling, where both the actor and critic update simultaneously.

To achieve this, the authors use a Lyapunov-based convergence analysis framework, which offers a unified and less conservative characterization of both the actor and the critic.

**Claims And Evidence:**

This is a theoretical paper so the claim is that they provide a state of the art sample complexity of the actor critic algorithm with  a single loop, no i.i.d assumption and continuous state action space.

**Essential References Not Discussed:**

To my knowledge there do not seem to be any significant references missed by the authors.

**Experimental Designs Or Analyses:**

There are no experiments performed as the paper is purely theoretical in nature.

**Methods And Evaluation Criteria:**

No methods or evaluations are given as the paper is theoretical in nature.

**Other Comments Or Suggestions:**

I would suggest the authors to explore removing the linear function assumption on the value function by implementing a local linearization method as laid out in (Ke et al., 2024)

An improved finite-time analysis of temporal difference learning with deep neural networks, Z Ke, Z Wen, J Zhang: ICML 2024.

**Other Strengths And Weaknesses:**

The one place I think there can be improvement is in the linear function assumption on the value function. That makes the result in my opinion less relevant than some existing ones where the value function is not restricted to be linear functions.

**Questions For Authors:**

None

**Relation To Broader Scientific Literature:**

The paper is essentially extending the analyses (Olshevsky & Gharesifard, 2023) and  (Chen & Zhao, 2024). it describes a  Lyapunov analysis framework similar to the one in (Tian et al., 2023) to obtain global convergence for a single loop actor critic with continous state action spaces.

Convergence of actor-critic with multi-layer neural networks, H Tian, A Olshevsky, Y Paschalidis , NeurIPS 2023.

**Theoretical Claims:**

The theorteical claims of the paper seem to be sound.

---

> ### Author Rebuttal · Authors · 2025-04-01
>
> **(On removing linear function assumption)**
>
> Thank you for your suggestion regarding the removal of the linear function approximation assumption for the value function. We are aware of recent works (e.g., Tian et al., 2023; Ke et al., 2024) that employ deep neural networks for value function approximation. However, these approaches often rely on additional assumptions (e.g., Tian et al., 2023) to ensure theoretical tractability. We plan to further investigate these results and explore how to extend our analysis to broader classes of function approximators under milder assumptions.

---

### Official Review · Reviewer_DMy1 · 2025-03-16

**Overall Recommendation:** 3

**Summary:**

This paper considers the problem of analyzing actor-critic algorithms for the discounted, continuous spaces setting when the actor and critic updates occur on the same timescale. The key idea in the analysis is to sample from two distinct processes: a "discounted process" corresponding to the discounted state occupancy measures of the sequence of policies, and an "undiscounted process" corresponding to the undiscounted state occupancy measures of those policies (the "discounted/undiscounted process" terminology is mine, not the paper's). Samples from the discounted process are used to perform actor updates, while samples from the undiscounted process are used to perform critic updates in Algorithm 1. Convergence results that recover state-of-the-art rates proved for less general settings are provided for Algorithm 1 under the assumption that linear function approximation is used for the critic.

---
**Post-rebuttal comment:** after the author rebuttal, I maintain that the proposed approach does not achieve the "holy grail" described in **Strengths** below. Nonetheless, after the rebuttal more clearly situated their analysis within that of the literature on discounted AC analyses, I am more convinced that the paper provides a useful, meaningful step in this direction. I am increasing my score accordingly.

**Claims And Evidence:**

The claims made are broadly supported by the results provided, with the caveats discussed in the Weaknesses part of **Strengths and Weaknesses** below.

**Essential References Not Discussed:**

None, to my knowledge.

**Experimental Designs Or Analyses:**

n/a

**Methods And Evaluation Criteria:**

n/a

**Other Comments Or Suggestions:**

n/a

**Other Strengths And Weaknesses:**

**Strengths**
Finite-time analysis of actor-critic algorithms has seen a great deal of activity over the past several years. The holy grail of such analyses is to establish finite-time convergence of the actor-critic algorithm most common in practice: a single sample generated from the same stochastic process is used to update both the actor and the critic (single-sample), and the actor and critic update stepsizes are constant factor multiples of one another (single-timescale). This has been achieved for the average-reward setting, but to my knowledge has remained open for the discounted setting. This paper partially addresses this gap, and the topic of the paper is therefore definitely of interest to the theoretical reinforcement learning (RL) community. Though the scheme of eq. (9) for sampling from the discounted occupancy measure has long been known (see, e.g., the thesis [Konda, 2002] or more recently [Zhang et al., 2020a]), the "operator-based analysis" used to establish the counterpart of the uniform ergodicity property provided in Proposition 3.2 is interesting and the proposition itself is of potentially broader usefulness to the RL researchers interested in policy gradient analyses.


**Weaknesses**
My primary concern is the nature of sampling procedure introduced in Algorithm 1 to forcibly and artificially "decouple" the discounted and undiscounted processes used in the actor and critic estimation procedures. Due to this, the algorithm under consideration is not single-sample, and no longer tracks the "holy grail" of finite-time actor-critic analyses described in **Strengths** above. One of the attractive features of the analysis of [Chen & Zhao, 2024] for the average-reward setting is its ability to directly cope with the coupling between the actor and critic in the single-sample regime and not resort to artificial means to force decoupling. The approach of the present work falls short of this, analyzing an impractical algorithm that does not resemble what is used in practice. This negatively impacts the signficance of the contribution and its potential relevance to the community.

**Questions For Authors:**

1. The "new operator-based analysis" results presented in Appendix B appear to be restatements of existing results (e.g., from [Chen & Zhao, 2024], [Chen et al., 2021], [Zhang et al., 2020a]). What are the key technical innovations that were required in extending these results to the continuous state and action space setting?
2. If line 5 of Algorithm 1 is correct as is, then the sequence $\{ \hat{O}_t \}$ does not follow the desired discounted process from eq. (13). Should $s_t$ be replaced by $\hat{s}_t$ in line 5 of Algorithm 1?
3. Algorithm 1 appears to require that two distinct stochastic processes $\{ O_t \}$ and $\{ \hat{O}_t \}$ be simulated in parallel and that the simulator used to generate $\{ \hat{O}_t \}$ allow arbitrary resets. Is this correct? If so, can you comment on how this algorithm relates to others considered in the literature?
4. If the current work is not "single-sample" for the reasons described above, can you elaborate on either (i) how the current work lays important foundations for subsequent development of a "holy grail" single-sample, single-timescale analysis, or (ii) why a single-sample analysis is not possible for the discounted setting. If the current work *is* single-sample, can you explain how?

**Relation To Broader Scientific Literature:**

The paper proposes and analyzes a single-timescale -- but not single-sample -- actor-critic algorithm for continuous spaces. The related works section adequately situates these results within the relevant context.

**Theoretical Claims:**

The proof sketch outlined in the main body appears sound. I did not verify the results in the appendix in detail.

---

> ### Author Rebuttal · Authors · 2025-04-01
>
> **(Weakness \& Q4: on single-simple)**
>
> 1. The terminology ''single-sample" follows the seminal work [Olshevsky \& Gharesifard, 2023], where they directly assume sampling from visitation distribution and stationary distribution for updating actor and critic, respectively. It refers to the fact that at each iteration, the critic and the actor are each updated using a single sample.
> 2. Our analysis can accommodate the case where ''a single sample generated from the same stochastic process is used to update both the actor and the critic'', as suggested by the reviewer. To achieve this, we can modify the sampling scheme so that the critic is also updated using the same samples from the Markov chain $(\pi_\theta, \hat{P})$ as the actor. We would like to highlight that our theoretical analysis would still apply under this alternative sampling scheme. The current presentation of sampling from $(\pi_\theta, P)$ for the critic update simply follows the existing setup in the literature of on-policy actor-critic analysis. The proposed modification can be viewed as an extension to analyze a special off-policy version.
> 3. **Sampling from two Markov processes does not force the decoupling or simplify the analysis.** In fact, our analysis directly copes with the coupling between the actor and critic, which can be seen in the Proof Sketch. In particular, the term $I_5$ in the critic error analysis is coupled with the actor error and is ultimately bounded by $\mathcal{O}(\mathbb{E}[\\|\Delta_t\\| \\|\nabla J(\theta_t)\\|])$ in Eq.(26). Conversely, $ I_4$ in the actor error analysis is coupled with the critic error and simplifies to $\mathcal{O}(\mathbb{E}[\\|\nabla J(\theta_t)\\| \\|\Delta_t\\|])$ in Eq.(30). Due to this mutual dependence, we define a novel Lyapunov function that captures both the actor and the critic errors, enabling a unified analysis of their convergence.
> 4. **Our work is by far the most practical analysis**, in the sense that we address the most widely used single-timescale _discounted reward formulation_, and our analysis does not require sampling from unknown distributions (i.e., the stationary distribution and the discounted state visitation distribution). Moreover, as mentioned in Point 2, our analysis still holds with minor modifications to accommodate the case of using the same samples for updating both critic and actor.
>
> **(Q1: Key innovations for extending previous results to continuous space)**
>
> Previous results derived for finite action space assume Lipschitz constants that scale with the number of actions ($|\mathcal{A}|$), which, however, becomes meaningless in the case of the uncountable continuous space. We observe that the dependence on $|\mathcal{A}|$ stems from the need to bound the evolution of the stochastic processes defined in Eq. (12) and Eq. (13) over the action space. This evolution is mainly governed by the policy $\pi_{\theta}$. We demonstrate (formally established in Proposition 4.4) that if the total variation distance between two policies is Lipschitz continuous, then the evolution of the stochastic process in the action space can be effectively controlled without assuming the finiteness of the action set. This novel insight and development allow us to extend several foundation lemmas (Appendix B) to the continuous action setting. Such extension is further enabled by the proposed operator-based analysis, which conveniently handles the continuous distribution throughout the proof, including in the verification of Propositions 3.1 and 3.2.
>
> **(Q2: on the notation)**
>
> Thanks for pointing out the typo. It should be $\hat{P}(\cdot | \hat{s}_t, \hat{a}_t)$.
>
> **(Q3: Relation to existing actor-critic analyzed in the literature)**
>
> Yes, it is correct. This is the same sampling scheme also employed in [Shen et al. 2023]. However, their analysis only handles the two-timescale AC algorithm on finite state-action space. In Konda's thesis, this artificial MDP is only constructed to utilize its property that its average reward equals the original MDP's discounted reward induced by $(\pi_\theta, P)$. It's not analyzed for convergence. In addition, [Zhang et al. 2020a] employ a different sampling scheme to approximate the discounted occupancy measure. However, their approach relies on a multi-sample procedure.  Moreover, they also require a simulator with a state-reset function.
>
> Existing literature on analyzing AC requires samples from the unknown visitation distribution for updating actor, which requires a simulator as well. [Shen et al. 2023] needs to assume the same simulator as ours. For i.i.d. sampling [Chen et al., 2021; Olshevsky \& Gharesifard (2023)], one has to run the simulator for sufficiently many steps under the current policy $\pi_{\theta_t}$ to approximate the visitation distribution. This inevitably takes a very long time and requires significant modification of the simulator as well. But our setting only requires a reset, which is much simpler and more time-efficient.

---

### Decision · Program_Chairs · 2025-05-01

**Decision:**

Accept (poster)

**Comment:**

This paper presents a finite-time convergence analysis for single-timescale actor-critic algorithms with Markovian sampling in continuous state and action spaces. The analysis leverages a Lyapunov-based framework to jointly analyze the actor and critic updates, enabling a less conservative and more unified treatment of their dynamics.

All reviewers agree on the correctness of the theoretical claims and acknowledge the significance of the problem setting. The authors did a good job clearly situating their analysis in the rebuttal session and resolved most of the reviews' concerns. I would encourage the authors to incorporate the proposed revisions in their camera-ready version